# Simulated Meteorological Impacts of Offshore Wind Turbines and Sensitivity to the Amount of Added Turbulence Kinetic Energy

Daphne Quint[1,2], Julie K. Lundquist[1,2,3], Nicola Bodini[2], and David Rosencrans[1]

[1]Department of Atmospheric and Oceanic Sciences, University of Colorado Boulder, Boulder, Colorado, 80309-0311, United States
[2]National Renewable Energy Laboratory, Golden, Colorado, 80401, United States
[3]Johns Hopkins University, Baltimore, Maryland, 21218, United States

**Correspondence:** Julie K. Lundquist (julie.lundquist@jhu.edu)

**Abstract.** Offshore wind energy projects are currently in development off the east coast of the United States and may influence the local meteorology of the region. Wind power production and other commercial uses in this area are related to atmospheric conditions, and so it is important to understand how future wind plants may change the local meteorology. In the absence of measurements of potential wind plant impacts on meteorology, simulations offer the next-best possible insight into wake effects on boundary-layer height, temperature, fluxes, and wind speeds. However, simulation tools that capture these effects offer multiple options for representing the amount of turbine-added turbulence that may impact assessments of micrometeorological effects. To explore this sensitivity, we compare one year of simulations from the Weather Research and Forecasting (WRF) model with and without wind plants incorporated, focusing on the lease area south of Massachusetts and Rhode Island. The simulations with wind plants are repeated to include both the maximum and minimum amounts of added turbulence to provide bounds on the potential impacts. We assess changes in wind speeds, 2 m temperature, surface heat flux, turbulence kinetic energy (TKE), and boundary layer height during different stability classifications and ambient wind speeds over the entire year, and compare results for the degree of added turbulence in the wind plant simulations. Because the wake behavior may be a function of boundary-layer stability, in this paper, we also present a machine learning algorithm to quantify the area and distance of the wake generated by the wind plant. This analysis enables us to identify the relationship between wake extent and boundary-layer height. We find that hub-height wind speed is reduced within and downwind of the wind plant, with the strongest impacts occurring during stable conditions and faster wind speeds in region 3 of the turbine power curve, although impacts lessen as wind speeds increase past 15 m s$^{-1}$. In contrast, wind speeds near the surface decrease when no turbine-added turbulence is included, but can increase for stably stratified conditions when 100% of possible TKE is included in the simulations. TKE increases at hub height in the simulations with added TKE for all stability classes, suggesting that atmospheric stability does not immediately modify the TKE generated by turbines. Negligible changes in hub-height TKE manifest in the simulations without the added TKE. At the surface, TKE increases in the simulations with maximum added turbulence only for unstable conditions. In the no-added-turbulence simulations, surface TKE decreases slightly in neutral and unstable simulations. Differences in 2 m temperatures and surface heat fluxes are small, but vary considerably with atmospheric stability and the amount of added TKE. Boundary-layer heights increase within the wind plant when turbine-added turbulence is included, and decrease slightly downwind during stable conditions. In contrast, with no added turbulence, the boundary

layer height is in general reduced in stable conditions with wind speeds less than 15 m s$^{-1}$ and slightly increased in neutral conditions. Finally, shallower upwind boundary-layer heights tend to correlate with larger wake areas and distances, though other factors likely also play a role in determining the extent of the wind plant wake. These simulation-based results provide a bound for micrometeorological impacts of wind plant wakes: simulations that couple the atmosphere to the ocean may reduce
these impacts, and we await observational verification.

*Copyright statement.* This work was authored in part by the National Renewable Energy Laboratory, operated by Alliance for Sustainable Energy, LLC, for the U.S. Department of Energy (DOE) under contract no. DE-AC36-08GO28308. Funding was provided by the U.S. Department of Energy Office of Energy Efficiency and Renewable Energy Wind Energy Technologies Office. The views expressed in the article do not necessarily represent the views of the DOE or the U.S. Government. The U.S. Government retains and the publisher, by
accepting the article for publication, acknowledges that the U.S. Government retains a nonexclusive, paid-up, irrevocable, worldwide license to publish or reproduce the published form of this work, or allow others to do so, for U.S. Government purposes.

## 1 Introduction

Wind plants, also known as wind farms, along the northeastern U.S. coast are projected to undergo rapid expansion in the coming years, aiming to achieve a capacity of 30 GW by 2030 and 110 GW by 2050 (U.S. Department of Energy, 2023).
Currently, 27 areas are leased along the mid-Atlantic Outer Continental Shelf for future offshore wind development (Bureau of Ocean Energy Management, 2023). Wind plants produce wakes that can influence local environments. Only a limited set of observations have quantified the effects of wind plants on local meteorology, and so a combination of observational and modeling studies have been used to assess micrometeorological impacts of wakes.

The aggregation of individual wind turbine wakes into a wind plant wake results in a wind speed deficit downwind of the
wind plant (Christiansen and Hasager, 2005; Platis et al., 2018, 2020), with the strongest deficit generally occurring near hub-height (Abkar and Porté-Agel, 2015) or in the top half of the turbine rotor disk (Vanderwende et al., 2016). Stronger and larger wakes are also associated with stable conditions, moderate wind speeds, low boundary-layer heights, and low ambient turbulence kinetic energy (TKE) (Christiansen and Hasager, 2005; Dörenkämper et al., 2015; Platis et al., 2018; Lundquist et al., 2019; Fischereit et al., 2022a; Pryor et al., 2021; Bodini et al., 2021; Rosencrans et al., 2024). TKE is enhanced
near wind plants, but this increase diminishes rapidly downwind (Bodini et al., 2021). The most pronounced enhancement in TKE typically occurs in the upper half of the turbine rotor disk near the rotor tip where the largest shear occurs (Abkar and Porté-Agel, 2015; Vanderwende et al., 2016). The degree of TKE increase is influenced by atmospheric stability: in unstable conditions, ambient TKE levels are high, resulting in only a slight increase in TKE in the presence of wind plants; in contrast, during stable conditions, wind plants significantly increase TKE levels (Bodini et al., 2021).
Wind plants can also cause changes in surface temperatures. Wind plants tend to cause a warming at the surface during stable conditions (Baidya Roy and Traiteur, 2010; Fitch et al., 2013; Rajewski et al., 2014; Siedersleben et al., 2018a; Golbazi et al., 2022). The hypothesis proposed by Baidya Roy and Traiteur (2010) suggests that enhanced vertical mixing induced by

wind turbine wakes causes this warming in stably stratified conditions. During stable conditions, temperature increases with height, so enhanced mixing at the surface would result in warmer air from aloft mixing with cooler air at the surface. This warming effect in stable conditions has been observed in wind plants near peat bogs in Scotland (Armstrong et al., 2016), wind plants in cropland in Iowa (Rajewski et al., 2014), a wind plant in farmland/grassland in China (Luo et al., 2021), a marsh in North America (Wu and Archer, 2021), and other locations. In complex terrain, however, this warming in stable conditions is not always observed (Moravec et al., 2018). Satellite observations also suggest apparent surface warming due to the mixing of warm air down to the surface in stably stratified conditions (Zhou et al., 2012b, a; Walsh-Thomas et al., 2012; Harris et al., 2014; Slawsky et al., 2015; Xia et al., 2016; Chang et al., 2016). The exact nature of the wind plant warming effect is likely due to complex interactions between the ambient atmosphere, the wind turbine wake, and surface characteristics. The height of the wind turbine can also influence temperature changes in the wind plant. Golbazi et al. (2022) compared surface temperature changes for turbines with a hub height of 81 m to turbines with hub heights of 119 m and 250 m. Warming was observed below the turbine for both cases; however, for the bigger turbines in this study, warming did not extend all the way to the surface. Instead, slight cooling was found at the surface.

Changes in surface temperatures are coupled with heat flux changes at the surface. When conditions are unstable, heat fluxes are typically positive (upward). Similarly, during stable conditions when warmer air lies over cooler air, heat fluxes are negative (downward) (Stull, 1988). According to Golbazi et al. (2022), the magnitude of the heat flux decreases in areas experiencing cooling. This weakening is indicated by a positive change when the atmosphere is stable and a negative change when the atmosphere is unstable. As part of the Crop Wind-Energy Experiment campaign, Rajewski et al. (2014) found that turbines have the largest influence on surface heat flux at night, in which a larger transport of heat was observed as warm air from aloft was brought to the surface by turbine-induced mixing. In contrast, during the day, the turbine-induced mixing was weaker than the ambient boundary-layer-scale mixing, resulting in inconsistent signals of warming and cooling differences between times with and without turbines. The large-eddy simulations of Lu and Porté-Agel (2011) also find that turbines induce strong changes on heat and momentum fluxes in a stable boundary layer.

In some circumstances, wind turbines' effects can also influence a deeper region of the atmosphere. The planetary boundary layer (PBL) height (PBLH) is the height of the layer of the atmosphere that is influenced by the surface. Wind plants increase the boundary-layer height in stable (Lu and Porté-Agel, 2011) and neutral (Wu and Porté-Agel, 2017) conditions. As wind turbines extract energy from incoming flow, kinetic energy is entrained from the flow above, resulting in an increased boundary layer depth. An internal boundary layer (IBL) develops above the wind plant as relatively slower flow at rotor height moves upwards, interacting with slightly faster air. The IBL grows; it may eventually interact with the free atmosphere and modify the PBLH (Wu and Porté-Agel, 2017; Gadde and Stevens, 2021). The growth of the IBL is dependent on the stratification of the boundary layer with which it interacts. During situations with weak stratification in the free atmosphere above the PBL, the IBL can nudge the PBL to deeper heights. However, when the layer above the PBL is strongly stratified, the IBL cannot necessarily push the PBL higher (Wu and Porté-Agel, 2017). Because the vertical growth of the boundary layer is limited for these strongly stratified conditions, flow instead goes around the wind plant in the horizontal direction (Gadde and Stevens, 2021).

The relationship of the boundary-layer height with wake length is also an area of active research. Simulations of Pryor et al. (2021) in this region found that lower PBL heights favor longer and more intense wakes, though stable conditions and low ambient TKE also increase wake strength and extent. In the German bight, large-eddy simulations Maas and Raasch (2022) also find longer wakes for shallower boundary layers, with some wakes exceeding 100 km.

Given the scarcity of comprehensive offshore observations along the U.S. East Coast, simulation-based studies are essential to assess potential wake impacts. This study aims to complete the first year-long evaluation of how modeled offshore wind plants influence the modeled local environment. We achieve this by comparing Wweather Research and Forecasting (WRF) model (Skamarock et al., 2021) simulations with and without the inclusion of wind plants. Our analysis focuses on the Massachusetts-Rhode Island offshore wind lease area, where we quantify the difference in hub-height and 10-m wind speed, TKE at hub height and at the surface, 2-m temperature, surface heat flux, and boundary-layer height. This focus on meteorology distinguishes this contribution from that of Rosencrans et al. (2024), who focus on hub-height wind speed and power production impacts.

Further, we explore the sensitivity of these results to the details of representation of wind turbines in the numerical weather prediction (NWP) model. The representation of wind plant in a mesoscale NWP model generally includes two effects: increased drag and increased added turbulence. Determining the appropriate amount of turbine-generated turbulence to incorporate into mesoscale wind plant parameterizations is an ongoing research question. The initial recommendation by Fitch et al. (2012) was to apply 100% of the added TKE. In later work, Archer et al. (2020) recommended a value of 25 % based on their large-eddy simulations of an individual turbine in neutrally-stratified simulations. More recent comparison with field observations, which incorporate a bug fix and corrected TKE advection (Larsén and Fischereit, 2021; Ali et al., 2023) both suggest that 100 % added TKE yields better agreement with observations, although García-Santiago et al. (2024) and Larsén et al. (2024) more recently suggest that 100 % over-represents added TKE. Given this uncertainty in the literature and the need to assess possible impacts, we explore two sets of wind plant simulations: one applying the maximum (100 %) and another the minimum (0 %) amount of added TKE, providing bounds on the possible wind plant wake impacts on local micrometeorology. In addition to the representation of turbine-added turbulence, atmospheric stability conditions are a critical factor in determining the simulated micrometeorological impacts, which also vary with different wind speeds as turbine operation changes. Finally, we assess the relationship between boundary-layer height and the extent of wind plant wakes, hypothesizing that deeper boundary layers will limit the extension of these wakes.

Given the scenario above, the following research questions guide this study:

  – What are the year-long impacts of offshore turbines on simulated local meteorology?

  – How does atmospheric stability influence the results?

  – How does varying the amount of added TKE in the WRF wind farm (or plant) parameterization (WFP) affect the results of the above questions?

  – Can a reliable method be developed to automatically estimate wake characteristics from WRF WFP simulations?

– What is the relationship between simulated boundary-layer height and the extent of simulated wind plant wakes?

In Sect. 2, we describe the dataset used for our analysis, and in Sect. 3 we introduce our method to determine the extension of the wakes using machine learning. We present our results in Sect. 4 and provide conclusions and suggested future work in Sect. 5.

## 2    Dataset

In this study, we use the NOW-WAKES dataset introduced in Rosencrans et al. (2024) as part of the 2023 National Offshore Wind dataset (Bodini et al., 2024b). Full details on the dataset can be found in Rosencrans et al. (2024), but a brief summary is provided here. We note that the WRF setup used here resulted from a comparison of 16 different WRF setups against an observational dataset (Bodini et al., 2024b); this setup was the best performer. The dataset was created using WRF version 4.2.1, and the wind plant parameterization of Fitch et al. (2012) updated with the bug fix of Archer et al. (2020) but with 100 % added TKE and 0 % added TKE to bound results, rather than 25 % added TKE as suggested by Archer et al. (2020). We use

results from the innermost domain (domain 2) of the two nested domains, bounded by 76.208W - 64.977W and 37.389N - 42.137N (Fig. 1). Domain 2 has a 2 km horizontal resolution and a 10m vertical resolution near the surface with stretching aloft. There were 34 vertical levels in the lowest 2,000 m, and 29 in the lowest 750 m. NOW-WAKES covers from 1 September 2019 00:00 UTC - 31 August 2020 23:50 UTC (chosen to overlap with lidar data availability in the region) at 10-minute output

resolution; we used hourly time steps for our analysis. The Rosencrans et al. (2024) domain is consistent with other datasets for this region (Xia et al., 2022; Redfern et al., 2023; Bodini et al., 2024b) and was initially chosen to optimize processor partitioning for the WRF simulations.

We consider three sets of NOW-WAKES simulations (Table 1). The first simulation ("no wind farm", NWF) does not include any wind turbines, and has been validated in comparison to floating lidar observations at two locations in the domain

(Rosencrans et al., 2024). The second and third simulations consider a wind turbine layout with turbines from all lease areas: LA100, which uses the 100 % added TKE factor, and LA0, which is the simulation with 0 % added TKE factor. For this analysis, we compare the NWF simulation to the LA100 and LA0 simulations, which use the same turbine layout, as this layout is most relevant for the region of interest marked in Fig. 1. Fig. 1 shows the locations of the wind turbines for each of these simulations. The turbines introduced in these simulations are 12 MW with a rotor disk extending from 30 m to 245 m

and a hub-height of 138 m. The power, thrust coefficient, and power coefficient curves for these turbines appear in Fig. 2. As explained in Rosencrans et al. (2024), individual turbines are sited at regular 1 nm, or 8.6 rotor diameters, spacing including an additional 0.5 nm from lease area boundaries. This spacing results in generally one turbine per cell (in 1018 cells), but 177 cells have two turbines, 6 cells have three turbines, and 7 cells have 4 turbines.

These turbine parameters are used in the Fitch WFP to represent the effects of turbines on the flow via two mechanisms,

drag and added turbulence. The drag force is:

$$\mathbf{F_{drag}} = \frac{1}{2}C_T\rho|\mathbf{V}|\mathbf{V}A \qquad (1)$$

where $\mathbf{V}$ is the horizontal wind speed, $C_T$ is the wind-speed-dependent turbine thrust coefficient, $\rho$ is the air density, and $A$ is the cross-sectional rotor area. The fraction of mean kinetic energy converted into TKE is a function of the turbine's thrust coefficient, $C_T$, the turbine's power coefficient, $C_P$, and a wind plant TKE factor, $\alpha$, by:

$$C_{TKE} = \alpha(C_T - C_P) \tag{2}$$

and the turbine-induced TKE tendency is given by:

$$\frac{\partial TKE_{ijk}}{\partial t} = \frac{0.5 N_{ij} C_{TKE} |\mathbf{V_{ijk}}|^3 A_{ijk}}{z_{k+1} - z_k} \tag{3}$$

where $i$, $j$, and $k$ are the zonal, meridional, and vertical grid cell indices, respectively; $N_{ij}$ is the turbine number density for a given cell; $|\mathbf{V}|_{\mathbf{ijk}}$ are the wind speed components; $A$ is the turbine rotor area; $C_{TKE}$ is the unitless TKE coefficient; and $z$ is the model level height. To assess the effect of including turbine-generated TKE, separate simulations with 0% ($\alpha$ = 0) and 100% ($\alpha$ = 1.0) added TKE are available; we consider the 100% TKE simulation as a default as comparisons to the large-eddy simulationsof Vanderwende et al. (2016) and García-Santiago et al. (2024) both suggest that added TKE is critical. In Vanderwende et al. (2016), when TKE generation within the wind plant parameterization is disabled, the model produced very different wind speeds, wind directions, and turbulence as compared to large-eddy simulations, with too small values of turbulence and too large of decreases in wind speed. Regardless, we include the 0 % simulations to provide a lower bound on the effect of added turbulence. Computational resources were not available to consider intermediate options like the 25 % option recommended by Archer et al. (2020).

**Table 1.** Summary of the three NOW-WAKES simulations considered in our analysis.

| Simulation Type | Acronym | Turbine Type | Time Period | Added TKE | $C_{TKE}$ | # Turbines |
|---|---|---|---|---|---|---|
| No wind plants | NWF | N/A | 09/01/2019 - 08/31/2020 | N/A | N/A | 0 |
| All lease areas | LA100 | 12 MW | 09/01/2019 - 08/31/2020 | 100% | 1.0 | 1418 |
| All lease areas | LA0 | 12 MW | 09/01/2019 - 08/31/2020 | 0% | 0.0 | 1418 |

## 2.1 Quantities of interest

We focus on assessing the impact of offshore wind turbines on wind speed, TKE (modeled in WRF as QKE, or twice the TKE), 2 m temperature, surface heat flux, and planetary boundary layer height. Figures of average modeled hub-height wind speed, 10 m wind speed, surface TKE, and 2 m temperature in each season are available in the appendix (Figs. A1, A2, A3, and A4). All other quantities are discussed in further detail here.

We assess the impact of wind plants on surface heat flux. Positive surface heat flux values refer to the transfer of heat upwards, which is common during unstable conditions when warmer air underlies cooler air. Negative surface heat flux values

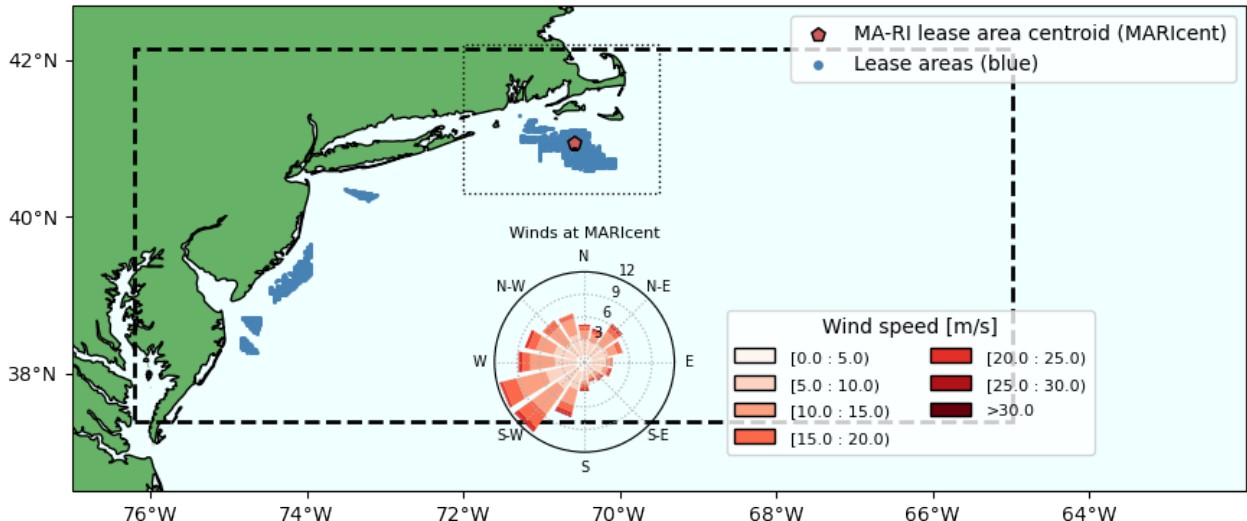

**Figure 1.** Simulation domain, including turbine locations and a wind rose near the centroid of the Massachusetts-Rhode Island lease area (MARIcent). The MARIcent location is marked in red. The turbine locations are shaded, and all wind plants included in the lease area simulations are in blue. The boundary of the NOW-WAKES innermost domain is outlined in the dashed black line. The dotted gray line indicates the region of interest for this study. The wind rose shows the wind speed (shading) and direction (angle) at 130 m over the 1 year modeled in NOW-WAKES. The distance from the center of the rose indicates the percentage of values in each bin.

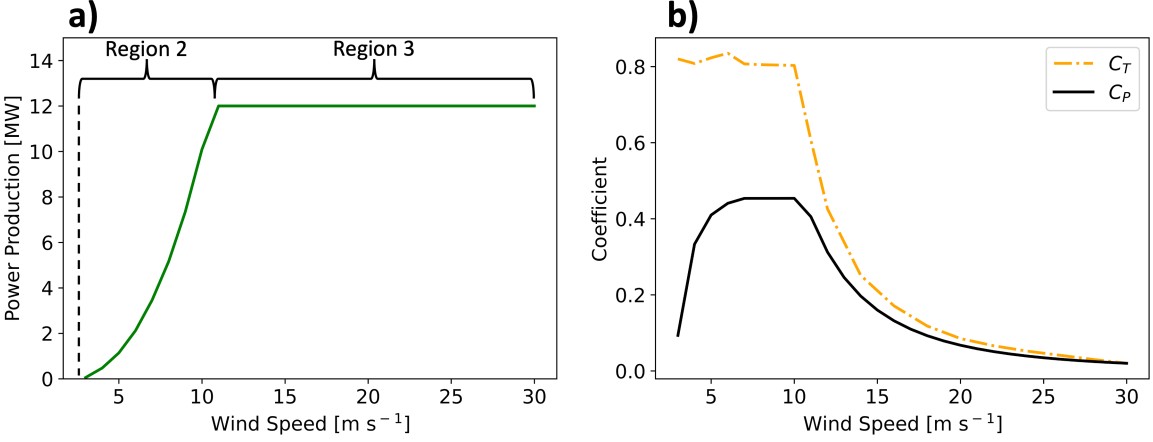

**Figure 2.** Characteristics of the 12-MW turbine used in the NOW-WAKES simulations. (a) The power curve and (b) curves showing the thrust coefficient ($C_T$) in dashed orange and power coefficient ($C_P$) in solid black. Figure courtesy of Rosencrans et al. (2024).

are common in stable conditions, and refer to the transfer of heat downwards. Heat flux varies seasonally and by location (Fig. 3). In spring and summer over water, heat fluxes are often negative due to the more frequent occurrences of stable conditions.

Heat fluxes are also relatively consistent across the region. In fall and winter, heat fluxes are positive due to more frequent unstable stratification. Heat fluxes during these months vary with distance from the coast, becoming more strongly positive further from the coast.

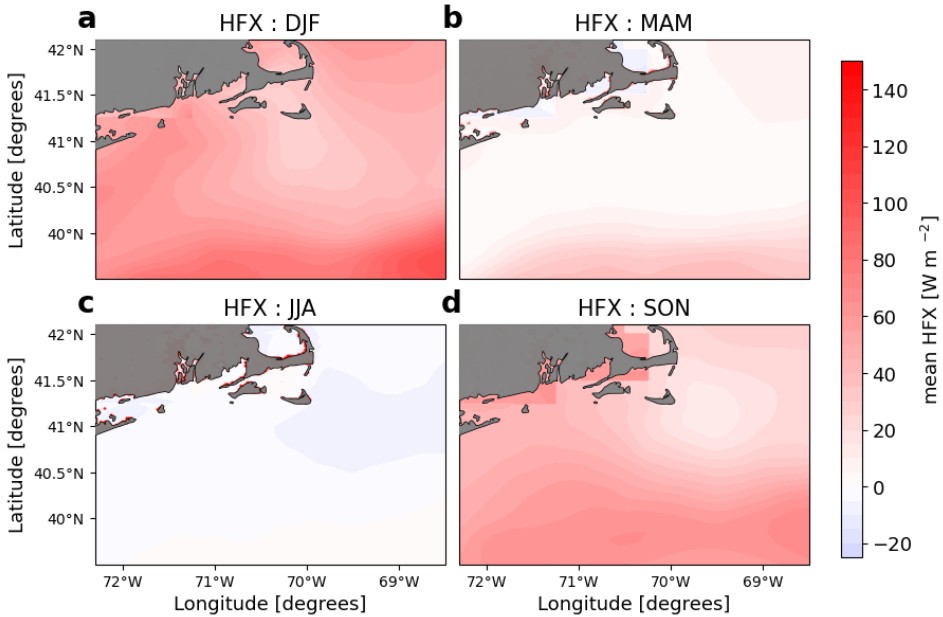

**Figure 3.** Average surface heat flux in the region from the NWF simulation is shown for each season from September 2019 – August 2020. (a) December, January and February (DJF) (b) March, April, and May (MAM) (c) June, July, and August (JJA) (d) September, October, and November (SON).

We also consider how wind plants influence turbulence at hub height and at the surface. WRF outputs QKE, which is defined as 2 times the TKE, and is the relevant variable from the Mellor–Yamada– Nakanishi–Niino (MYNN) PBL scheme used here. In the MYNN PBL scheme, TKE created by sub-grid processes such as wind plants is not always advected horizontally, but TKE advection (Archer et al., 2020) is turned on in these simulations (Rosencrans et al., 2024). TKE estimates the intensity of turbulence and is produced mechanically by wind shear or by buoyant thermals. The change in TKE over time depends on

advection, shear generation, buoyant production, turbulent transport, and the viscous dissipation rate (Nakanishi and Niino, 2009), and, in the LA100 simulations here, production due to wind turbine operation. Hub-height TKE varies seasonally and by location (Fig. 4). In the spring and summer, TKE at hub height is small due to the prevalence of stable conditions. In the fall and winter, TKE is larger due to more frequent unstable conditions as cold air advects over colder water. (A discussion of how atmospheric stability is determined is found below in Section 2.2.)

Wind turbines generally operate within the boundary layer, and thus may influence and be influenced by the height of the PBL. Deeper planetary boundary layer heights are associated with faster wake recovery due to the larger volume of air that can be drawn into the wake (Maas and Raasch, 2022; Pryor and Barthelmie, 2024). In the MYNN PBL scheme used in the

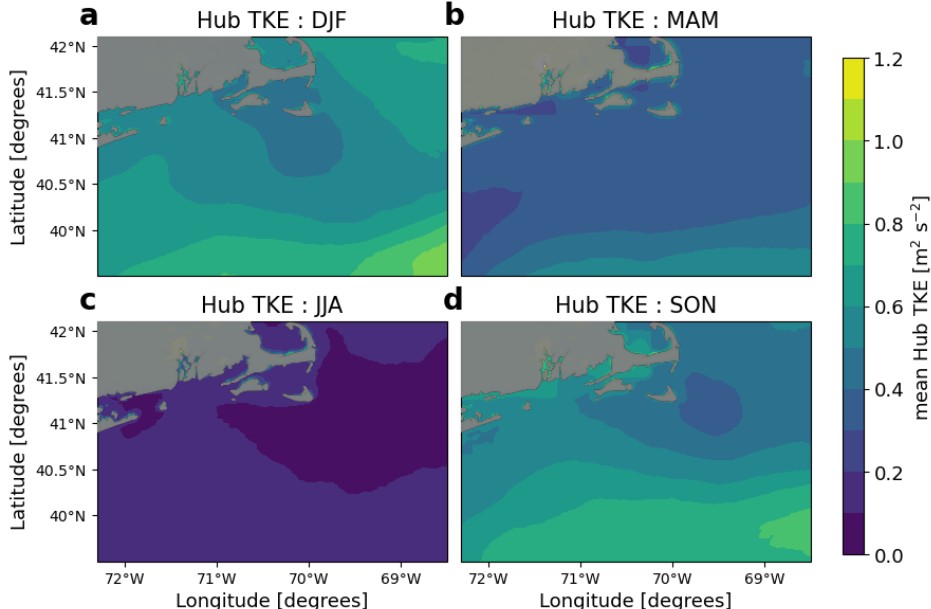

**Figure 4.** Same as in Fig. 3, but for hub-height TKE.

NOW-WAKES WRF simulations, the PBLH is derived from a blend of a potential temperature and TKE-based definitions, depending on the stability of the atmosphere (Olson et al., 2019). For stable conditions, most relevant for significant wakes, the TKE definition dominates, and the boundary layer height is the height at which turbulent motions drop to 5% of the surface TKE value (Olson et al., 2019). During neutral and unstable conditions, the potential temperature definition dominates, and the boundary-layer height is the height at which the surface inversion ends. The algorithm of Nielsen-Gammon et al. (2008) determines the minimum virtual liquid water and ice potential temperature in the lowest 200 m of the atmosphere ($\theta_{vli\_min}$). Then, the boundary layer height is the height at which $\theta_{vli} = \theta_{vli\_min} + \Delta\theta_{vli}$, where $\Delta\theta_{vli}$ is set to be 0.5 K over water and 1.25 K over land.

In this region, the average PBLH varies spatially and seasonally (Fig. 5). Mean PBL heights range from 112 m to 1,086 m. During the summer, when stable conditions are more common, the shallowest PBL heights occur, with a mean PBL height of 453 m. Unstable conditions in the fall and winter lead to the deepest PBL heights, with mean values of 715 m in the fall and 698 m in the winter. PBL heights generally increase with distance from the coast. In the summer and fall, a minimum in PBLH is observed to the east of Nantucket.

## 2.2 Partitioning of the dataset

In this study, we quantify how hub-height wind speed, boundary-layer height, 2 m temperature, surface heat flux, and turbulence kinetic energy at the surface and hub height vary between the NWF, LA100, and LA0 simulations. While presenting our results,

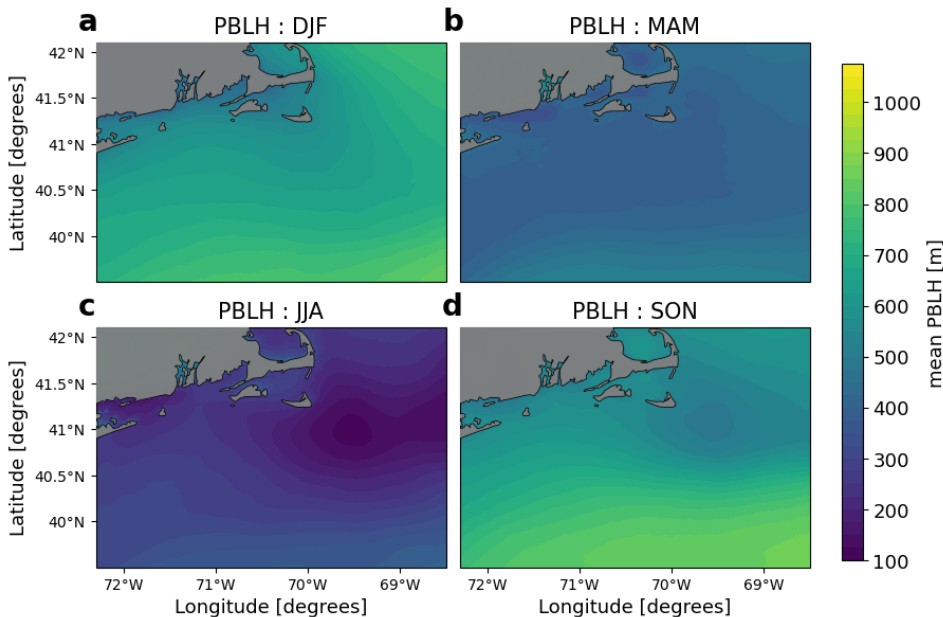

**Figure 5.** Same as in Fig. 3, but for PBLH.

**Table 2.** The dataset is partitioned in seven different ways, using labels in Column 1. The stability classification, wind speed range, wind direction range, and total number of hours in each partition are listed in Columns 2, 3, 4, and 5, respectively. Figures considering stable conditions with wind speed between 0-3 m s$^{-1}$ are not shown.

| Label | Stability | Wind speeds [m s$^{-1}$] | Wind directions [degrees] | Total # of hours |
|-------|-----------|--------------------------|---------------------------|------------------|
| a | stable | all | all | 3898 |
| b | unstable | all | all | 3899 |
| c | neutral | all | all | 987 |
| - | stable | 0-3 (Region 1) | 180-270 | 61 |
| d | stable | 3-11 (Region 2) | 180-270 | 1157 |
| e | stable | 11-15 (slower end of Region 3) | 180-270 | 697 |
| f | stable | 15+ (faster end of Region 3) | 180-270 | 637 |

we highlight the variability of such changes as a function of atmospheric stability and wind speed at hub height (Table 2), both evaluated at the centroid of the Massachusetts-Rhode Island lease area (MARIcent, Fig. 1).

Wakes tend to be stronger and last longer in stably stratified conditions (Lundquist et al., 2019; Pryor et al., 2021; Rosencrans et al., 2024). To test that behavior here, we classify each time period with its surface-based atmospheric stability, determined

by calculating the Obukhov length ($L$):

$$L = -\frac{u_*^3 \overline{\theta_v}}{\kappa g(\overline{w'\theta'})} \tag{4}$$

where $u_*$ is the friction velocity, $\theta_v$ is the virtual potential temperature, $\kappa$ is the von Kármán constant of 0.4, $g$ is gravitational acceleration, and $\overline{w'\theta'}$ is the vertical turbulent surface heat flux. The virtual potential temperature is calculated from the 2-m temperature, the surface pressure, and the surface water vapor mixing ratio. Values between 0 m and 500 m are considered stably stratified conditions, and values from -500 m to 0 m are considered unstable. Values outside of this range are considered neutral (Gryning et al., 2007; Sathe et al., 2011). We note that while the inverse Obukhov Length can be output directly from 225 WRF, we calculate it directly from the fluxes as discussed in Quint et al. (2025).

We note that the variability of atmospheric stability often reflects changes in wind direction in the region (Fig. 6). Winds in this region are often southwesterly and parallel to the coastline, but other wind directions do occur in the region (Fig. 1). Stable conditions almost always occur when winds are southwesterly. Wind directions vary more for unstable and neutral conditions: in unstable conditions, winds tend to be northerly, with a preference for northwesterly flow; in neutral conditions, winds are 230 typically westerly or northeasterly. Because of this correlation of stability and wind direction, further exploration of results as a function of wind direction is not presented, as they are redundant with the results as a function of atmospheric stability.

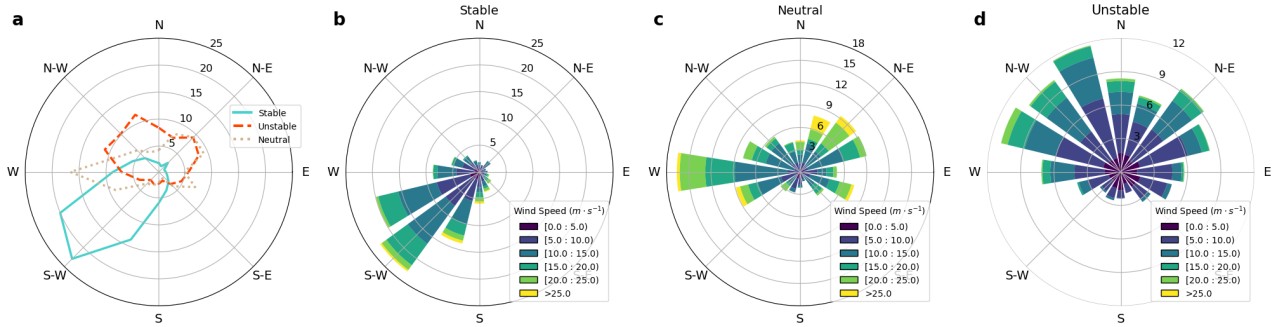

**Figure 6.** (a) The frequency of each 22.5-degree wind direction bin for each stability classification. Wind roses for stable, neutral, and unstable conditions are shown in panels b, c, and d, respectively. In all panels, radial distance from the center refers to the percentage of values in each 22.5 degree bin.

We also partition the dataset by considering four wind speed ranges based on the power curve and thrust coefficient curve of the turbines used in the NOW-WAKES simulations (Fig. 2): 0-3 m s$^{-1}$ (Region 1), 3-11 m s$^{-1}$ (Region 2), 11-15 m s$^{-1}$ (Region 3 with changing thrust coefficient), and 15+ m s$^{-1}$ (Region 3 with small thrust coefficient). These modeled turbines 235 do not produce any power when winds are 0-3 m s$^{-1}$. Then, power production steadily rises as wind speeds increase from 3-11 m s$^{-1}$ (Region 2 on the power curve). At 11 m s$^{-1}$, turbine blades begin to pitch, resulting in the rated power production of the turbine being reached and not increasing further, while the thrust coefficient decreases between 11-15 m s$^{-1}$ (Region 3). At wind speeds faster than 15 m s$^{-1}$, the turbine is operating at full power but with a small thrust coefficient, which has implications on the character of the wake. Cases with wind speeds below 3 m s$^{-1}$, when the turbine is not operating and the

240 primary impact is from the small amount of drag exerted by the standing thrust coefficient of the turbine, are not shown as there are few such cases. To more clearly identify the effect of the variability in hub-height wind speed on wake impacts from those caused by differences in stability and wind direction, when showing the variability of the wake impacts with hub-height wind speed, we only consider periods with stable conditions and southwesterly (180 - 270 °) winds.

## 3    Characterizing wind plant wake area and length with machine learning

We characterize the wind plant wake area and wake length for each of the 3,898 hours of the year with a stably stratified atmosphere. We first determine the difference in wind speed between the LA100 and NWF simulations. The wind speed $U$ is calculated at the model level closest to the turbine hub height ($\sim 138$ m). Waked locations are identified as points with a wind speed deficit of at least 1 m s$^{-1}$. This wake definition is stronger than the 0.5 m s$^{-1}$ threshold used in Golbazi et al. (2022); Rybchuk et al. (2022); Rosencrans et al. (2024), and was chosen to aid in identifying contiguous wakes. A relative 250 wake definition proved problematic by making the wake field even noisier.

Next, the points that show a wind speed deficit that should be attributed to the wind plant wake need to be identified. As an example, in Fig. 7a, the wind plant wake should be contained in the cluster of points around the wind plants, but some other locations distant from wind plants, not contiguous with the wind plant wake, also show a wind speed deficit of at least 1 m s$^{-1}$. The deficits at these locations distant from the wind plants are presumed to be numerical noise as identified in Ancell et al. 255 (2018); Lauridsen and Ancell (2018) and discussed in Appendix F of Rosencrans et al. (2024) and by Golbazi et al. (2022). To handle these situations, we employ the Density-Based Spatial Clustering of Applications with Noise (DBSCAN) method (Ester et al., 1996) to accurately identify the wind plant wake. The algorithm groups points together based on their distance to dense regions. DBSCAN requires two main parameters. The first is the maximum distance between two points for one to be considered in the neighborhood of another. The second parameter is the minimum number of points in a neighborhood for a 260 point to be considered as a core point. Here, we use a maximum distance value of 0.1 degrees and a minimum points value of 30. Using these parameters, points are grouped into clusters that are assigned a label, as in Fig. 7b and 7e.

The next step is to select the one cluster at each time step that is likely to be the wind plant wake. For each cluster identified by the DBSCAN algorithm, we determine the percentage of points in the cluster that are within the wind plant. All clusters with fewer than 15% of points in the wind plant are excluded from consideration. Then, the cluster with the most points overall 265 is selected. In Fig. 7b, the small clusters -1, 0, 1, and 3-7 in the east of the domain are not contiguous with the wind turbines and lack 15% of their points within the wind plant, so they are excluded from consideration as the wind plant wake.

Once the wind plant wake cluster is selected, we create a concave hull shape around its points (Fig. 7c and 7f). We define the area of the wind plant wake as the area of this concave hull polygon. The wake length is the maximum distance from MARIcent to any bounding point of the identified wind plant wake polygon (red outline in Fig. 7c and 7e).

Many cases have a clearly defined wake and a straightforward wake detection process, as shown in Fig. 7d, 7e and 7f. However, 15.2% of hours have ill-defined wakes, so that they are flagged by our algorithm and not included in the final results of our analysis. An hour can be flagged if either of the following two situations occurs:

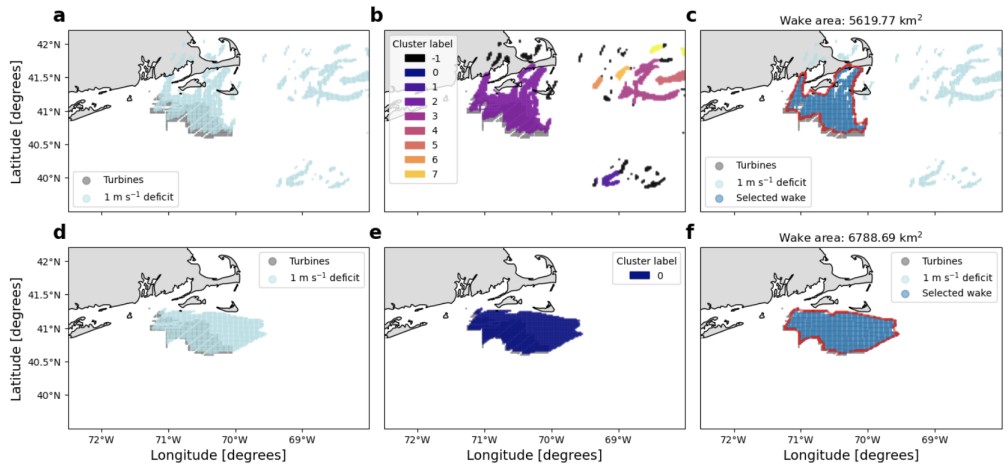

**Figure 7.** Example of the three steps taken to identify the wind plant wake on 14 December 2019 at 22:00 UTC (a, b, and c) and on 04 February 2020 at 07:00 UTC (d, e, and f) in the LA100 simulations. Turbines are marked in gray. (a and d) All points identified with a 1 m s$^{-1}$ deficit are marked in light blue. (b and e) The points with a wind speed deficit are clustered, and a label is assigned to each cluster by the DBSCAN algorithm. The -1 label indicates points classified as noise. (c and f) The selected wind plant wake is in darker blue, with its wake boundary in red.

1. Less than 40% of the coordinates located in the wind plant are classified as part of the wind plant wake. This ensures that the identified wind speed deficit is connected to the wind plant, which is the source of the wake.

2. The ratio of the number of points in the wind plant wake cluster to the total number of points with a wind speed deficit of at least 1 m s$^{-1}$ is less than 0.2. This flags exceptionally noisy times when the selected wake represents a small number of the locations with a significant wind speed deficit.

In addition, no wake is identified at all if there are less than 4 points with a wind speed deficit greater than 1 m s$^{-1}$, or if less than 5% of points identified by DBSCAN are located within the wind plant. An additional 5.5% of hours are excluded from 280   our analysis due to these criterion.

   We demonstrate this flagging process in Fig. 7. While the case in the top row of Fig. 7 has some noise, the identified wake clearly originates from the wind plant and covers most of the wind plant area. In addition, the points not selected for the wake represent a small fraction of all points with a wind speed deficit greater than 1 m s$^{-1}$. On the other hand, Fig. 8 shows two examples of cases that are flagged by our algorithm: in (a), fewer than 40% of the wind plant is covered by the wake; in (b), 285   the identified wake includes less than 20% of all the locations with a wind speed deficit greater than the 1 m s$^{-1}$ threshold.

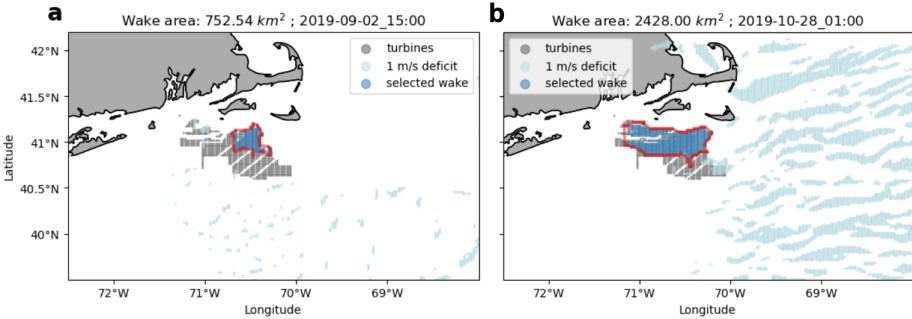

**Figure 8.** Example of two cases that are flagged by the wake detection algorithm and therefore not considered for further analysis. These plots follow the same convention as the right panel in Fig. 7.

## 4 Results

### 4.1 Wind plant wake impacts on wind speed

#### 4.1.1 Hub-height wind speed

Hub-height wind speeds in wakes are reduced most in stable conditions and least in unstable conditions (Fig. 9a and 9c for
LA100; Fig. 10a and 10c for LA0), regardless of the amount of added TKE. In the LA100 simulations, within the wind plant, average wind speeds are reduced by up to 2.7 m s$^{-1}$ in stable conditions, and up 1.5 m s$^{-1}$ in unstable conditions, as turbines extract momentum from the flow. In stable conditions, there is also a significant decrease in wind speed downwind of the wind plant (to the northeast, given the dominant southwesterly wind direction during stable condition; see Fig. 6), while in neutral and unstable conditions the extent of the wake is much smaller. During unstable conditions, wind speeds are replenished faster
consistent with increased mixing from aloft (Abkar and Porté-Agel, 2013), which reduces the extent of the wake, as also reported in Fig. 12 of Rosencrans et al. (2024).

For both LA100 and LA0, in stable conditions and when the wind is from the southwest, hub-height wind speeds are reduced more with increasing ambient wind speeds (Figs. 9d, 9e, 10d, 10e) up to about 15 m s$^{-1}$, with the wake effect receding slightly at wind speeds faster than 15 m s$^{-1}$ due to the decreasing thrust coefficient (Figs. 9f and 10f). Only differences of 0.2 m
s$^{-1}$ exist between the simulations with and without turbines for quiescent winds, when wind turbines are not operational, as expected, as the only turbine influence is that of the standing thrust coefficient (not shown). In LA100, hub-height wind speeds are reduced by up to 2.5 m s$^{-1}$ for wind speeds in Region 2 of the turbine power curve, and up to 3.6 m s$^{-1}$ for wind speeds above 11 m s$^{-1}$ in Region 3 of the power curve. For both considered modeled setups, the wake also extends further for faster wind speeds: a wind speed deficit of at least 0.5 m s$^{-1}$ extends out to the northern edge of Nantucket Island for wind speeds
between 3 and 11 m s$^{-1}$ (Figs. 9d and 10d), and several kilometers further than this for faster wind speeds (Figs. 9e,f and 10e,f).

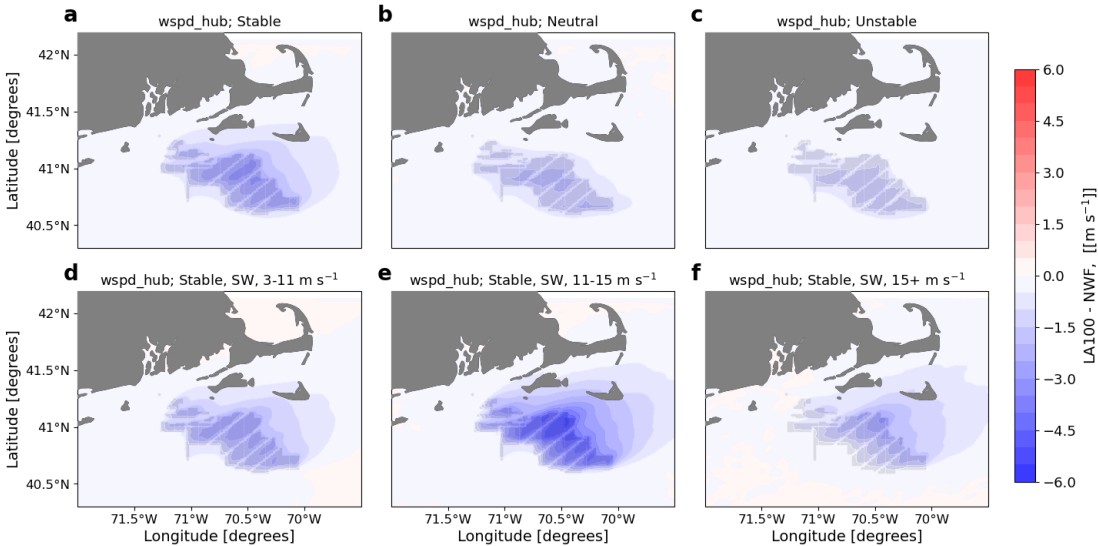

**Figure 9.** Mean hub-height wind speed difference between the LA100 and NWF simulations in (a) stable conditions, (b) neutral conditions, (c) unstable conditions, (d) 3-11 m s$^{-1}$, (e) 11-15 m s$^{-1}$, and (f) 15+ m s$^{-1}$. Panels d, e, and f are only during times with stable conditions and southwest winds. Turbines are marked in gray.

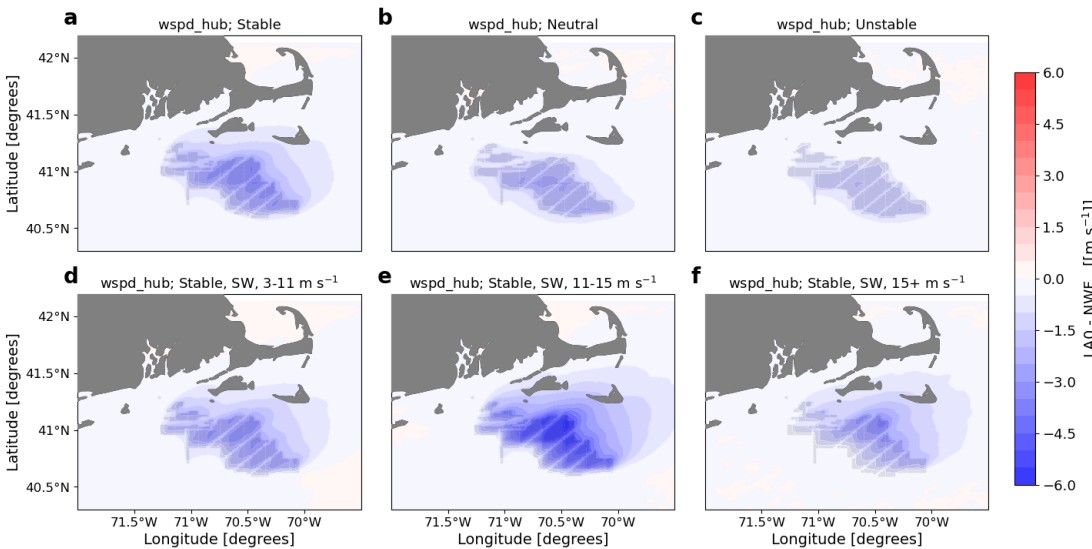

**Figure 10.** Mean hub-height wind speed difference between the LA0 and NWF simulations in (a) stable conditions, (b) neutral conditions, (c) unstable conditions, (d) 3-11 m s$^{-1}$, (e) 11-15 m s$^{-1}$, and (f) 15+ m s$^{-1}$. Panels d, e, and f are only during times with stable conditions and southwest winds. Turbines are marked in gray.

These results are consistent with previous observational studies, that found hub-height wind speeds decrease within and downwind of the wind plant, with the largest impacts occurring during stable conditions and for faster ambient wind speeds at the slower end of Region 3 (11 -15 m s$^{-1}$). Wind speed deficits in the wake of wind plants are well documented for cases both onshore and offshore (Christiansen and Hasager, 2005; Fitch et al., 2013; Dörenkämper et al., 2015; Platis et al., 2018; Fischereit et al., 2022a). In the central United States, Fitch et al. (2013); Rajewski et al. (2013); Smith et al. (2013) also find that the largest wind speed deficits occur for faster wind speeds, stable conditions, and when turbulent mixing is inhibited. For individual turbines, the magnitude of the wake also varies with wind speed (Rhodes and Lundquist, 2013): maximum wake wind speed deficits occur near rated power when the turbine thrust coefficient is near its maximum value, with decreasing wake wind speed deficits as wind speeds increase and thrust coefficient decreases.

### 4.1.2  10-m wind speed

Wind plant wake impacts on 10 m wind speed depends on the stability of the atmosphere and the amount of turbulence added by the wind farm parameterization, with the most significant differences occurring in stable conditions (Figs. 11a,d,e,f and 12a,d,e,f). When 100 % TKE is added, 10 m wind speeds accelerate within the wind plant by up to 0.4 m s$^{-1}$ in stably stratified conditions, with the largest accelerations in fastest wind speeds (Fig. 11f). In contrast, when no turbulence is added by the wind farm parameterization, 10 m wind speeds slow by up to 1.0 m s$^{-1}$. In neutral and in unstable conditions, 10 m wind speed decreases in both LA100 (Fig. 11b,c) and LA0 conditions (Fig. 12b,c).

The acceleration near the surface within the wind plant in stable conditions for the LA100 simulations can be understood as acceleration around an obstacle in stably-stratified flow: when the flow cannot pass through the rotor disk and cannot rise above the rotor disk due to stable stratification, it must pass through a more confined region (under the rotor disk) and therefore accelerates to conserve mass. In neutral or unstable conditions, vertical motion is not constrained and so the flow does not need to accelerate under the rotor disk. The difference in this response between the LA100 and the LA0 simulations could be due to the fact that the larger amount of TKE enhances the obstacle nature of the rotor disks. This difference in the surface wind impact of added TKE is consistent with the onshore observations and simulations of Bodini et al. (2021), the observations of Rajewski et al. (2013), and the large-eddy simulations of Vanderwende et al. (2016).

Outside of the wind plant, 10 m wind speeds are reduced in all stability classifications for both amounts of added TKE, with the largest deficit in stable conditions, as also observed for hub-height wind speed. The spatial extent of the wake is smaller for unstable conditions than for neutral and stable conditions: during unstable conditions, wind speeds are replenished faster due to increased mixing from turbines above, which reduces the extent of the wake. Under stable conditions and southwesterly winds, the downwind deficit in 10 m wind speeds generally increases more with increasing ambient wind speeds up to 15 m s$^{-1}$ (Figs. 11d, e, f, and 12d, e, f). Downwind of the wind plant, 10 m wind speeds are reduced by up to 0.60 m s$^{-1}$ for wind speeds in Region 2 of the turbine power curve, and up to 0.70 m s$^{-1}$ for wind speeds 11-15 m s$^{-1}$. As already noted for hub-height wind speed, when the wind speeds exceed 15 m s$^{-1}$ and the thrust coefficient gets very small, the wind speed deficit begins to decrease again. The wake also extends further for faster wind speeds: a wind speed deficit of at least 0.1 m s$^{-1}$ extends out

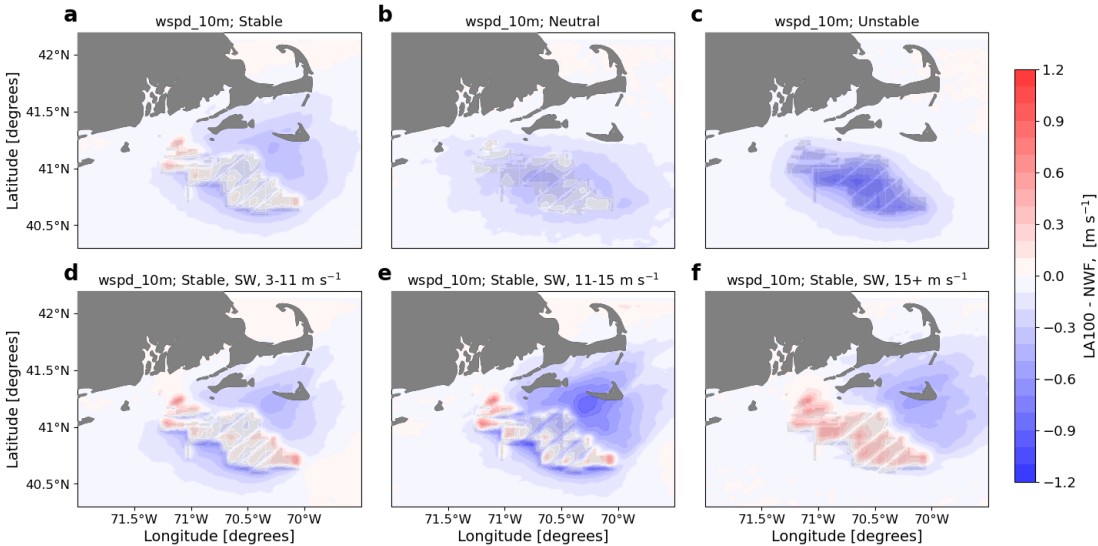

**Figure 11.** Mean 10 m wind speed difference between the LA100 and NWF simulations in (a) stable conditions, (b) neutral conditions, (c) unstable conditions, (d) 3-11 m s$^{-1}$, (e) 11-15 m s$^{-1}$, and (f) 15+ m s$^{-1}$. Panels d, e, and f are only during times with stable conditions and southwest winds. Turbines are marked in gray.

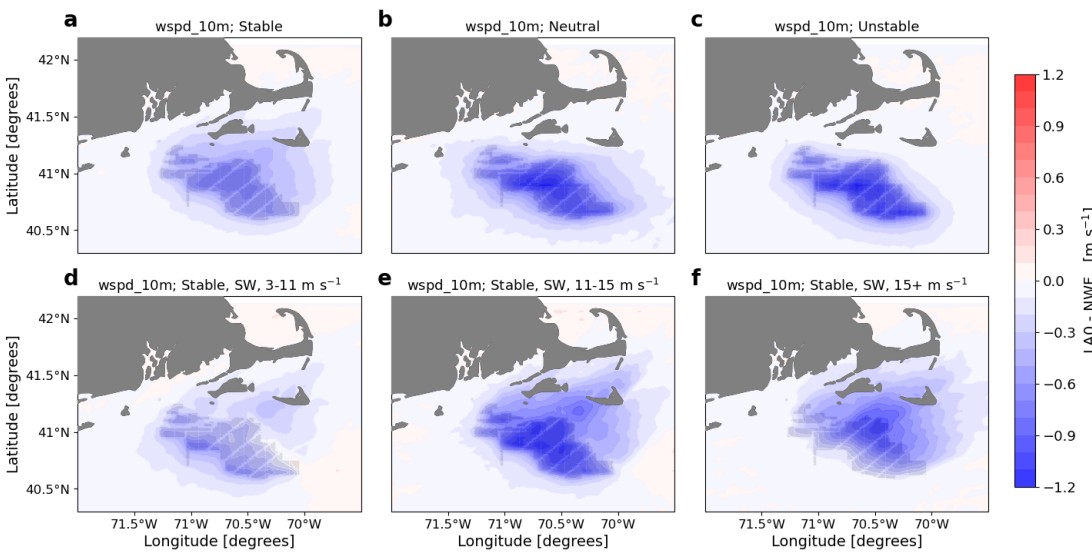

**Figure 12.** Mean 10 m wind speed difference between the LA0 and NWF simulations in (a) stable conditions, (b) neutral conditions, (c) unstable conditions, (d) 3-11 m s$^{-1}$, (e) 11-15 m s$^{-1}$, and (f) 15+ m s$^{-1}$. Panels d, e, and f are only during times with stable conditions and southwest winds. Turbines are marked in gray.

340 to the northern edge of Nantucket Island for moderate wind speeds, and several kilometers further than this for faster wind speeds.

In the stable cases with only southwesterly flow (Fig. 11d, e, f, and Fig. 12d, e, f), a slight reduction in wind speeds upwind of the wind plant emerges. The extent of this effect is slightly larger for moderate wind speeds than for stronger wind speeds. Upwind of the wind plant, wind speeds are reduced slightly, especially when winds are in Region 2 of the power curve and

345 the thrust coefficient is near its maximum value. Wu and Porté-Agel (2017) identifies this effect in large-eddy simulations of neutral boundary layers.

In general, downwind wake wind speed deficits at 10 m are much smaller than at hub-height: wind speed reductions in the wake at the surface are about 4.5 times weaker than at hub-height. At 10 m, wind speeds accelerate slightly (less than 1 m s$^{-1}$) within the wind plant during stable conditions only, a phenomenon that was also observed in the simulations and lidar

350 observations of Bodini et al. (2021) and the large-eddy simulations of Vanderwende et al. (2016); García-Santiago et al. (2024).

### 4.2 Wind plant wake impacts on TKE

In the simulations with added TKE (LA100), wind turbines impact TKE at both hub height and at the surface, with no sizeable impacts on TKE in the LA0 simulations, which inject no TKE. The wind turbine parameterization injects turbulence into the simulations at the altitudes of the turbine rotor disks, with more turbulence injected into cells with more turbines.

355 #### 4.2.1 Hub-height TKE

In the vicinity of the wind plants, TKE increases at hub height in the simulations with added TKE (Fig. 13). A similar magnitude in this increase occurs for all stability classes, suggesting that stability does not immediately modify the TKE generated by turbines (Fig. 13a,b, and c). The largest increases in TKE occur in the grid cells populated by turbines, where the turbulence is directly introduced by the WRF wind farm parameterization. This hub-height TKE increase rapidly erodes downwind of the

360 wind plants. The amount of added turbulence directly relates to the number of turbines in each grid cell, resulting in a grid pattern of larger TKE values corresponding to cells with more turbines. More added TKE is apparent at wind speeds between 11-15 m s$^{-1}$ (Fig. 13e) than at wind speeds faster than 15 m s$^{-1}$ (Fig. 13f) because the amount of added TKE is a function of the difference between $C_T$ and $C_P$ (Fig. 2) and wind speed itself (Equation 3). When no TKE is added in the simulations, negligible changes in hub-height TKE manifest as expected (Fig. 14).

365 #### 4.2.2 Surface TKE

Differences in surface TKE are subtler than differences at hub height but show more variability with stability and with the amount of added TKE (Figs. 15 and 16). During stable conditions, TKE at the surface is largely unaffected by the presence of wind turbines. Vertical mixing is suppressed during stable conditions, making it unlikely that turbulence from the turbines, injected at the rotor disk altitudes, can reach the surface. During neutral conditions, when some mixing could occur between the

370 surface and the hub height, the LA100 simulations (Fig. 15b) show a slight increase in surface TKE, while the LA0 simulations

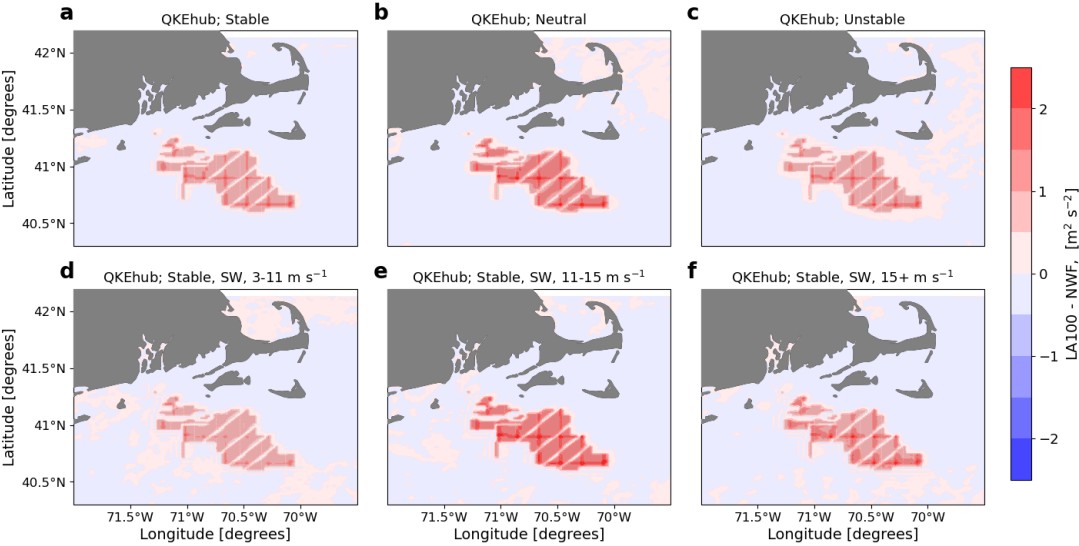

**Figure 13.** Mean hub-height QKE difference between the LA100 and NWF simulations in (a) stable conditions, (b) neutral conditions, (c) unstable conditions, (d) 3-11 m s$^{-1}$, (e) 11-15 m s$^{-1}$, and (f) 15+ m s$^{-1}$. Panels d, e, and f are only during times with stable conditions and southwest winds. Turbines are marked in gray.

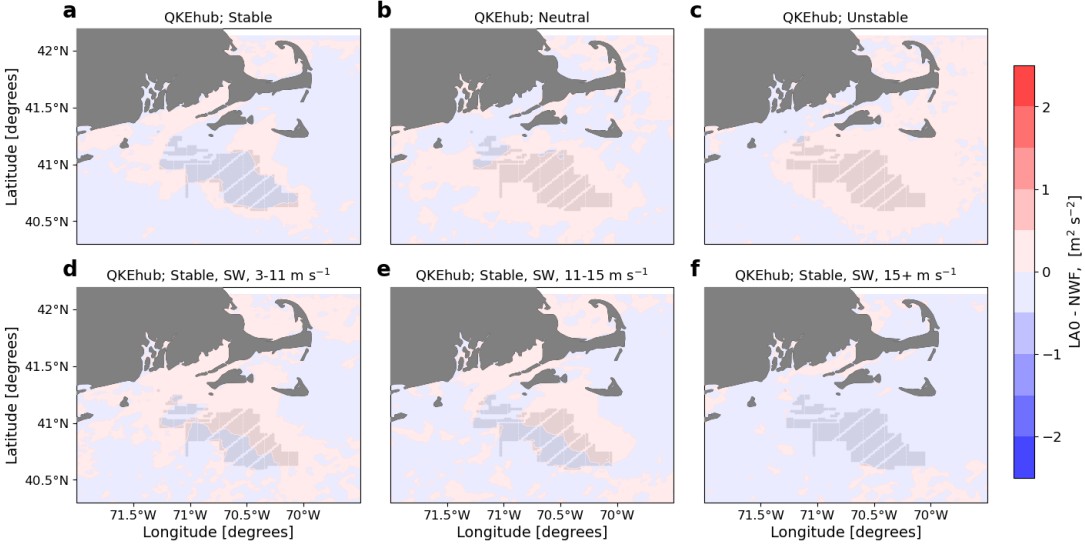

**Figure 14.** Mean hub-height QKE difference between the LA0 and NWF simulations in (a) stable conditions, (b) neutral conditions, (c) unstable conditions, (d) 3-11 m s$^{-1}$, (e) 11-15 m s$^{-1}$, and (f) 15+ m s$^{-1}$. Panels d, e, and f are only during times with stable conditions and southwest winds. Turbines are marked in gray.

(Fig. 16b) show a decrease in surface TKE. These LA100 surface TKE increases within the wind plant during neutral conditions are limited to areas close to turbines, but the LA0 decreases in surface TKE extend kms downwind, suggesting that this decrease is due to the drag effect of the wind plant reducing shear-generated turbulence, as shown previously by Xia et al. (2019).

Contrasts between the LA100 and LA0 simulations also occur during unstable conditions, where for LA100 TKE increases throughout the entire lease area (Fig. 15c), albeit by a factor of four less than at hub height, thanks to enhanced vertical mixing that causes the TKE injected at hub height to also reach the surface. Surface TKE again decreases for the LA0 unstable simulations, although with a weaker decrease than in the neutral conditions. Finally, regardless of the amount of added TKE, wake regions outside of the wind plant all show slight decreases in surface TKE (Fig. 15 b, e, and f and Fig. 16b, e, and f).

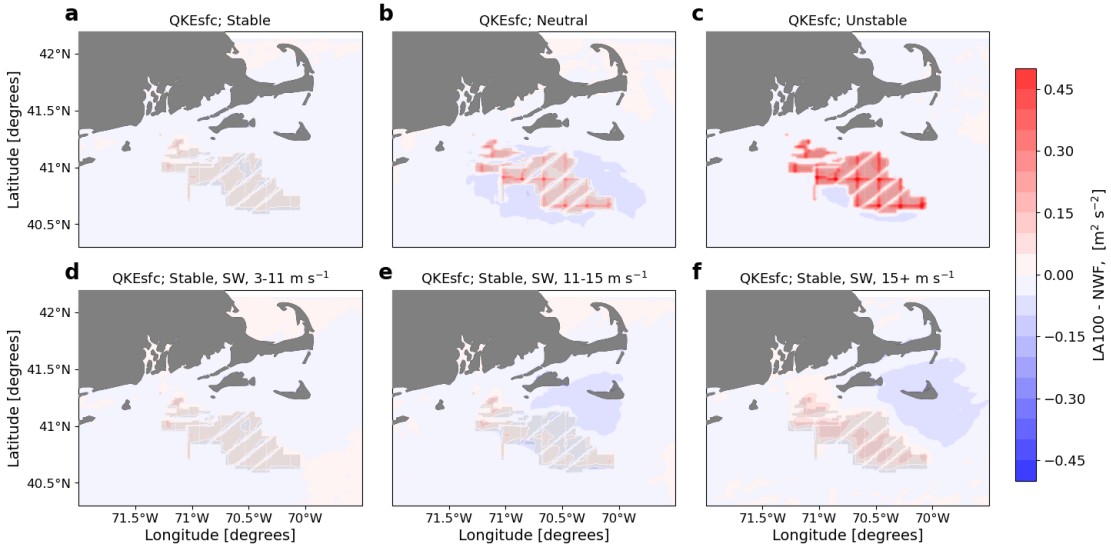

**Figure 15.** Mean surface QKE difference between the LA100 and NWF simulations in (a) stable conditions, (b) neutral conditions, (c) unstable conditions, (d) 3-11 m s$^{-1}$, (e) 11-15 m s$^{-1}$, and (f) 15+ m s$^{-1}$. Panels d, e, and f are only during times with stable conditions and southwest winds. Turbines are marked in gray.

## 4.3 Wind plant wake impacts on 2-m temperature

Differences in 2-m temperature are small, less than 0.3 K for the LA100 simulations (Fig. 17) and less than 0.4 K for the LA0 simulations (Fig. 18), when wind plants are present. However, the sign of these temperature differences changes with the amount of turbine-added TKE, emphasizing the importance of validation of this parameter with observations. Any differences are the most significant during stable conditions (Fig. 17a, Fig. 18a). During stable conditions in LA100, temperatures increase by around 0.1 K within the wind plant and decrease by a similar amount downwind of the wind plant. In LA0, however, temperatures decrease by about 0.1 K both within and downwind of the wind plant. In neutral and unstable conditions for both sets of simulations, temperature changes are negligible (Fig. 17b,c, 18b,c), likely because the boundary layer is already well

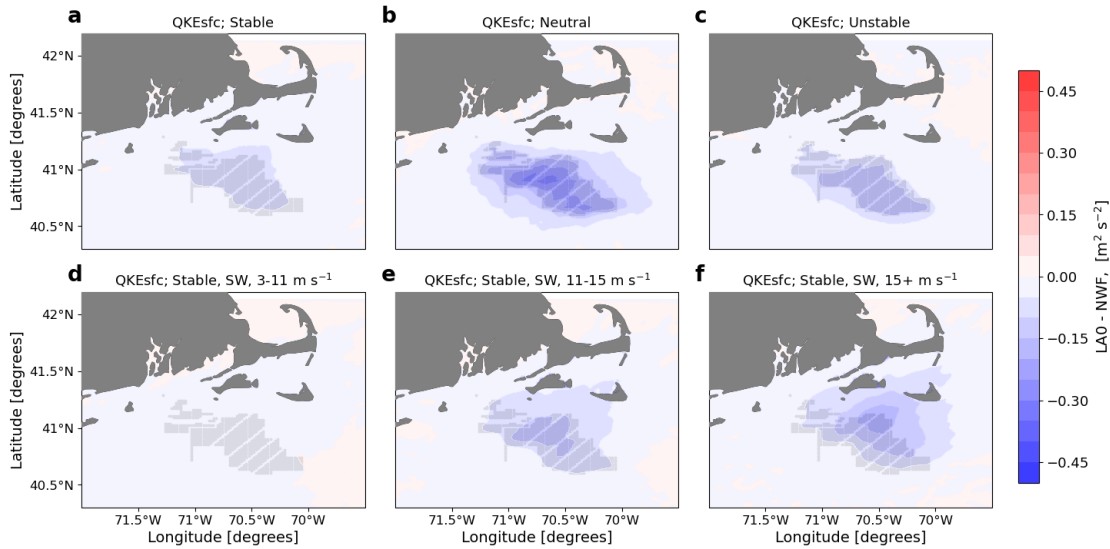

**Figure 16.** Mean surface QKE difference between the LA0 and NWF simulations in (a) stable conditions, (b) neutral conditions, (c) unstable conditions, (d) 3-11 m s$^{-1}$, (e) 11-15 m s$^{-1}$, and (f) 15+ m s$^{-1}$. Panels d, e, and f are only during times with stable conditions and southwest winds. Turbines are marked in gray.

mixed and so air with similar potential temperatures is mixed from the rotor regions down to the surface. Any mixing by wind turbines is simply remixing a well-mixed layer.

During stable conditions with turbine-added turbulence (Fig. 17a,d,e,f), turbines mix warmer air from aloft down to the surface, resulting in what appears to be a temperature increase but is really just redistribution of heat (as also discussed in Fitch et al. (2013) and Siedersleben et al. (2018a), among others). This mixing shows the strongest effect on temperature for faster wind speeds. In contrast, the stably-stratified simulations with no turbine-added turbulence (LA0, Fig. 18a,d,e,f), consistently show cooling at the surface.

Therefore, the amount of added TKE exerts an impact on observed temperature changes. Several other studies with added TKE have identified warming below the turbine hub, with the strongest impacts occurring during stable conditions (Baidya Roy and Traiteur, 2010; Lu and Porté-Agel, 2011; Fitch et al., 2013; Siedersleben et al., 2018a; Golbazi et al., 2022). The simulations used here use wind turbines with a hub-height of 138 m, which resembles the "extreme-scale" turbine of Golbazi et al. (2022). While they find cooling at the surface for bigger turbines (using 25% added TKE), we still find warming at the surface during stable conditions for LA100 simulations but cooling for LA0 simulations. In stably stratified mesoscale simulations onshore (Texas), Xia et al. (2019) find that the turbine-added turbulence drives the surface warming signal by enhancing vertical mixing. In contrast, distant from the wind farm, the turbine drag component causes the downwind surface cooling by reducing shear and promoting near-surface thermal stratification. A similar process occurs here: the LA0 simulations only incorporate the drag component and cooling, while the LA100 simulations promote surface temperature increases. Regardless of the interest in the




opposing sign of these changes between the LA100 and the LA0 simulations, it is important to recognize these changes are
subtle, on the order of 0.3-0.4 K.

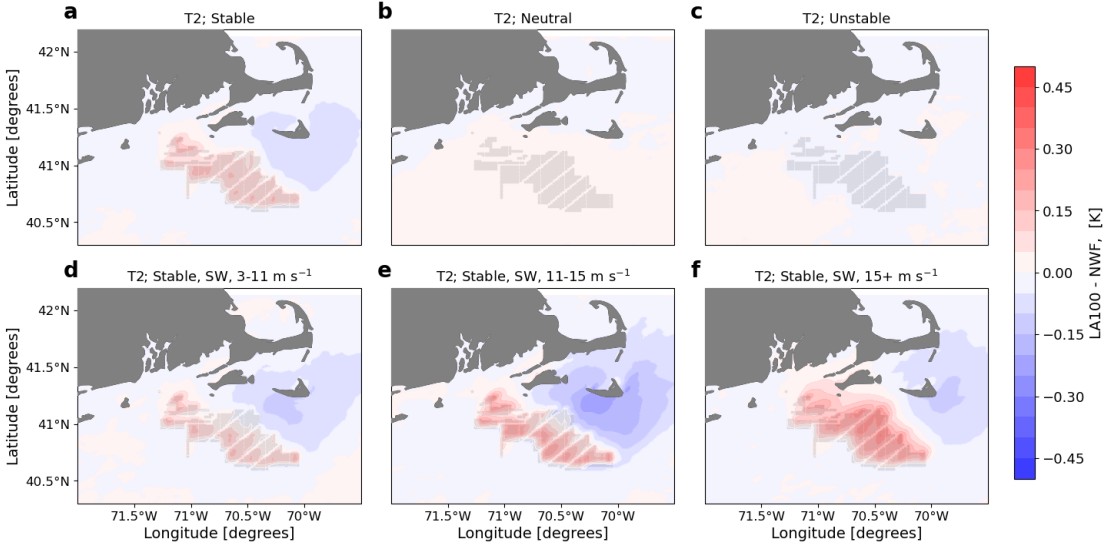

**Figure 17.** Mean 2 m temperature difference between the LA100 and NWF simulations in (a) stable conditions, (b) neutral conditions, (c) unstable conditions, (d) 3-11 m s$^{-1}$, (e) 11-15 m s$^{-1}$, and (f) 15+ m s$^{-1}$. Panels d, e, and f are only during times with stable conditions and southwest winds. Turbines are marked in gray.

## 4.4 Wind plant wake impacts on surface sensible heat flux

As with the 2-m temperature changes, heat flux changes induced by wind plants are small but vary with stability and with the amount of turbine-added turbulence (Figs. 19 and 20). The Appendix includes the average values for the heat fluxes for the NWF simulations (Fig. C1), the LA100 simulations (Fig. C2), and the LA0 simulations (Fig. C3) to emphasize the subtle
nature of the wind-plant induced changes. Heat fluxes are least modified by the wind plants in neutral conditions (Figs. 19b and 20b), with changes of less than 1 W m$^{-2}$ throughout the region. In unstable conditions (Figs. 19c and 20c), with the introduction of wind plants, wind plants slightly reduce the upward heat flux. The reduction is about 3 W m$^{-2}$ for LA100 and 5 W m$^{-2}$ for LA0.

In stable conditions, the effects show more spatial variability and are strongly linked to the amount of turbine-added tur-
bulence. The ambient (NWF) background heat flux is slightly negative, as the water is typically colder than the air (Fig. C1). Within the wind farm, for the LA100 simulations, the magnitude of the heat flux increases (Fig. 19 d,e,f). This stronger downward heat flux leads to warming inside the wind farm (Fig. 17 d,e,f). In contrast, in the LA0 stable cases (Fig. 20a,d,e,f), where turbine-added turbulence is absent, there is no warming inside the wind farms. The reduction in TKE results in cooling in the wake region (Fig. 18 d,e,f), consistent with the increase in heat flux magnitude downwind (Fig. 20 d,e,f). These results are

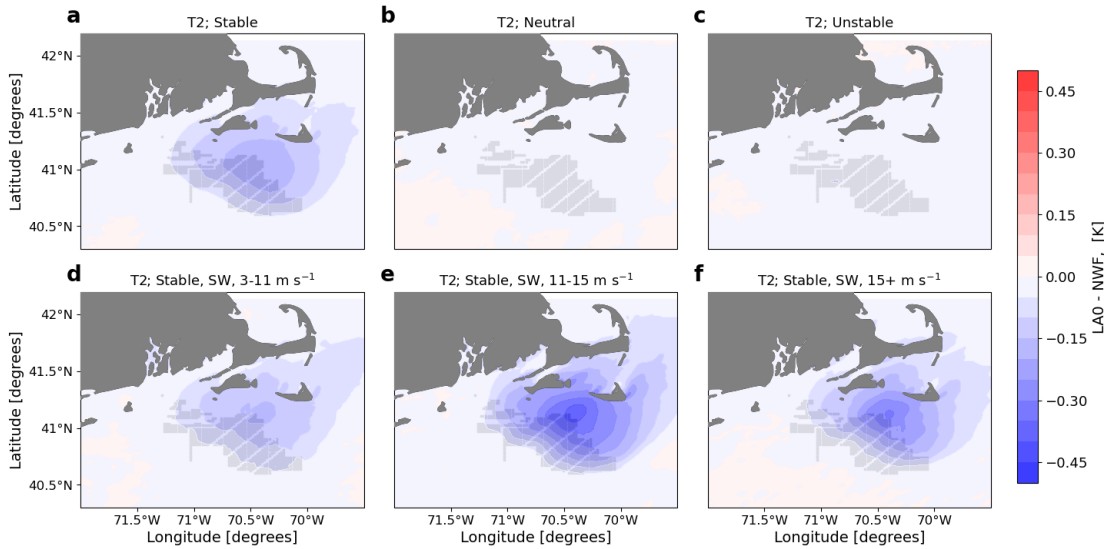

**Figure 18.** Mean 2 m temperature difference between the LA0 and NWF simulations in (a) stable conditions, (b) neutral conditions, (c) unstable conditions, (d) 3-11 m s$^{-1}$, (e) 11-15 m s$^{-1}$, and (f) 15+ m s$^{-1}$. Panels d, e, and f are only during times with stable conditions and southwest winds. Turbines are marked in gray.

in agreement with Golbazi et al. (2022), demonstrating how changes in surface TKE impact heat flux magnitudes, ultimately influencing surface temperature patterns both within and downstream of the wind farms.

**4.5 Wind plant wake impacts on planetary boundary layer height**

The simulated effects of wind plants on the PBL height change with atmospheric stability and the amount of wind-turbine-added turbulence. When turbulence is added to the simulations (Fig. 21), in the vicinity of wind plants, the PBL height increases

for all stability classifications. For the LA100 simulations, atmospheric stability modulates the strength of this effect: the PBL height increases by up to 99 m during stable conditions from a mean value of 268 m, and up to 67 m and 39 m for neutral and unstable conditions, respectively, from means of 556 and 644 m, respectively. During stable conditions, NWF PBL heights are generally lower and often within the rotor region, which results in a larger overall change in PBL height. During neutral and unstable conditions, PBL heights are generally deeper; thus, turbines are less likely to interact with air in the free atmosphere.

The largest increases in PBLH, in the LA100 simulations, occur during stably stratified conditions with winds between 11- 15 m s$^{-1}$. Additionally, downwind of the wind plant, a rebound effect occurs, with decreases in boundary layer height in slow (3-11 m s$^{-1}$) and moderate (11-15 m s$^{-1}$) wind speeds. Distant from the wind plant, PBL heights are reduced by up to 45 m during stable conditions as compared to the NWF simulation, likely due to the decreased shear in the wake of the wind plant. During neutral and unstable conditions, PBL heights increase throughout the region during neutral conditions, while

the impact of wind plants is limited to the lease areas during unstable conditions. This variability concurs with the large-eddy simulations of Wu and Porté-Agel (2017), which found increased PBL heights in the wind plant and depressed PBL heights in

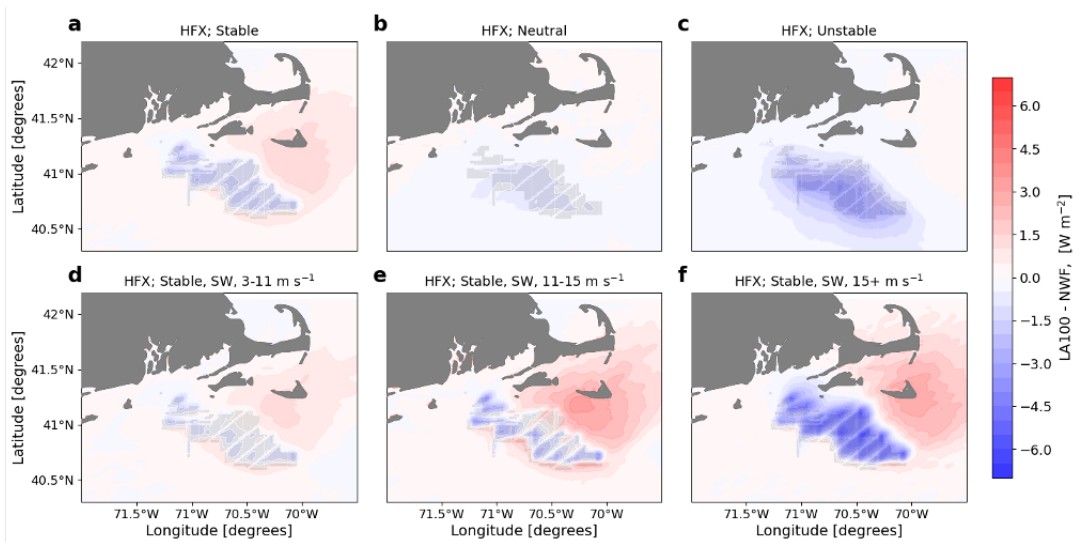

**Figure 19.** Mean heat flux difference between the LA100 and NWF simulations in (a) stable conditions, (b) neutral conditions, (c) unstable conditions, (d) 3-11 m s$^{-1}$, (e) 11-15 m s$^{-1}$, and (f) 15+ m s$^{-1}$. Panels d, e, and f are only during times with stable conditions and southwest winds. Turbines are marked in gray.

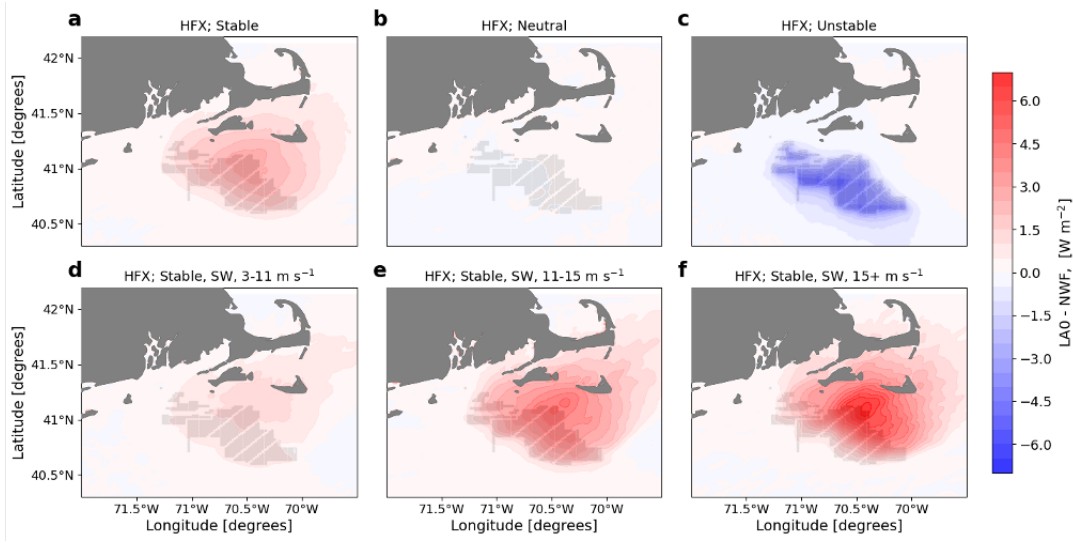

**Figure 20.** Mean heat flux difference between the LA0 and NWF simulations in (a) stable conditions, (b) neutral conditions, (c) unstable conditions, (d) 3-11 m s$^{-1}$, (e) 11-15 m s$^{-1}$, and (f) 15+ m s$^{-1}$. Panels d, e, and f are only during times with stable conditions and southwest winds. Turbines are marked in gray.

the exit region of the wind plant. Fitch et al. (2013) also find, in mesoscale simulations, that wind plants elevate the height of

the PBL height by up to 145 m. PBLH values were also slightly depressed downwind of the wind plant during stably stratified conditions.

In contrast, the stable conditions of the LA0 simulations show subtle decreases in boundary-layer height (Fig. 22a,d,e,f) because these simulations lack turbine-added turbulence that would nudge the boundary layer up. In these cases without added turbulence, the turbine-induced drag prevents the increase of PBL height. The slower winds in the wake may also reduce the development of the boundary-layer height, thereby reducing it. Only in the neutrally-stratified cases are there possible increases in boundary layer height, albeit constrained to 20 m or so. Observations (Abraham et al., 2024) and large-eddy simulations (Wu and Porté-Agel, 2017; Gadde and Stevens, 2021) indicate that the boundary-layer height should increase in the wake, so the LA0 simulations fail to represent a critical physical process.

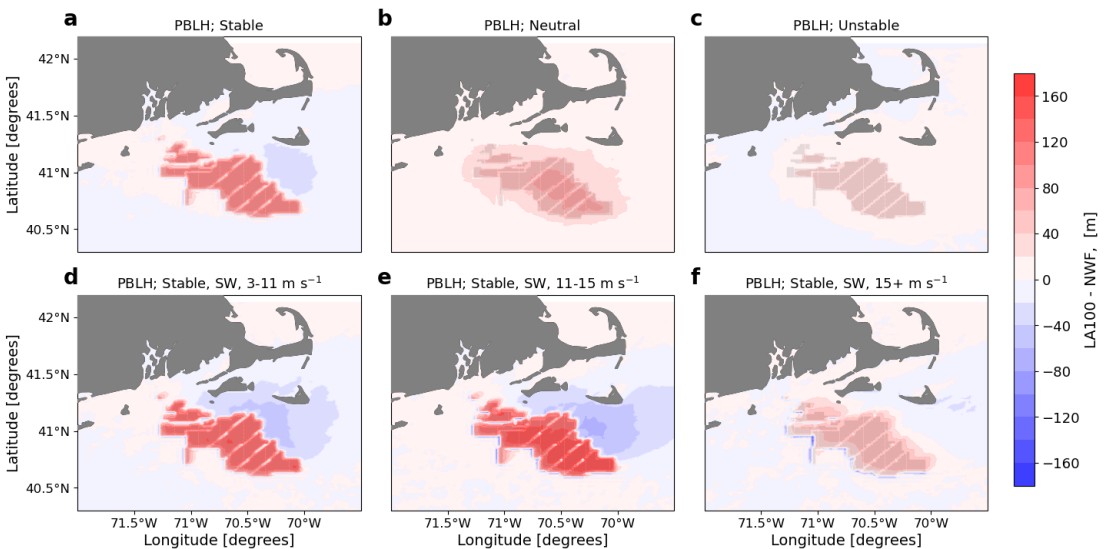

**Figure 21.** Mean PBLH difference between the LA100 and NWF simulations in (a) stable conditions, (b) neutral conditions, (c) unstable conditions, (d) 3-11 m s$^{-1}$, (e) 11-15 m s$^{-1}$, and (f) 15+ m s$^{-1}$. Panels d, e, and f are only during times with stable conditions and southwest winds. Turbines are marked in gray.

## 4.6   Relationship between PBLH and wake extent for LA100 simulations

The wind plant wake extent in stable conditions depends on the upwind PBLH in the Massachusetts-Rhode Island lease area in a statistically significant way (Fig. 23). Wake area and length are determined using the methodology described in Section 3. Wake area and length generally decrease with increasing PBLH for PBLH values between 100 m and 600 m. As the upwind PBLH increases, the volume of air available to recover the wake increases, so wakes in deeper PBLs tend to have a smaller area and length. When the PBLH is shallow, lower than 100 m, the wind plant wake area increases with increasing PBLH (not shown). These lower PBLH values are below hub height and are associated with smaller wakes because the wake mostly propagates in the free atmosphere, where a large volume of air is available to quickly recover the wake. In Fig. 23, a linear relationship

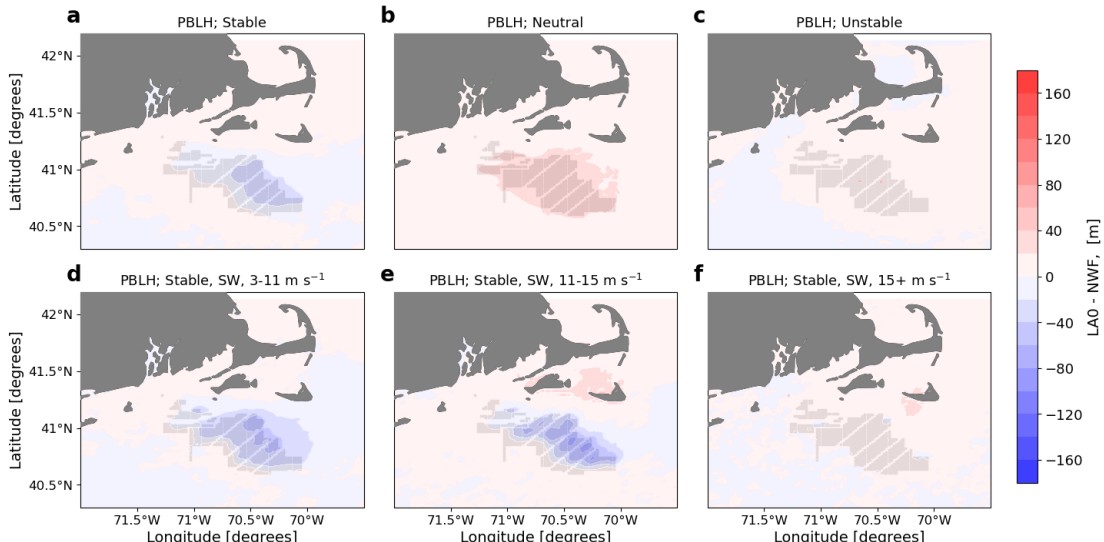

**Figure 22.** Mean PBLH difference between the LA0 and NWF simulations in (a) stable conditions, (b) neutral conditions, (c) unstable conditions, (d) 3-11 m s$^{-1}$, (e) 11-15 m s$^{-1}$, and (f) 15+ m s$^{-1}$. Panels d, e, and f are only during times with stable conditions and southwest winds. Turbines are marked in gray.

appears between mean lease area PBLH and the mean wake area or length for each bin. For every 1 meter increase in PBLH, the wake area decreases by approximately 10.4 km$^2$, and the wake distance decreases by about 0.1 km. The significance of these results is supported by the p-values, which are significantly below the 0.05 threshold. Still, the standard deviation is quite large, indicating considerable variability, likely associated to the fact that other factors such as ambient TKE and wind speed also influence the wake extent in conjunction with the PBLH.

The analysis of stable wake areas and lengths in the LA100 simulations demonstrates that deeper PBLs support faster wake recovery in a statistically significant way. The PBLH is inversely proportional to wake area and length, but other variables likely contribute to influencing the variability of wake extent. In a 55-day set of mesoscale simulations, Pryor et al. (2021) also suggest a negative relationship between PBLH and a normalized wake extent parameter. The large-eddy simulations of Maas and Raasch (2022) also find that wakes, at least for very large wind plants, tend to be longer with shallower boundary layers.

## 5 Conclusions

Here, we assess the simulated meteorological impact of offshore wind plants over an annual cycle using one year of WRF simulations with and without wind plants incorporated into the model, testing the limits of different amounts of turbine-added turbulence. We assess the difference in hub-height wind speed, 10 m wind speed, hub-height turbulence, surface turbulence, 2 m temperature, surface heat flux, and boundary-layer height during different stability classifications and ambient wind speeds in the Massachusetts-Rhode Island lease area. We also develop and demonstrate a machine-learning approach to identify wind

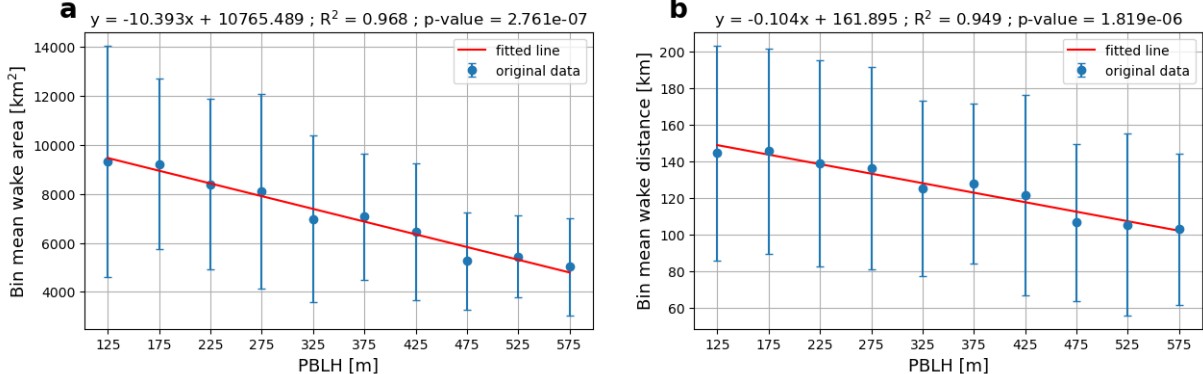

**Figure 23.** The mean (a) LA100 wake area and (b) LA100 length as a function of the mean lease area PBLH during stable conditions, where the PBLH values are binned every 50 m from 125 m to 575 m. The mean PBLH values are calculated as the average PBLH across the wind lease area. For each bin, the blue dots show the mean value, and the blue vertical lines represent the $\pm 1$ standard deviation error bar. The best-fit line for the bin means is in red. Each panel displays the line of best fit equation, $R^2$ value, and p-value on the top.

plant wakes, and use this method to demonstrate the relationship between boundary-layer height and both the area and length of the wind plant wake. These simulations, without wind turbines, were validated in comparison to offshore profiling lidar observations in Rosencrans et al. (2024). We distinguish the micrometeorological impacts of wind turbines by atmospheric stability. Further, for stably stratified cases, when wake effects are expected to be the most significant, we distinguish three ranges of wind speed related to the operation of the turbine: 3-11 m s$^{-1}$ (Region 2 of the turbine), 11-15 m s$^{-1}$ (the early part of Region 3 of the turbine, when $C_T$ can be large), and 15+ m s$^{-1}$ (Region 3 when $C_T$ is small).

Only the hub-height wind speed is affected by the presence of wind turbines in similar ways regardless of wind turbine operating parameters: the introduction of wind turbines into the simulations reduces hub-height wind speeds most in stable conditions and least in unstable conditions, regardless of the amount of added turbulence. In contrast, wind speeds near the surface decrease when no turbine-added turbulence is included, but can increase for stably stratified conditions when 100 % of possible TKE is included in the simulations. In the vicinity of the wind plants for the simulations with added TKE, TKE increases at hub height in the simulations with added TKE for all stability classes, suggesting that atmospheric stability does not immediately modify the TKE generated by turbines. Negligible changes in hub-height TKE manifest in the simulations without the added TKE. At the surface, TKE increases in the LA100 simulations only for unstable simulations when boundary-layer wide circulations can transport turbine-generated TKE to the surface. In the no-added-turbulence simulations, surface TKE decreases slightly in neutral and unstable simulations, again due to vertical transport of flow with reduced shear-generated turbulence and no turbine-added turbulence.

Thermodynamic quantities also show a dependence on the model parameterization of turbine-added turbulence and atmospheric stability, with changing signs of temperature and heat flux changes, although these changes are of small magnitude, less than 0.4 K. With added turbulence (LA100 simulations), a subtle (less than 0.3 K) increase in 2-m temperature within the

wind plant can occur; this increase is strongest during stable conditions and faster wind speeds. Also during stable conditions, temperature decreases slightly downwind of the wind plant, as has been observed in mesoscale simulations in other regions. In contrast, the simulations with no added turbulence show cooling within and downwind of the wind plant. Changes in heat fluxes are also small and coupled to the changes in 2 m temperature. In simulations with added turbulence, this turbulence nudges the boundary layer up in the immediate vicinity of the wind turbines, especially during stable conditions and when winds are in Region 2 of the power curve, with a rebounding effect downwind of slight reductions in boundary layer height. In contrast, with no added turbulence, the boundary layer height is in general reduced in stable conditions with wind speeds less than 15 m s$^{-1}$ and slightly increased in neutral conditions.

Using a machine-learning approach for quantifying both the area covered by a wind turbine wake and the along-wind-direction length of the wind turbine wake, we find a statistically-significant relationship between the upwind boundary-layer height and the extent of the wake. Shallower boundary layer heights promote larger wakes.

These simulations use one type of turbine and the planned 1 nautical mile spacing for these wind plants. Further investigations could assess how these impacts change with varying turbine layouts and sizes, as in Golbazi et al. (2022); Pryor et al. (2021). While Golbazi et al. (2022) considers larger turbines similar to turbines used here, more research into their impacts on local meteorology is needed.

Our analysis is based on one complete year of simulations, rather than one season (Golbazi et al., 2022) or selected multiday studies (Pryor et al., 2021). However, interannual variability does affect wind resources (Lee and Lundquist, 2017; Bodini et al., 2016) and so may affect the effects of wakes as well. Therefore, a multiyear study could provide more insights into interannual variability and how it might affect wakes.

Further, simulated winds (Draxl et al., 2014; Bodini et al., 2024a; Liu et al., 2024) and simulated wakes (Rybchuk et al., 2022) show dependence on the PBL scheme chosen for the model simulations. At the moment, the Fitch wind farm parameterization is coupled only with the MYNN PBL scheme used here and with the 3DPBL scheme (Kosović et al., 2020; Juliano et al., 2021). Future work could assess how micrometeorological responses to wind plant wakes depend on the choice of PBL scheme.

Of course, this study relies on the accurate representation of wakes in the Fitch WRF wind farm parameterization. While wakes simulated with this parameterization compare reasonably well with the limited sets of observations available (Lee and Lundquist, 2017; Siedersleben et al., 2018b, a, 2020; Ali et al., 2023; Larsén and Fischereit, 2021), the availability of observations of wake effects at multiple distances and heights from wind plants, especially offshore, is limited. Ongoing experiments such as AWAKEN (Moriarty et al., 2024) may provide more extensive datasets to support modifications to wind farm parameterizations in mesoscale models. Additionally, comparisons of these mesoscale representations to more finely resolved large-eddy simulations of wind plants (Vanderwende et al., 2016; Peña et al., 2022; García-Santiago et al., 2024) may suggest other improvements, although these comparisons should be carried out for a range of atmospheric stability conditions and wind plant geometries. Particular attention should be paid to effects on surface meteorology as well as dynamics directly relevant to wind turbine power production.

Further, because these present simulations are not coupled with a wave model and ocean model, other feedbacks between the ocean and atmosphere may be relevant. Over water, wind plant wakes may influence ocean dynamics (Raghukumar et al.,

2022, 2023; Liu et al., 2023), including upwelling. Therefore, coupling with wave and ocean models could provide insight into potential wake impacts on the ocean. Daewel et al. (2022) considers the impact of offshore wakes on primary production, but additional analysis on surface currents would provide a more complete picture of wake impacts. The ocean's response may also mediate these effects of wakes on surface meteorology, as suggested by the simulations of Fischereit et al. (2022b) in the North Sea. Extended simulations, such as those shown here, with a coupled atmosphere-ocean-wave model, could provide more accurate insight into the ocean's role in modulating wake impacts. Such work is ongoing.

Finally, while we focused on one lease area in the northeastern U.S. wind regions, it is important to conduct further analysis on other lease areas in the region and worldwide. Variability in wind patterns, boundary-layer heights, and sea surface temperatures are likely to interact with wind plant wakes in ways that can modify the results presented here.

*Code and data availability.* The NOW-WAKES simulation data is available at https://dx.doi.org/10.25984/1821404 and code is available at https://zenodo.org/doi/10.5281/zenodo.10993297.

## Appendix A: Additional variable climatologies

### A1    Hub-height wind speed climatology

Hub-height wind speeds are fastest during the winter and spring, with means of 10.5 and 11.3 m s$^{-1}$, respectively (Fig. A1).
In the summer and fall, hub-height wind speeds are slightly slower, with means of 7.6 and 9.5 m s$^{-1}$, respectively. During all
seasons, hub-height wind speeds generally increase towards the east.

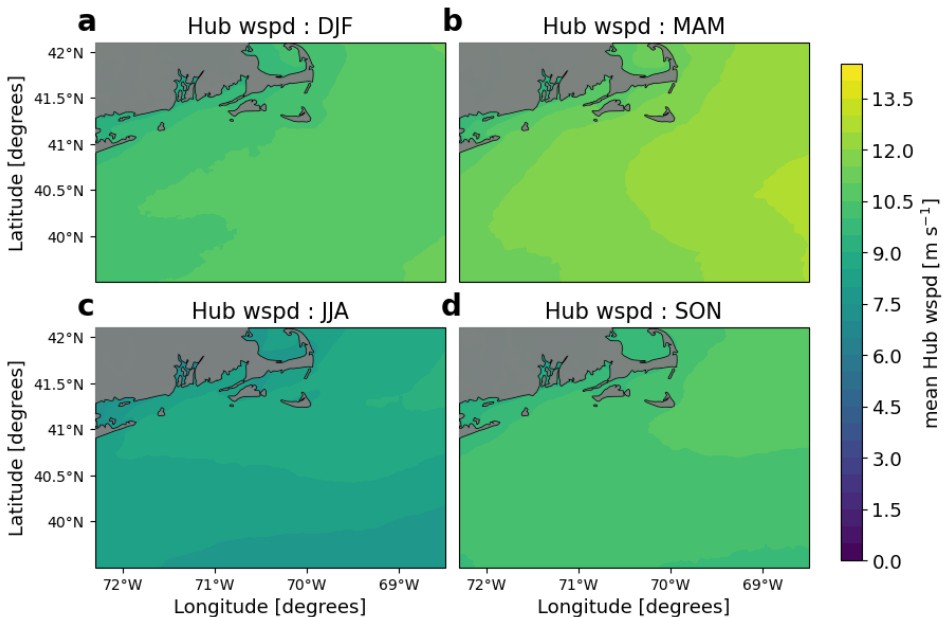

**Figure A1.** Same as in Fig. 5, but for hub-height wind speed

### A2    10 m wind speed climatology

Ten meter wind speeds are fastest during the fall, winter, and spring, with means between 7 and 8 m s$^{-1}$ (Fig. A1). Wind
speeds generally increase with distance from the coast during these seasons. In the summer, 10 m wind speeds are slower, with
a mean of 5.2 m s$^{-1}$. A minimum in mean 10 m wind speeds exists to the east of Nantucket in the summer.

### A3    Surface TKE climatology

Surface TKE is highest in the fall and winter and smallest in the summer, with mean values of 0.75, 0.8, and 0.3 m$^2$ s$^{-2}$ (Fig.
A3). In all seasons, surface TKE is larger to the south, away from the coast.

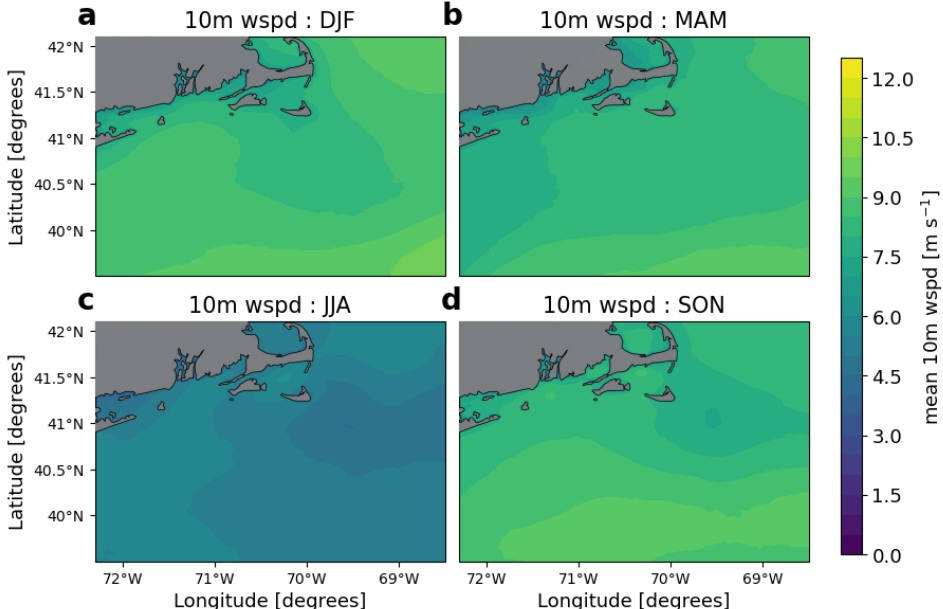

**Figure A2.** Same as in Fig. 5, but for 10 m wind speed

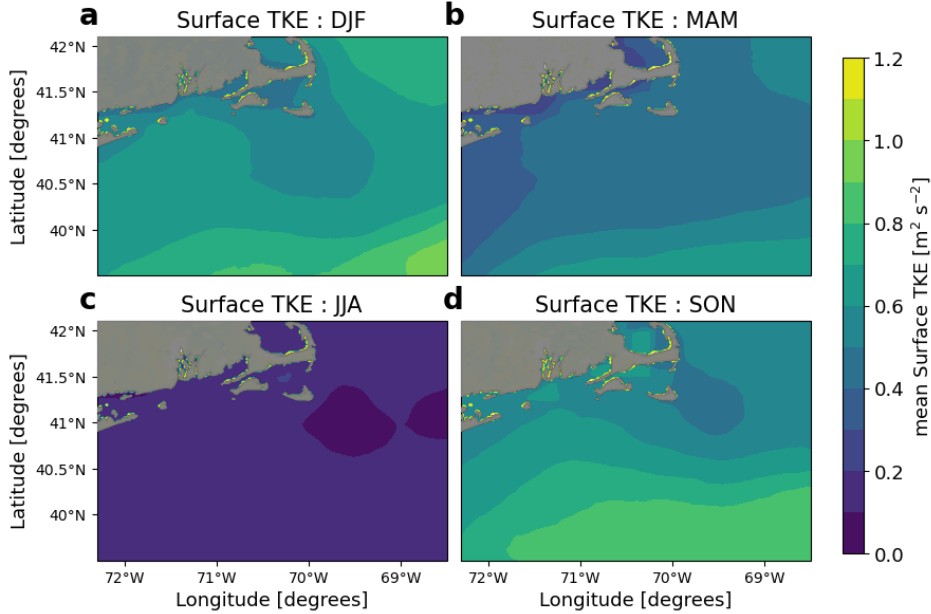

**Figure A3.** Same as in Fig. 5, but for surface TKE.

## A4  2 m temperature climatology

Mean 2 m temperature (T2) is warmest in the summer with a mean of 300 K, and coldest in the winter with a mean of 298 K. T2 is generally warmer to the south of the considered area for all seasons. In the summer, a T2 minimum occurs east of Nantucket. In the winter, minimum temperature occurs close to land. (Fig. A4).

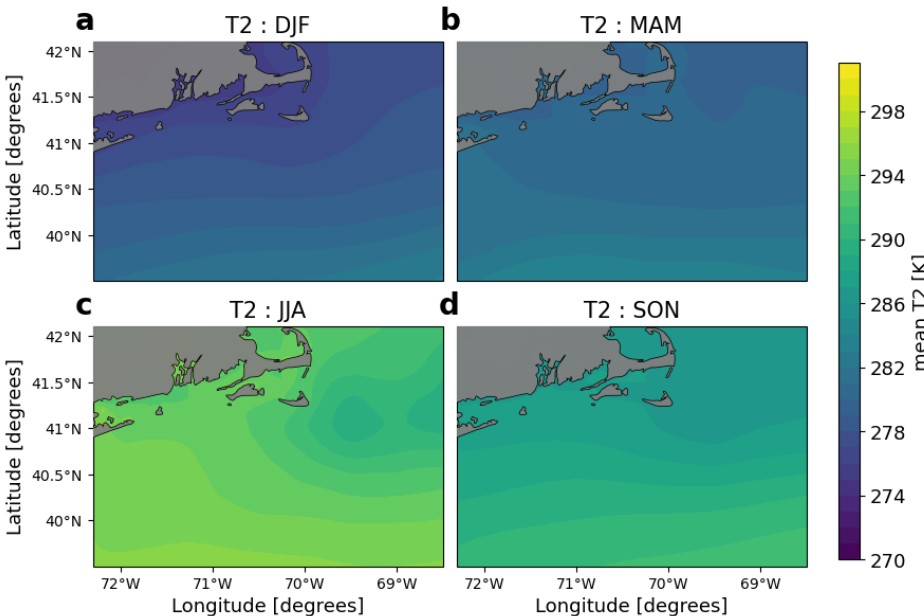

**Figure A4.** Same as in Fig. 5, but for 2 m temperature.

## Appendix B:  Meteorological impacts of wind plant wakes as a function of wind direction

Meteorological impacts of offshore wind plants also vary with wind direction. These results are likely driven by effects of atmospheric stability, as each stability class is associated with a dominant range of wind directions (Fig. 6). In particular, southwesterly winds almost always occur during stable conditions.

### B1  Wind plant wake impacts on hub-height wind speed

All wind directions observe a wind speed deficit both within and downwind of the wind plant (Fig. B1). Southwesterly winds have the strongest wind speed deficit within the wind plant and a larger wake than the other wind directions. In fact, southwesterly winds are typically associated with stable conditions, which are associated with larger wakes.

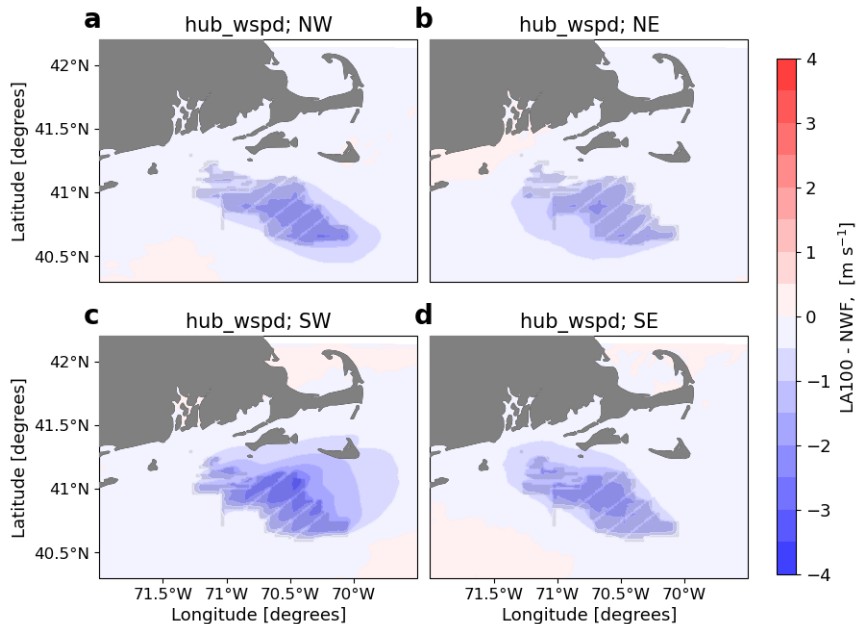

**Figure B1.** Mean hub-height wind speed difference between the LA100 and NWF simulations for (a) NE, (b) NW, (c) SE, and (d) SW wind directions. Turbines are marked in gray.

## B2    Wind plant wake impacts on boundary layer height

In the vicinity of wind plants, the PBLH increases for all wind directions (Fig. B2). This effect is the strongest when there is southwesterly flow (i.e., stable conditions) and weaker for the other three quadrants. Southwesterly and northeasterly wind both exhibit a downwind reduction in PBL, while this effect is minimal for southeasterly winds and nonexistent for northwesterly winds. The downwind reduction in PBLH is strongest for southwesterly winds.

## B3    Wind plant wake impacts on 2-m temperature

The largest temperature differences occur during southwesterly flow (Fig. B3). Temperatures increase by around 0.1 K in the lease area, and decrease by a similar amount downwind of the wind plant. Temperature differences are much smaller for other wind directions. In general, southerly winds are associated with an increase in temperature within the wind plant and a reduction in temperature outside of the wind plant. Temperature differences are very small for northerly winds. When winds are from the northeast, a slight warming is observed downwind of the wind plant.

## B4    Wind plant wake impacts on surface sensible heat flux

Heat fluxes are reduced within the wind plant for all wind directions (Fig. B4). The strongest reduction in heat flux occurs for northwesterly winds, with reductions of up to 3 W m $^{-2}$. The other 3 quadrants see differences of around 2 W m $^{-2}$

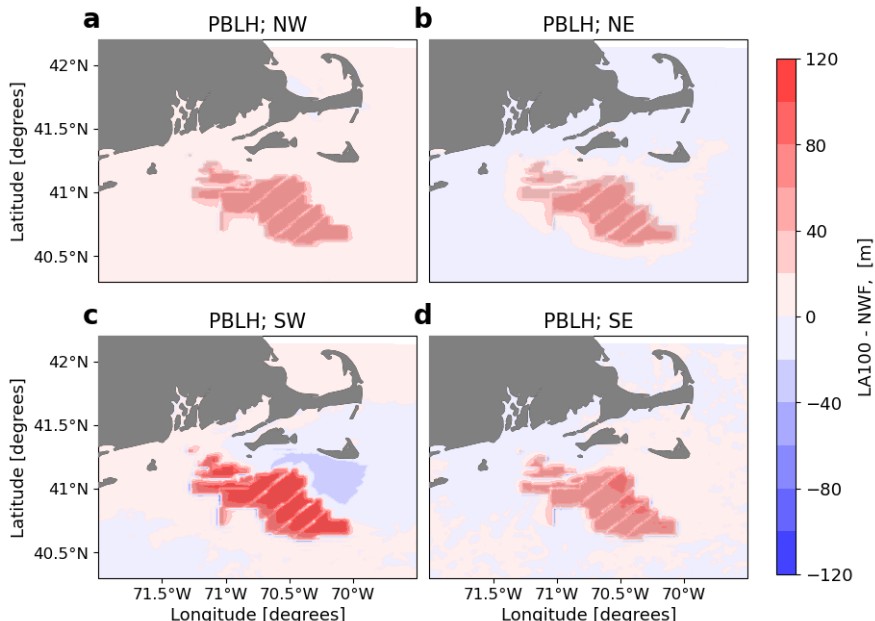

**Figure B2.** Mean PBLH difference between the LA100 and NWF simulations for (a) NE, (b) NW, (c) SE, and (d) SW wind directions. Turbines are marked in gray.

within the wind plant. When winds are northerly, heat flux is also reduced downwind of the wind plant, with the magnitude decreasing with distance from the wind plant. In contrast, heat flux increases downwind of the wind plant when winds are from the southwest. In addition, heat flux is slightly elevated in the immediate upwind direction.

### B5   Wind plant wake impacts on hub-height TKE

Hub-height TKE increases similarly for all wind directions (Fig. B5).

### B6   Wind plant wake impacts on surface TKE

Surface TKE increases within the wind lease area for all wind directions (Fig. B6). Surface TKE increases more when winds are from the northwest or the northeast, with values over 0.3 J kg$^{-1}$. Southerly winds are associated with an increase in TKE closer to 0.2 J kg$^{-1}$. We also notice a downwind reduction in surface TKE for all wind directions, although the magnitude of the reduction is larger for northerly winds.

### Appendix C:  Climatological mean values of heat flux

The interpretation of changes in heat flux (Fig. ), particularly in stable conditions, is facilitated by examination of the mean heat flux in each of the simulations. In the NWF simulation (Fig. C1, the mean heat flux in each of the stability conditions matches

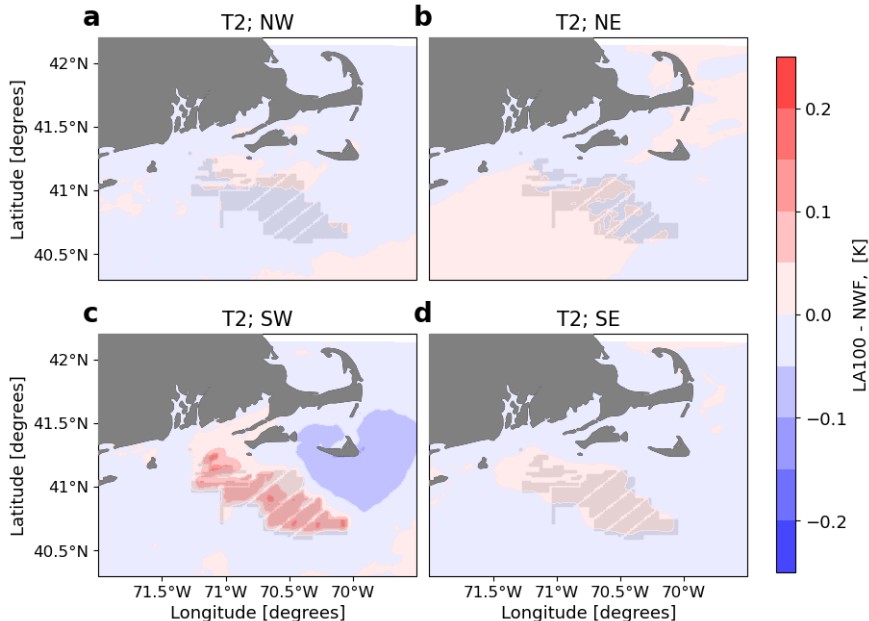

**Figure B3.** Mean 2-m temperature difference between the LA100 and NWF simulations for (a) NE, (b) NW, (c) SE, and (d) SW wind directions. Turbines are marked in gray.

expectations (with negative values in stable conditions, values close to zero in neutral conditions, and strongly positive values in unstable conditions) but also demonstrates spatial heterogeneity. The differences in the heat flux in the LA100 (Fig. C2) and LA0 (Fig. C3) from the NWF simulation (Fig. C1) are small compared to the ambient background values.

*Author contributions.* JKL and NB conceptualized the project and acquired funding and resources for the project. DR completed the WRF simulations and provided the dataset. DQ carried out the formal analysis and investigation, including developing software and carrying out the visualization, with supervision from JKL and NB. DQ prepared the initial draft. All authors reviewed and edited the publication.

*Competing interests.* At least one of the (co-)authors is a member of the editorial board of Wind Energy Science.

*Acknowledgements.* This research was performed using computational resources sponsored by the U.S. Department of Energy's Office of Energy Efficiency and Renewable Energy and located at the National Renewable Energy Laboratory. This work was supported in part by the U.S. Department of Energy, Office of Science, Office of Workforce Development for Teachers and Scientists under the Science Undergraduate Laboratory Internship program.

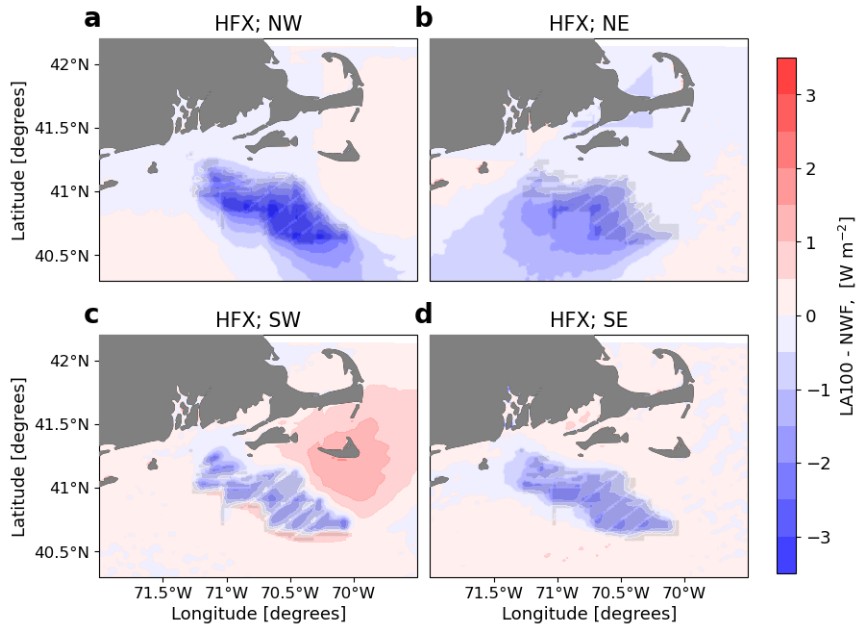

**Figure B4.** Mean heat flux difference between the LA100 and NWF simulations for (a) NE, (b) NW, (c) SE, and (d) SW wind directions. Turbines are marked in gray.

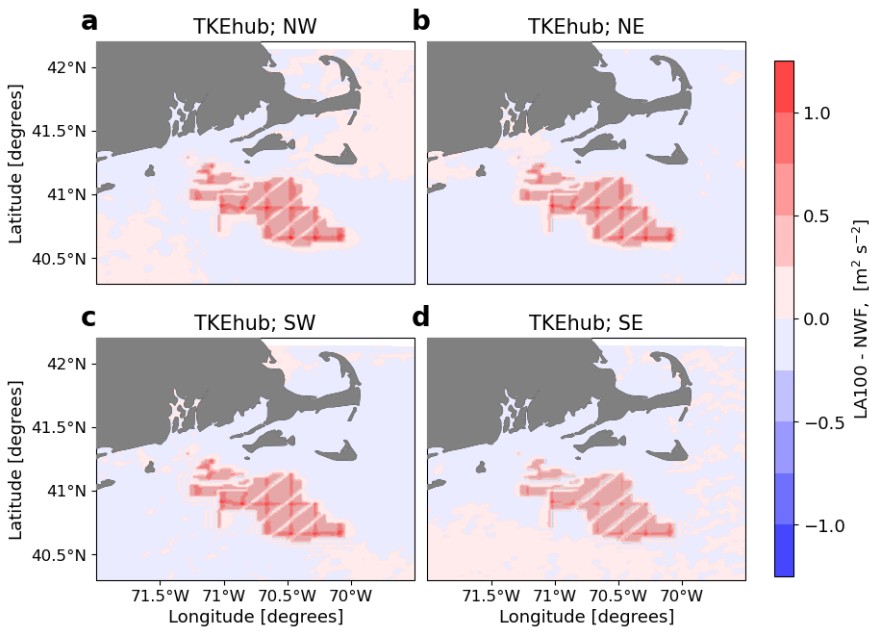

**Figure B5.** Mean hub-height TKE difference between the LA100 and NWF simulations for (a) NE, (b) NW, (c) SE, and (d) SW wind directions. Turbines are marked in gray.

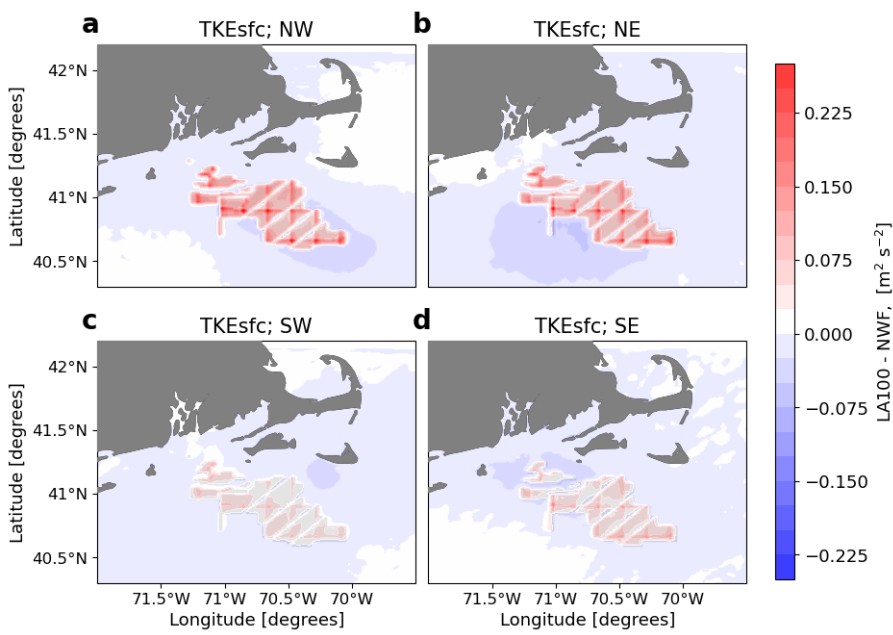

**Figure B6.** Mean surface TKE difference between the LA100 and NWF simulations for (a) NE, (b) NW, (c) SE, and (d) SW wind directions. Turbines are marked in gray.

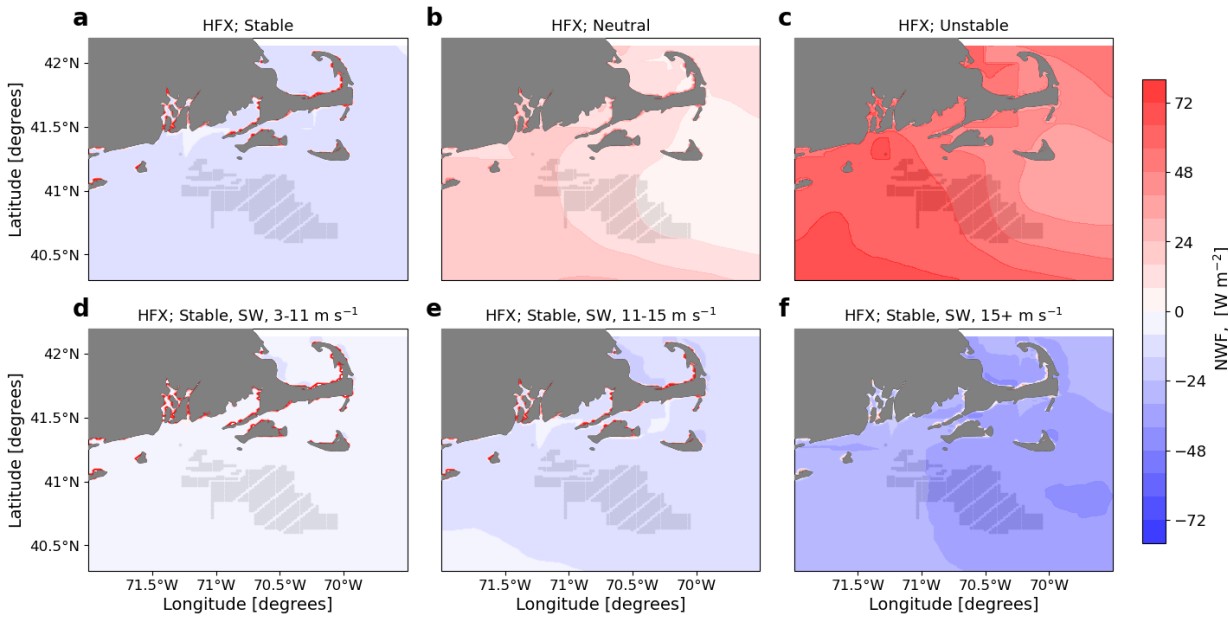

**Figure C1.** Mean surface heat flux for the NWF simulation in (a) stable conditions, (b) neutral conditions, (c) unstable conditions, (d) 3-11 m s$^{-1}$, (e) 11-15 m s$^{-1}$, and (f) 15+ m s$^{-1}$. Panels d, e, and f are only during times with stable conditions and southwest winds. Turbines are marked in gray..

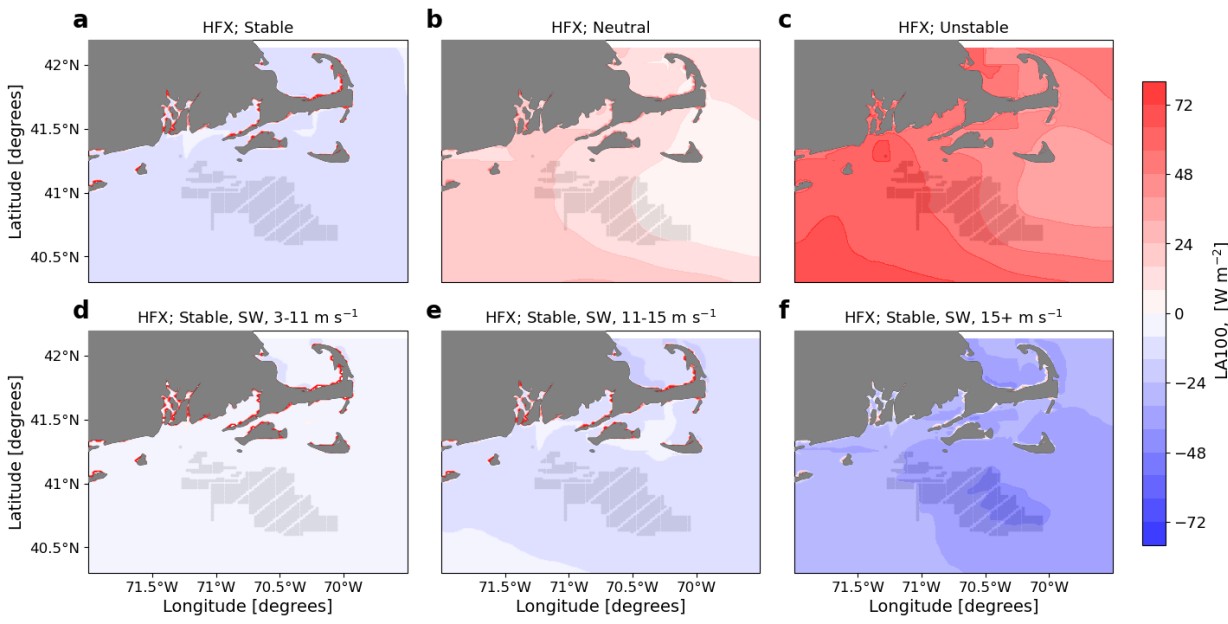

**Figure C2.** Mean surface heat flux for the LA100 simulation in (a) stable conditions, (b) neutral conditions, (c) unstable conditions, (d) 3-11 m s$^{-1}$, (e) 11-15 m s$^{-1}$, and (f) 15+ m s$^{-1}$. Panels d, e, and f are only during times with stable conditions and southwest winds. Turbines are marked in gray..

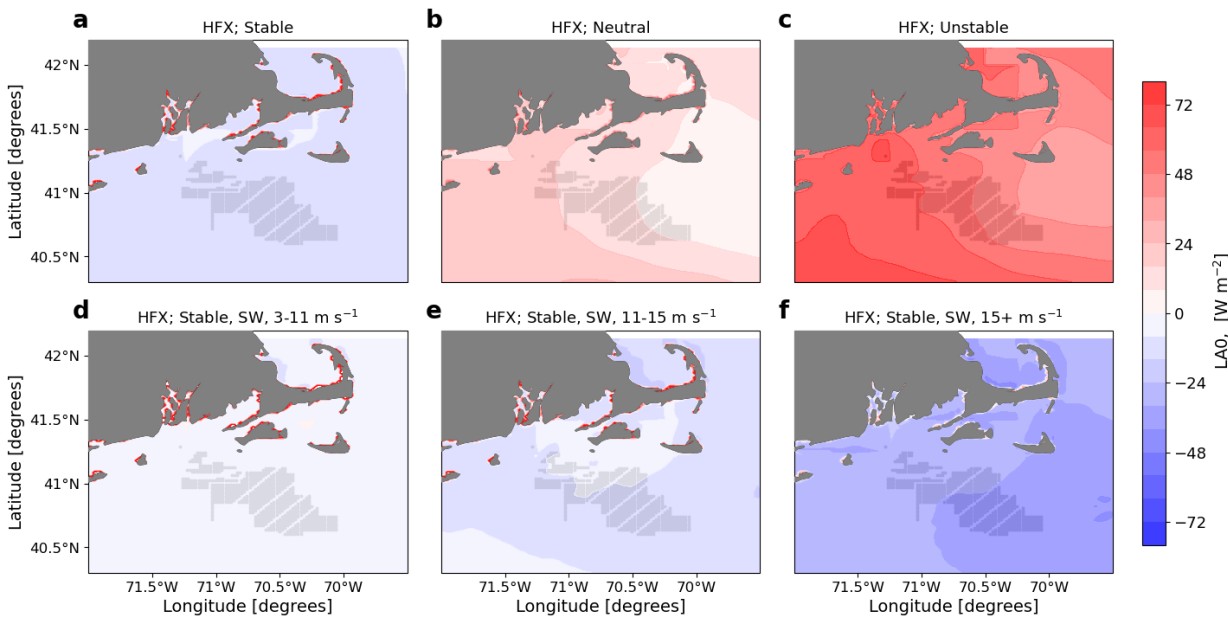

**Figure C3.** Mean surface heat flux for the LA0 simulation in (a) stable conditions, (b) neutral conditions, (c) unstable conditions, (d) 3-11 m s$^{-1}$, (e) 11-15 m s$^{-1}$, and (f) 15+ m s$^{-1}$. Panels d, e, and f are only during times with stable conditions and southwest winds. Turbines are marked in gray..

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
