# Peer review of "Simulated Meteorological Impacts of Offshore Wind Turbines and Sensitivity to the Amount of Added Turbulence Kinetic Energy"

_Wind Energy Science, 2024_

## Community Comment (CC1)

**Comment to Preprint wes-2024-53**

Andrea N. Hahmann, DTU Wind and Energy Systems, Denmark

June 28, 2024

Dear authors,

I have two comments regarding your manuscript:

**1 Possible feedbacks from the ocean surface**

In my opinion, the manuscript overlooks a crucial conceptual point that could significantly impact the manuscript results and conclusions. In the WRF model, the state of the land surface is controlled by a land surface model. Over the ocean, such a model is not often included. In your simulations, the `namelist.input` file shows that sea surface temperatures (SSTs) are specified from an input file. Using slowly varying but fixed SSTs throughout the simulation could lead to inaccuracies in calculating changes in heat fluxes and 2-meter temperatures and possibly other derived quantities. These inaccuracies result from omitting essential thermodynamic feedback processes from the ocean to the surface layer above. This factor must be included in your manuscript to ensure a fair discussion of your findings.

A possible mechanism **in nature** will be the following. In the stable case with strong winds, your results show a change in the heat from the air to the water of the order of $3\,\mathrm{W\,m^{-2}}$. This change in flux is positive downward, thus possibly increasing the skin temperature of the sea surface. This temperature change will decrease the vertical temperature gradient, which results in a reduced heat flux. The reduced heat flux could also alter the 2-meter temperature and the surface layer's stability. Thus, in nature, the ocean could respond to minimize the changes caused by wind farms. Or not. It could be argued, however, that the ocean has a large heat capacity and, thus, an excess of $3\,\mathrm{W\,m^{-2}}$ will quickly be mixed in the water column without altering the ocean's temperature. This will be linked to the stability of the ocean's

surface layer. Knowing which process will dominate is only possible with measurements and accurate simulations, including the thermodynamic effect on the ocean surface. Similar processes exist in the simulation of tropical cyclones, which often consider the possible ocean surface changes.

In your article, you cite the work of Golbazi et al. (2022), which carried out similar shorter WRF model simulations with various sizes of wind turbines. From their methods and namelist, I can also see that their simulations were done with fixed (but spatially more precise) SSTs. So, their results and discussion also disregard the possible effects of the surface ocean's response to changes in surface fluxes and temperatures. So, the article cannot be used to substantiate your results.

Through the results and discussion section of your manuscript, I often find that you mix land and offshore publications. The processes are different in these two environments in nature and the models and should not be mixed.

The issue of the possible impact of fixed SSTs should be discussed and addressed in your manuscript. This discussion is crucial for the reliability of your findings. It is outside the scope of your manuscript, but WRF model simulations, including the effects of a slab ocean, are possible. Running with a fully coupled ocean will be even better but expensive.

**2 Dependence of your results on PBL and wind farm parameterization**

While your results are exciting, they represent only one possible scenario with one WRF PBL scheme and one wind farm parameterization. Recent publications have shown that the impact of wind farms on the atmosphere is highly dependent on the PBL and wind farm parameterization used. The paper should emphasize this point and acknowledge that observations have yet to verify most aspects of the simulated impacts of large wind farms on the atmospheric flow.

Statements like the one in the abstract, "Offshore wind energy projects are currently in development off the east coast of the United States and will likely influence the local meteorology of the region." should be avoided. It is all a question of degrees and assessment of significance.

---

## Author Comment (AC1)

*Reviewer's comments appear in italics,* **our responses appear in boldface blue text**.

*The manuscript presents an additional analysis of a subset of the simulations conducted previously by the team and published in Rosencrans et al. (2024). The current analysis uses only the simulations that were conducted with 100% of the added TKE by the turbines available, which was the default value in the Fitch parameterization in older versions of the WRF. The default value in new versions of the WRF is 25%, not 100%. There is no justification in the paper as to why the value of 25% was not used.*

*By contrast, it is known that using 100% causes an overestimation of added TKE at the grid cells with the turbines. For example:*

1. *Eriksson et al. (2015) report excessive TKE by about a factor of 2 (their Figures 15 and 19);*
2. *Abkar and Porté-Agel (2015) find that TKE is overestimated by 50%-200% using the 100% factor (their Figure 5);*
3. *Pan and Archer (2018) find that using the 100% value causes an overestimation of at least a factor of two in several WRF simulations of commercial-scale offshore wind farms (their Fig. 6);*
4. *Archer et al. (2022) propose the 25% value because using 100% causes an overestimation by up to 300% in TKE (their Figure 6, case 4).*

*As the current manuscript uses the 100% value, which is unjustified and likely to cause incorrect distributions of heat and momentum fluxes, the manuscript as is should not be published. The team needs to rerun the simulations using the 25% value or, at the very least, rerun the simulations with the 25% value for selected months (say one per season) and present a sensitivity analysis of the results.*

**We thank the reviewer for their time and consideration in reviewing our manuscript. The issue of the correct amount of turbine-added TKE is an area of active research and scientific discussion, so we are surprised at the reviewer's stance that 25% is a settled issue, especially since the first three papers they cite predate the TKE bug fix which may have influenced the conclusions. The Archer et al. (2020) recommendation of 25% was based on large-eddy simulations of one turbine in neutrally-stratified conditions and is not likely to be representative of all circumstances in which wind farm parameterizations will be used.**

**Further, several papers (not from our research group) have recently concluded that the 25% value recommended by Archer et al. (2022) does not perform optimally:**

> **1.      Ali et al. (2023) state that "Overall, wind speed and TKE magnitudes predicted by Fitch's parameterization were close to the measured values. This result does not necessarily contradict other studies that found excessive turbulence generation using Fitch's parameterization, because the current comparison is local to the FINO-1 mast and the transect flights over the Gode Wind farms. However, setting $\alpha$ = 0.25 was not optimal in the conducted experiments."**

2.     Larsén and Fischereit (2021) state that "It also remains inconclusive which correction factor should be used in connection with the turbine-induced TKE generation in the Fitch scheme: we only tested two factors (1 and 0.25) here and we observe a better performance when using α = 1 than α = 0.25, which does not support the conclusion from Archer et al. (2020)."

Further, other groups that have implemented versions of the Fitch wind farm parameterization into other mesoscale numerical weather prediction simulation codes (Akhtar et al., 2024; Chatterjee et al., 2016; van Stratum et al., 2022), and have also chosen to use 100%. Our approach is not an outlier in the scientific community and is justified.

Finally, the reviewer requests an expensive sensitivity study. A short sensitivity for a range of wind speeds was already conducted and presented in Rosencrans et al. (2024), Supplemental Figure 1, which showed subtle impacts on the heat and moisture fluxes. We reproduce that figure here for the reviewer's convenience:

[Figure]

*Figure A1 The effects of modifying the amount of turbulent kinetic energy (TKE) during test runs. Panels show (a) hub-height wind speed, (b) surface moisture flux, (c) normalized power productionv (d) surface heat flux, (e) planetary boundary layer (PBL) height, and (f) 2 m temperature. Values are collected from a point centered on the RIMA block. Power production is the sum of all cells containing wind turbines. TKE_100 is shown in orange, TKE_50 in blue, TKE_25 in gray, TKE_0 in black, and NWF in purple dashes.*

**Given these very small differences, we find the considerable computational expense of such a sensitivity study to be unjustified.**

**Regardless, we should have included some discussion of the fact that this topic is an active area of research. Therefore, we include the following text in the manuscript starting at the end of Section 2, just before Section 2.1:**

> **The topic of how much turbine-generated turbulence to incorporate into mesoscale wind farm parameterizations is an active area of research. The initial default value recommended by Fitch et al. (2012) was 100%. In later work, Archer et al. (2020) recommended a value of 25% based on their large-eddy simulations of an individual turbine in neutrally-stratified conditions. More recent comparison with field observations, using the bug fix and corrected TKE advection (Larsén and Fischereit, 2021; Ali et al., 2023) both suggest that 100% added TKE yields better agreement with observations. Given this uncertainty in the literature and the experience of the sensitivity study of Rosencrans et al. (2024) (their Supplemental Figure A1b and d) that the amount of added TKE makes a very subtle impact on surface fluxes, we return to the standard default value of 100% added TKE for these simulations.**

*Minor issues*

1. *It is unclear why the runs named "d" were separated out and studied here. The wind speed (0-3 m/s) is below the cut-in wind speed of the turbines, thus there is no power extraction in these runs. The results should look identical to those of the NoWF, but they do not, possibly because some turbines may be experiencing wind speeds locally that are above the cut-in. Nonetheless, I do not see any value in the analysis of these results.*

**In fact, the results for wind speeds between 0-3 m/s should not look identical to the NWF runs because there is a standing thrust coefficient included in the WRF wind farm parameterization that is still activated even at wind speeds below cut-in. See lines 304/305 of the WRF code at https://github.com/wrf-model/WRF/blob/master/phys/module_wind_fitch.F. The value of this analysis is in assessing the effect of the drag of the inactive blades as represented by the standing thrust coefficient.**

2. *Figure 4: do not show QKE but TKE. QKE is only used inside the WRF model but TKE is the well-known, physically-meaningful variable of interest. Same for the discussion in Section 4.4, focus on the real variable TKE, not on the WRF-specific QKE.*

**QKE and TKE differ by a factor of two as explained in section 2.1. Regardless, the revised version of the manuscript will modify the figures and discussion to correct for this factor of two to focus on TKE.**

3. *Eq. 1: The flux should be the **virtual** heat flux, not just the heat flux, typo perhaps. Also, I assume this is calculated from the WRF output fields, but it is unclear which level(s) was chosen for the mean theta_v.*

**We have corrected the equation and the text to note that we calculated the mean value of virtual potential temperature from the 2-m temperature, surface pressure, and surface water vapor mixing ratio. We use heat flux (which is available from WRF), not virtual heat flux (which is not available).**

4. *Remove the sentence at line 225: your results are indeed the same as those of Rosencrans et al. (2024), you can't say that they are confirming them.*

**The reviewer is correct, we have revised the sentence to:**

> **During unstable conditions, wind speeds are replenished faster due to increased mixing from aloft, which reduces the extent of the wake, as also reported in Figure 12 of Rosencrans et al. (2024).**

5. *Explain why there is a slight increase in 10-m wind speed at the wind farms in stable conditions. What mechanism can cause this unusual finding? Is it possible that it has to do with the excessive TKE from the 100% coefficient?*

**This increase in 10-m wind speed in stable conditions is not "unusual", but has been observed in long-term measurements of wake effects in stable conditions as reported in Bodini et al., (2021), their Figure 4 and in the measurements of Rajewski et al. (2013), their Table 2. It has also been seen in stably-stratified large-eddy simulations (Vanderwende et al., 2016). When stably-stratified flow encounters an obstacle (such as a wind turbine, which is a porous obstacle but an obstacle nonetheless), flow is diverted around the obstacle and accelerates to fulfill mass conservation. In this case, the flow is diverted not just around the obstacle of the wind turbine rotor disk but also under it.**

6. *Line 294: is the downward heat flux is reduced, thus less heat comes down to the surface from the layer above it, how come "more heat is transferred to the surface"?*

**Thank you for catching this typo and unclear statement. We have corrected the text to**

> **"During stable conditions, heat flux is negative. When heat flux is moderately reduced within the wind plant, it becomes less negative implying that the cooling typical of stable conditions slows, as is also seen in the T2 changes of Figure 11. We also observe an increase in heat flux downwind of the wind plant of around 1.5 W m-2 during stable conditions, implying that the cooling typical of stable conditions accelerates. No downwind effect on heat flux occurs during neutral and unstable conditions."**

7. *Figure 13: the square patterns are concerning … if they are truly due to the fact that there are more turbines in the grid cells along the lines (line 321), then why are we not seeing a similar pattern in the wind speed deficits, only in the added TKE?*

**The difference between the effect of TKE and the effect of the wind speed deficits is related to the magnitude of the changes. The added TKE at hub-height as seen in Figure 13 is more than a factor of two greater than the values of TKE in the NWF simulations, and so the addition (especially for cells with multiple turbines) is a factor of two larger than the ambient background values. In contrast, the wind speed deficit introduced by the wind farm effect is smaller than the ambient wind speed**

background values, and so the grid pattern is not clearly visible. The grids are slightly visible in other quantities that have smaller ambient background values compared to the wind-farm-induced differences, like the heat flux in stable conditions for fast winds (Figure 12f).

8. *All figures: add units*

We have double-checked all figures for units and realized they were inadvertently omitted. The revised manuscript will add them to Figure 6 (within the wind roses), and to the color bars in Figures 9, 10, 11, 12, 13, 14, 15, and B1-B6.

**References**

Akhtar, N., Geyer, B., and Schrum, C.: Larger wind turbines as a solution to reduce environmental impacts, Sci. Rep., 14, 6608, https://doi.org/10.1038/s41598-024-56731-w, 2024.

Ali, K., Schultz, D. M., Revell, A., Stallard, T., and Ouro, P.: Assessment of Five Wind-Farm Parameterizations in the Weather Research and Forecasting Model: A Case Study of Wind Farms in the North Sea, Mon. Weather Rev., 151, 2333–2359, https://doi.org/10.1175/MWR-D-23-0006.1, 2023.

Archer, C. L., Wu, S., Ma, Y., and Jiménez, P. A.: Two Corrections for Turbulent Kinetic Energy Generated by Wind Farms in the WRF Model, Mon. Weather Rev., 148, 4823–4835, https://doi.org/10.1175/MWR-D-20-0097.1, 2020.

Bodini, N., Lundquist, J. K., and Moriarty, P.: Wind plants can impact long-term local atmospheric conditions, Sci. Rep., 11, 22939, https://doi.org/10.1038/s41598-021-02089-2, 2021.

Chatterjee, F., Allaerts, D., Blahak, U., Meyers, J., and van Lipzig, N. p. m.: Evaluation of a wind-farm parametrization in a regional climate model using large eddy simulations, Q. J. R. Meteorol. Soc., 142, 3152–3161, https://doi.org/10.1002/qj.2896, 2016.

Larsén, X. G. and Fischereit, J.: A case study of wind farm effects using two wake parameterizations in the Weather Research and Forecasting (WRF) model (V3.7.1) in the presence of low-level jets, Geosci. Model Dev., 14, 3141–3158, https://doi.org/10.5194/gmd-14-3141-2021, 2021.

Rajewski, D. A., Takle, E. S., Lundquist, J. K., Oncley, S., Prueger, J. H., Horst, T. W., Rhodes, M. E., Pfeiffer, R., Hatfield, J. L., Spoth, K. K., and Doorenbos, R. K.: Crop Wind Energy Experiment (CWEX): Observations of Surface-Layer, Boundary Layer, and Mesoscale Interactions with a Wind Farm, Bull. Am. Meteorol. Soc., 94, 655–672, https://doi.org/10.1175/BAMS-D-11-00240.1, 2013.

Rosencrans, D., Lundquist, J. K., Optis, M., Rybchuk, A., Bodini, N., and Rossol, M.: Seasonal variability of wake impacts on US mid-Atlantic offshore wind plant power production, Wind Energy Sci., 9, 555–583, https://doi.org/10.5194/wes-9-555-2024, 2024.

van Stratum, B., Theeuwes, N., Barkmeijer, J., van Ulft, B., and Wijnant, I.: A One-Year-Long Evaluation of a Wind-Farm Parameterization in HARMONIE-AROME, J. Adv. Model. Earth Syst., 14, e2021MS002947, https://doi.org/10.1029/2021MS002947, 2022.

Vanderwende, B. J., Kosović, B., Lundquist, J. K., and Mirocha, J. D.: Simulating effects of a wind-turbine array using LES and RANS, J. Adv. Model. Earth Syst., 8, 1376–1390, https://doi.org/10.1002/2016MS000652, 2016.

---

## Author Comment (AC2)

*Dr. Hahmann's comments appear in italics,* **our responses appear in boldface blue text**.

*Dear authors,*
*I have two comments regarding your manuscript:*

**1 Possible feedbacks from the ocean surface**

*In my opinion, the manuscript overlooks a crucial conceptual point that could significantly impact the manuscript results and conclusions. In the WRF model, the state of the land surface is controlled by a land surface model. Over the ocean, such a model is not often included. In your simulations, the namelist.input file shows that sea surface temperatures (SSTs) are specified from an input file. Using slowly varying but fixed SSTs throughout the simulation could lead to inaccuracies in calculating changes in heat fluxes and 2-meter temperatures and possibly other derived quantities. These inaccuracies result from omitting essential thermodynamic feedback processes from the ocean to the surface layer above. This factor must be included in your manuscript to ensure a fair discussion of your findings.*

**Thank you for emphasizing this point. While we did mention the important role of coupling in our original conclusion, we have now expanded this discussion to directly address ocean mediation of wake effects. Our preliminary (unpublished) work with coupling WRF with a wave model and an ocean model indicates that, depending on the depth of the ocean mixed layer in the ocean model, the ocean may or may not react to minimize the changes caused by wind farms.**

**Our new text states:**

**"Further, because these present simulations are not coupled with a wave model and ocean model, other feedbacks between the ocean and atmosphere may be relevant. Over water, wind plant wakes may influence ocean dynamics (Raghukumar et al., 2022, 2023; Liu et al., 2023), including upwelling. Therefore, coupling with wave and ocean models could provide insight into potential wake impacts on the ocean. Daewel et al. (2022) considers the impact of offshore wakes on primary production, but additional analysis on surface currents would provide a more complete picture of wake impacts. The ocean's response may also mediate these effects of wakes on surface meteorology, as suggested by the simulations of Fischereit et al. (2022b) in the North Sea. Extended simulations, such as those shown here, with a coupled atmosphere-ocean-wave model, could provide more accurate insight into the ocean's role in modulating wake impacts. Such work is ongoing."**

*A possible mechanism in nature will be the following. In the stable case with strong winds, your results show a change in the heat from the air to the water of the order of 3 W m−2. This change in flux is positive downward, thus possibly increasing the skin temperature of the sea surface. This temperature change will decrease the vertical temperature gradient, which results in a reduced heat flux. The reduced heat flux could also alter the 2-meter temperature and the surface layer's stability. Thus, in nature, the ocean could respond to minimize the changes caused by wind farms. Or not. It could be argued, however, that the ocean has a large heat capacity and, thus, an excess of 3 W m−2 will quickly be mixed in the water column without altering the ocean's temperature. This will be linked to the stability of the ocean's surface layer. Knowing which process will dominate is only possible with measurements and accurate simulations, including the thermodynamic effect on the ocean surface. Similar processes exist in the*

*simulation of tropical cyclones, which often consider the possible ocean surface changes.*

**Thank you for the nice example demonstrating that the depth of the ocean mixed layer will play a large role in mediating wake response. We are also looking forward to having observations and/or coupled results to potentially confirm this!**

*In your article, you cite the work of Golbazi et al. (2022), which carried out similar shorter WRF model simulations with various sizes of wind turbines. From their methods and namelist, I can also see that their simulations were done with fixed (but spatially more precise) SSTs. So, their results and discussion also disregard the possible effects of the surface ocean's response to changes in surface fluxes and temperatures. So, the article cannot be used to substantiate your results.*

**We do not cite Golbazi et al. in our conclusion paragraph regarding ocean-wave-atmosphere coupling.**

*Through the results and discussion section of your manuscript, I often find that you mix land and offshore publications. The processes are different in these two environments in nature and the models and should not be mixed.*

**While we agree that many processes offshore are different from those onshore (i.e., the seasonal cycle of stability is stronger offshore while the diurnal cycle of stability is stronger onshore), we think that the fundamental physics governing these processes are the same. For example, Hogstrom and Smedman did very nice work drawing an analogy between offshore and onshore winds, in their explanation of offshore low-level jets that develop from decoupling of flow from onshore to offshore similar to the evening decoupling of the atmospheric boundary layer over land (Smedman et al., 1993, 1995, 1996). If there are specific locations where Dr. Hahmann finds inappropriate references, we will be happy to respond to these specific instances.**

*The issue of the possible impact of fixed SSTs should be discussed and addressed in your manuscript. This discussion is crucial for the reliability of your findings. It is outside the scope of your manuscript, but WRF model simulations, including the effects of a slab ocean, are possible. Running with a fully coupled ocean will be even better but expensive.*

**We certainly agree that coupled simulations are needed and this work is ongoing but outside of the scope of this particular paper. This particular contribution is envisioned as a first step. A planned follow-on paper would investigate this same set of simulations with coupled modeling. We have expanded a paragraph in the conclusions to more thoroughly discuss coupled modeling.**

**2 Dependence of your results on PBL and wind farm parameterization**

*While your results are exciting, they represent only one possible scenario with one WRF PBL scheme and one wind farm parameterization. Recent publications have shown that the impact of wind farms on the atmosphere is highly dependent on the PBL and wind farm parameterization used. The paper should emphasize this point and acknowledge that observations have yet to verify most aspects of the simulated impacts of large wind farms on the atmospheric flow.*

**We have added a paragraph in the conclusions to point out the dependence on PBL scheme:**

*"Further, simulated winds (Draxl et al., 2014; Bodini et al., 2024a; Liu et al., 2024) and simulated wakes (Rybchuk et al., 2022) show dependence on the PBL scheme chosen for the model simulations. At the moment, the Fitch wind farm parameterization is coupled only with the MYNN PBL scheme used here and with the 3DPBL scheme (Kosović et al., 2020; Juliano et al., 2021). Future work could assess how micrometeorological responses to wind farm wakes depend on the choice of PBL scheme."*

*Statements like the one in the abstract, "Offshore wind energy projects are cur- rently in development off the east coast of the United States and will likely influence the local meteorology of the region." should be avoided. It is all a question of degrees and assessment of significance.*

**Respectfully, we have received many many specific questions about micrometeorological impacts of offshore wind projects from industry and from the public, so it is important to acknowledge that impacts may occur. We have softened the sentence from "will likely" to "may".**

**References**

- **Bodini, N., Optis, M., Liu, Y., Gaudet, B., Krishnamurthy, R., Kumler, A., Rosencrans, D., Rybchuk, A., Tai, S.-L., Berg, L., Musial, W., Lundquist, J., Purkayastha, A., Young, E., and Draxl, C.: Causes of and Solutions to Wind Speed Bias in NREL's 2020 Offshore Wind Resource Assessment for the California Pacific Outer Continental Shelf, Tech. Rep. NREL/TP–5000-88215, 2318705, MainId:88990, https://doi.org/10.2172/2318705, 2024a.**
- **Daewel, U., Akhtar, N., Christiansen, N., and Schrum, C.: Offshore wind farms are projected to impact primary production and bottom water deoxygenation in the North Sea, Communications Earth & Environment, 3, 1–8, https://doi.org/10.1038/s43247-022-00625-0, number: 1 Publisher: Nature Publishing Group, 2022.**
- **Draxl, C., Hahmann, A. N., Peña, A., and Giebel, G.: Evaluating winds and vertical wind shear from Weather Research and Forecast- ing model forecasts using seven planetary boundary layer schemes, Wind Energy, 17, 39–55, https://doi.org/10.1002/we.1555, _eprint: https://onlinelibrary.wiley.com/doi/pdf/10.1002/we.1555, 2014.**
- **Fischereit, J., Larsén, X. G., and Hahmann, A. N.: Climatic Impacts of Wind-Wave-Wake Interactions in Offshore Wind Farms, Frontiers in Energy Research, 10, https://doi.org/10.3389/fenrg.2022.881459, publisher: Frontiers, 2022b.**
- **Juliano, T. W., Kosović, B., Jiménez, P. A., Eghdami, M., Haupt, S. E., and Martilli, A.: "Gray Zone" Simulations using a Three- Dimensional Planetary Boundary Layer Parameterization in the Weather Research and Forecasting Model, Monthly Weather Review, -1, https://doi.org/10.1175/MWR-D-21-0164.1, publisher: American Meteorological Society Section: Monthly Weather Review, 2021.**
- **Kosović, B., Jimenez Munoz, P., Juliano, T. W., Martilli, A., Eghdami, M., Barros, A. P., and Haupt, S. E.: Three-Dimensional Planetary Boundary Layer Parameterization for High-Resolution Mesoscale Simulations, Journal of Physics: Conference Series, 1452, 012 080, https://doi.org/10.1088/1742-6596/1452/1/012080, 2020.**
- **Liu, K., Du, J., Larsén, X. G., and Lian, Z.: Spatiotemporal Variations of Ocean Upwelling and Downwelling Induced by Wind Wakes of Off- shore Wind Farms, Journal of Marine Science**

and Engineering, 11, 2020, https://doi.org/10.3390/jmse11102020, number: 10 Publisher: Multidisciplinary Digital Publishing Institute, 2023.

- Liu, Y., Gaudet, B., Krishnamurthy, R., Tai, S.-L., Berg, L. K., Bodini, N., Rybchuk, A., and Kumler, A.: Identifying Meteorological Drivers for Errors in Modeled Winds along the Northern California Coast, Monthly Weather Review, 152, 455–469, https://doi.org/10.1175/MWR-D-23-0030.1, publisher: American Meteorological Society Section: Monthly Weather Review, 2024.

- Raghukumar, K., Chartrand, C., Chang, G., Cheung, L., and Roberts, J.: Effect of Floating Offshore Wind Turbines on Atmospheric Circu- lation in California, Frontiers in Energy Research, 10, https://doi.org/10.3389/fenrg.2022.863995, publisher: Frontiers, 2022.

- Raghukumar, K., Nelson, T., Jacox, M., Chartrand, C., Fiechter, J., Chang, G., Cheung, L., and Roberts, J.: Projected cross-shore changes in upwelling induced by offshore wind farm development along the California coast, Communications Earth & Environment, 4, 1–12, https://doi.org/10.1038/s43247-023-00780-y, publisher: Nature Publishing Group, 2023.

- Rybchuk, A., Juliano, T. W., Lundquist, J. K., Rosencrans, D., Bodini, N., and Optis, M.: The sensitivity of the fitch wind farm parameteriza- tion to a three-dimensional planetary boundary layer scheme, Wind Energy Science, 7, 2085–2098, https://doi.org/10.5194/wes-7-2085- 2022, publisher: Copernicus GmbH, 2022.

- Smedman, A.S., Tjernström, M. and Högström, U., 1993. Analysis of the turbulence structure of a marine low-level jet. Boundary-layer meteorology, 66, pp.105-126.

- Smedman, A.S., Bergström, H. and Högström, U., 1995. Spectra, variances and length scales in a marine stable boundary layer dominated by a low level jet. Boundary-Layer Meteorology, 76(3), pp.211-232.

- Smedman, A.S., Högström, U. and Bergström, H., 1996. Low level jets–a decisive factor for off-shore wind energy siting in the Baltic Sea. Wind Engineering, pp.137-147

---

## Author Comment (AC3)

*Reviewer's comments appear in italics,* **our responses appear in boldface blue text**.

*General comments:*

*This article explores the impact on a range of meteorological fields by the presence of large wind farm cluster off the USA east coasts using the WRF mesoscale model and using the WFP wind farm parameterization, using the 100% added TKE option.*

*Overall I find the article outlines a repeatable methodical approach and describes results, but lacks clarity on the motivation for the investigation and lacks discussion on the limitations of the method. What conclusions are to be drawn beyond describing the response of a model (WRF) in these "no wind farm" and "with wind farm" simulations? What research question is being asked? What is the hypothesis being tested? Please revise to address this.*

**We thank the reviewer for their time and consideration in reviewing our manuscript. We have added the following paragraph to the Introduction:**

**"Given the scarcity of comprehensive offshore observations along the U.S. East Coast, this study aims to complete the first year-long assessment of how modeled offshore wind plants influence the modeled local environment. We achieve this by comparing WRF model (Skamarock et al. 2021) simulations with and without wind plants included. Our analysis focuses on the Massachusetts-Rhode Island offshore wind lease area, where we quantify the difference in hub-height and 10-m wind speed, boundary-layer height, 2-m temperature, surface heat flux, and TKE at the surface and at hub height. Our expectation is to demonstrate that different stability conditions are a key driver of the simulated micrometeorological impacts, and that these impacts also vary with different wind speeds, as wind turbine operation changes. Furthermore, we aim to assess the relationship between boundary-layer height and the extent of wind plant wakes, hypothesizing that deeper boundary layers will limit the extension of these wakes."**

**As detailed in our other responses to the reviewer's comments, we have also added more discussion throughout the manuscript, including the conclusions.**

*The choice of one WRF set-up, one WRF wind farm parameterization, and one setting for the added TKE option, is a severe limitation of the article. It means that the whole paper becomes a description of model results, rather than focussing on what might actually happen in nature itself.*

**While we agree that the question of what actually happens in nature itself is extremely interesting, a very limited set of observations are available to quantify the real-world impact of wakes. In the absence of extensive observations, modeling studies such as the one presented here are needed, and typically used, to fill the knowledge gap. The modeling tools used here have repeatedly demonstrated results consistent with available observations in other locations (i.e., the comparison of the modeling studies of Xia et al. with the observations of Zhou et al., the intercomparison of aircraft observations with modeling studies of Siedersleben et al. 2018, 2020, and the validation studies of Larsén and Fischerei 2021 and Ali et al. 2023). However, the micrometeorological impacts of wakes over a complete annual cycle in this region with intensively planned offshore wind development has not yet been investigated, hence the effort here.**

Regarding the choice of one WRF set-up: this set-up was based on a comparison of 16 set-ups in this region (Bodini et al. 2024 ESSD) and was the best-performing set-up. We have added a sentence to the beginning of Section 2:

*"We note that the WRF setup used here resulted from a comparison of 16 different WRF setups against an observational dataset; this setup was the best performer (Bodini et al., 2024)."*

*The manuscript needs to be revised to include a comprehensive discussion of the limitations of WRF-WFP, and what that might mean for the given results.*

We have added a discussion to the conclusion about the existing validation of WRF-WFP and other ongoing research efforts that might affect the given results.

*"Of course, this study relies on the accurate representation of wakes in the Fitch WRF wind farm parameterization. While wakes simulated with this parameterization compare reasonably well with the limited sets of observations available (Lee and Lundquist, 2017; Siedersleben et al., 2018b, a, 2020; Ali et al., 2023; Larsén and Fischereit, 2021), the availability of observations of wake effects at multiple distances and heights from wind farms, especially offshore, is limited. Ongoing experiments such as AWAKEN (Moriarty et al., 2024) may provide more extensive datasets to support modifications to wind farm parameterizations in mesoscale models. Additionally, comparisons of these mesoscale representations to more finely resolved large-eddy simulations of wind farms (Vanderwende et al., 2016; Peña et al., 2022) may suggest other improvements, although these comparisons should be carried out for a range of atmospheric stability conditions and wind farm geometries. Particular attention should be paid to effects on surface meteorology as well as dynamics directly relevant to wind turbine power production".*

*Please include more justification for the model set-up, for example, why only a one year simulation? How might a longer period or different year impact the results?*

While of course a longer set of simulations would be interesting, the work presented here includes a complete annual cycle, which goes far beyond other wake studies in this region, i.e., 55 days (Pryor et al. 2021) or three months (Golbazi et al. 2022) investigated in previous work. This particular year was chosen, as discussed in Rosencrans et al. (2024), because of the availability of lidar data for validation of the no-wind-farm simulation.

Because this year includes a range of stability conditions and wind speeds typical for this region these results are not particularly sensitive to the choice of this particular year.

*I think there is a lack of physical mechanisms, and where mechanisms are conjectured, no model fields are used to back these up (see specific comments).*

We list below the main physical explanations we provide in the paper about the (modeled) changes in atmospheric variables. We have added some and expanded most of the existing ones, on top of what already discussed in the specific comments below:

- Hub-height wind speed:

  *"Within the wind plant, wind speeds are reduced by up to 2.7 m s-1 in stable conditions, and up 1.5 m s-1 in unstable conditions, as turbines extract momentum from the flow."*

*"During unstable conditions, wind speeds are replenished faster due to increased mixing from aloft (Abkar et al. 2013), which reduces the extent of the wake"*

[revised manuscript text omitted]

*The paper several times states where results confirm what is already published, as a reader I would like more clarity on what are the most novel parts of the study and what led to these novel parts being of interest for investigation. Please revise to address this.*

**As noted above, we have introduced a section in the introduction clearly stating the novelty of the study in looking at variability of wake impacts over an entire annual cycle. We have also revised the conclusions to emphasize the annual variability as well as the machine-learning approach demonstrated here for wake area and wake length characterization:**

*"We also develop and demonstrate a machine-learning approach to identify wind plant wakes, and use this method to demonstrate the relationship between boundary-layer height and both the area and length of the wind plant wake."*

*Latter sections seem a bit rushed.*

**If the reviewer is requesting additions to the conclusion, we have expanded the conclusions and discussion of the results therein considerably.**

*Adding to the limitation discussion, it would be good to include what would be good further studies to pursue, and what might be an approach to the difficult question of validation. Please revise to address this.*

**As noted above, we have added a discussion to the conclusion about the existing validation of WRF-WFP and other ongoing research efforts that might affect the given results.**

*"Of course, this study relies on the accurate representation of wakes in the Fitch WRF wind farm parameterization. While wakes simulated with this parameterization compares reasonably well with the limited sets of observations available (Lee and Lundquist, 2017; Siedersleben et al., 2018b, a, 2020; Ali et al., 2023; Larsén and Fischereit, 2021), the availability of observations of wake effects at*

*multiple distances and heights from wind farms, especially offshore, is limited. Ongoing experiments such as AWAKEN (Moriarty et al., 2024) may provide more extensive datasets to support modifications to wind farm parameterizations in mesoscale models. Additionally, comparisons of these mesoscale representations to more finely resolved large-eddy simulations of wind farms (Vanderwende et al., 2016; Peña et al., 2022) may suggest other improvements, although these comparisons should be carried out for a range of atmospheric stability conditions and wind farm geometries. Particular attention should be paid to effects on surface meteorology as well as dynamics directly relevant to wind turbine power production."*

*Specific comments:*

*L16: "exceeding 100 m" -> "exceeding 1000 m"?*
**We have removed numbers from this sentence to avoid ambiguities.**

*L52: "extreme scale" , suggest changing this term. "Extreme" 10 years ago is not "extreme" today.*
**We have replaced it with "bigger".**

*L81: The sentence "determine … how ..  influence the local environment", it should be reformulated to say this is modelled local environment  being investigated, not the actual environment in nature.*
**We have changed it to "modeled environment".**

*L107: Please detail more about what is meant by "the model produced unrealistic wind speeds, … ". Please describe and state what it is that is unrealistic.*
**We have expanded this discussion of the literature:**
*"Vanderwende et al. (2016) suggest that added TKE is critical. In that study, when TKE generation within the wind farm parameterization is disabled, the model produced unrealistic wind speeds, wind directions, and turbulence as compared to large-eddy simulations, with too-small of values of turbulence and too large of decreases in wind speed."*

*L96 and Table 1: Why was this period chosen?*
**We have rephrased as "NOW-WAKES covers from 1 September 2019 00:00 UTC - 31 August 2020 23:50 UTC (chosen to overlap with lidar data availability in the region) at 10- minute resolution".**

*L94 and Figure 1: Why was the domain chosen as it is? What is the reason for the far eastward extent?*
**We have added a sentence explaining:**
*"The Rosencrans et al. (2024) domain is consistent with other datasets for this region (Xia et al., 2022; Redfern et al., 2021; Bodini et al., 2024) and was initially chosen to optimize processor partitioning for the WRF simulations."*

*Figure 3: It is strange to have a caption referring to a later caption.*
**We have changed the captions of Figures 3, 4, and 5 accordingly.**

*L136: In the description of the BLH definitions, what happens in transitions from one stability condition to another, is there a discontinuity in the BLH? Could the authors use a sentence or to to justify the use of the approach of Olson et al (2019) for this analysis. What is the most relevant BLH determination for a wind farm do the authors think or recommend?*

**Yes, during stability transitions, there may be discontinuities in the estimation of BLH by the WRF model because of the transition from one approach to another. Because these simulations use the MYNN PBL scheme, the authors recommend using the PBLH estimation approach included in that scheme (the Olson et al. (2019) approach) for consistency. This approach has performed well in comparison to observations for some case studies (Bauer et al. 2023).**

*Table 2: It might be better to have "region 1" and "region 2" also part of the wind speed column in this table, to remind the reader of the reasoning behind the wind speed partitioning.*
**We have added references to Region 1, 2, and 3 in the Table.**

*L162: I am a bit wary about this statement about the "tight coupling" because it suggests that everything can be explained by atmospheric stability, but there may be very important other aspects of the profile, and these might be overlooked by this approach. Please expand on the justification of the approach.*
**We have replaced it as "correlation" to soften the message of the sentence.**

*L166: "a leveling of the power production", I think a better term here would be "the rated power production being reached and not increasing further".*
**Changed.**

*L167: "To isolate", again similar to the L162 comment. It is not just wind speed that is varying, even though you keep stability and direction within a certain band. Please discuss other things in the profile that might vary, given this constraint on stability and direction.*
**We have rephrased it as "To more clearly identify".**

*L175: Why is 1 m/s deficit chosen as the measure of a wake? Why not other measures, such as relative deficit? What are the advantages of this measure, what is the impact of different wind speeds (NWF simulations) on this wake definition?*
**We chose an absolute (rather than relative) definition of the wake threshold to be consistent with previous work. A relative deficit requires comparison with spatially heterogeneous unwaked fields which can make the assessment of the wake even noisier than an absolute definition. In the text, we have added an explanation:**
***"This wake definition is stronger than the 0.5 m s-1 threshold used in Golbazi et al. (2022); Rybchuk et al. (2022); and Rosencrans et al. (2024), and was chosen to aid in identifying contiguous wakes. A relative wake definition proved problematic by making the wake field even noisier."***

*L178: Please explain why there are "not contiguous" wind speed perturbations, could they be related to the wake? How do you discount that there may be a distant response to the wind farm, perhaps oscillation in wake above and below the 1 m/s threshold that has been chosen.*

**The WRF wind farm parameterization is known to produce noise in wind fields similar to these remote patterns. We have added a sentence:**

***"The deficits at these remote locations are presumed to be numerical noise as identified in Ancell et al. (2018); Lauridsen and Ancell (2018) and discussed in Appendix F of Rosencrans et al. (2024)."***

*L192: "ill defined" wakes. This seems a bit subjective to me, perhaps wakes are not neat and tidy as we might expect. Please justify. And is the 15.2% of hours with "ill defined" wakes not quite a significant share of the time?*

**Of course we do acknowledge that this definition is by necessity somewhat subjective. To address this subjectivity and to enable the analysis to be replicated by other research groups, we have clearly defined the criteria used to make a distinction between the 85% clearly defined wakes and the 15% ill defined wakes. While 15% of the wakes is a not trivial share of the time, it is clearly a minority of the time. Further, it is consistent with other machine learning approaches used to identify wind turbine wakes in heterogeneous fields, such as 87.18% in Aird et al. (2021), although in Aird et al. they are identifying wakes from individual turbines and not wind farm wakes.**

*L211: Please can the authors explain why the wake is compared in wind speed across the different stability classes? Is the mean wind speed the same for the different stability classes, if not, the difference in wake deficit can be partly due to this effect.*

**We have addressed the roles of wind speed and stability by partitioning our results by both stability and by wind speed within the stable stratification class. We first emphasize stability classes because of the long history of observations that wakes are stronger in stably stratified conditions (e.g., see the summary in section 2.3 of Porté-Agel et al. 2020, with a sample of over 20 investigations documenting wake variability with atmospheric stability). In Figure 6 of the current manuscript we already demonstrate that the wind speed and direction distributions are different for stable vs neutral vs unstable conditions, demonstrating that faster winds occur in stable conditions. By further partitioning the stable results into the different wind speed regimes, we identify the differences in wakes due just to wind speed variation.**

*L214: The authors write ""due to increased mixing from aloft", but this statement is not argued with data from the model, but appears to be more like a hypothesis for a possible, and plausible mechanism. Please justify the statement or rephrase it.*

**We justified the statement by including a reference to the sensitivity of wake replenishment from above (Abkar and Porte-Agel 2013).**

*L219: It would help the reader to refer to region 3 next to the "above 11 m/s".*

**We have modified the sentence to read "*Hub-height wind speeds are reduced by up to 2.5 m s-1 for wind speeds in Region 2 of the turbine power curve, and up to 3.6 m s-1 for wind speeds above 11 m s-1 in Region 3 of the power curve.*"**

*L225-228: Does this effect also show when wind speeds are in the range 15 m/s - 25 m/s where the thrust is dropping significantly? See Fig 2b.*

**Thank you for this suggestion. As seen in the comparison of e) and f) below, the magnitude of the wind speed deficit decreases for wind speeds faster than 15 m s-1. We have added this figure to the appendix and expanded the discussion in the text:**

***"Of note, when the wind speeds exceed 15 m s-1 when the thrust coefficient is very small, the wind speed deficit starts to decrease again (see Appendix Fig. B1)."***

[Figure]

*L233: Same question as above.*

**Thank you also for this suggestion. As seen in the comparison of e) and f) below, the magnitude of the 10-m wind speed acceleration increases for wind speeds faster than 15 m s-1. We have added this figure to the appendix and expanded the discussion in the text:**

***"For wind speeds faster than 15 m s-1, the accelerations are more widespread within the wind plant but the maximum accelerations are not faster than those in the range of 11 - 15 m s-1 (see Fig. B2)."***

[Figure]

*L240: "reduced more", more than what? Does the deficit increase, or does the absolute wind speed reduce? It reads more like the latter, but I think it is the former.*

**We have rephrased it to "Under stable conditions and southwesterly winds, the deficit in 10~m wind speeds increases more with increasing ambient wind speeds".**

*L251: Please quantify "increase slightly".*

**We have rephrased to "At 10 m, wind speeds accelerate slightly (less than 1 m s-1 within the wind plant during…"**

*L260: Temperature increases by "around 0.05 degrees". Is this significant?*

**As the topic sentence of this paragraph suggests, these changes are "small". We have added "only" to the sentence including "0.05 degrees".**

*L261-263: Are these statements conjecture or justified by model fields of fluxes? Please reformulate so it is clearer.*

**These statements are not conjecture but are rather consistent with a wide body of literature discussing mixing mechanisms dating back to Baidya Roy et al. 2004 and demonstrated in Fitch et al. (2012). We have reformulated the sentences as follows:**

**"During stable conditions, turbines mix warmer air from aloft down to the surface, resulting in a temperature increase (as also discussed in Fitch et al. (2013) and Siedersleben et al. (2018a), among others). In unstable conditions, the boundary layer is already well mixed, so that any mixing by wind turbines is simply remixing a well-mixed layer."**

*L289: The heading "heat flux", please clarify what kind of heat flux is being looked at. Surface heat flux, vertical heat flux, sensible heat flux, etc, etc.*
**We have renamed the heading to "Wind plant wake impacts on surface sensible heat flux".**

*L407: The use of the word "promote" infers a causal relationship, is that what is meant?*
**Yes, we intend to suggest a causal relationship.**

---

## Author Comment (AC4)

**We have noticed that our previous answer to the reviewer's minor issue #6 had an error. The correct answer is below, and we have updated the text in the draft accordingly:**

*Line 294: if the downward heat flux is reduced, thus less heat comes down to the surface from the layer above it, how come "more heat is transferred to the surface"?*

**Thank you for catching this unclear statement. We have corrected the text to**

*"During stable conditions, heat flux is negative (downward). When heat flux is moderately reduced within the wind plant, it becomes more negative implying that more heat is transferred to the surface, which is consistent with the T2 changes of Figure 11. We also observe an increase in heat flux downwind of the wind plant of around 1.5 W m-2 during stable conditions, implying that the cooling typical of stable conditions accelerates. No downwind effect on heat flux occurs during neutral and unstable conditions."*

---

## Referee Report (RR1)

**Second review of "Simulated Meteorological Impacts of Offshore Wind Turbines and Sensitivity to the Amount of Added Turbulence Kinetic Energy" by Quint, Lundquist, Bodini, and Rosencrans, submitted to Wind Energy Science**

The authors are praised for their excellent job at improving the paper. I appreciate their efforts in doing additional simulations with 0% added TKE, although I would argue that 0% added TKE is unrealistic. The revised paper now provides upper and lower bounds to the effects of TKE and even interesting insights to what is going on in the ABL. As such, I recommend that the manuscript be published after the minor revisions below.

**Minor Remarks**

- 1. L. 45: The authors state that the maximum wind speed deficit is "generally occurring in the top half of the rotor disk", while it is generally near or at hub height. For example, all analytical wake loss models, like Jansen or Gaussian, predict it at hub height. The literature is rich of such evidence (e.g., Abkar and Porte-Agel 2015, Fig. 5; Xie and Archer 2017 Fig. 4). The paper cited to support the statement about the upper rotor peak (Bodini et al. 2021) refers to a figure for one location at two instantaneous times during which the maximum was at about 100 m, for a hub height of, I believe, 91 m. This is not a big deal by any means, but I would prefer to read that the maximum wind speed deficit is near hub height.
- 2. L. 50: By contrast, here the authors mention that the peak in TKE occurs "in the upper part of the rotor" with no reference. Generally the peak in TKE is actually near the rotor tip (e.g., Abkar and Porte-Agel 2015, Fig. 8; Lu and Porte-Agel 2015, Fig. 12). No big deal, but it would be more accurate to state that the peak in TKE is generally near the rotor tip.
- 3. L. 139: How can the temporal resolution of the WRF model be 10 minutes? Perhaps you mean the resolution of the output, not that of the NOW\_WAKES simulations?
- 4. L. 150: How are the wind turbines distributed in the grid cells (i.e., one or two per grid cell)? What layout was used (if regular, along which directions)? Please add a few sentences to clarify this, since it becomes important later in the discussion of the added TKE spatial patterns.
- 5. L. 163: There appears to be confusion between  $C_{TKE}$  and  $\alpha$ . It is  $\alpha$  that is 1.0 in the case of 100% added TKE, not  $C_{TKE}$ , as  $C_{TKE}$  is not constant and not equal to one.
- L. 164: unfinished sentence? Perhaps it should be: "default, since comparisons with large-eddy simulations by Vanderwende et al. (2016) and Garcia-Santiago et al. (2024) suggest ...".
- 7. L. 185: How do you determine stability here? One value for all lease areas or one value per grid cell? I believe you talk about it later (Eq. 4); if it's the same approach, then perhaps tell the reader that the info about stability will be discussed shortly.

- 8. Eq. 4: I am confused about the heat flux here. Since it is based on  $\theta'$ , it is a potential temperature flux, thus not the same as the heat flux that is discussed, for example, in Section 4.4. How do you calculate it? Or, is L a direct output of the WRF? If so, please state it.
- 9. L. 250: also Golbazi et al. (2022) reported and discussed the same numerical noise issue.
- 10. Figure 7: How do you ultimately exclude the light-blue numerical patterns in c)?
- 11. Figures 9–22: It is difficult to discern the boundaries of the wind farms when they are grey-shaded but hidden below the shades of whichever field is being plotted. Would it be possible to replace the grey-shaded areas with black contours? This is not a requirement, just a suggestion.
- 12. Figure 10: What is the small grey shape in the middle of e)?
- L. 309: I would also conclude that the hub-height wind speed deficit is basically insensitive to the amount of added TKE, consistent with Archer et al. (2020), Fig. 7, as long as TKE advection is on.
- 14. L. 341: The sentence starting with "At 10 m" is a repetition of what was already discussed at L. 316–324. Consider removing it.
- 15. L. 356: Similarly, the sentence starting with "Increases in TKE" is redundant.
- 16. Section 4.2.2: An important finding to add is that TKE at the surface is reduced in neutral and stable (fast-wind) conditions in the wakes regardless of the amount of TKE added.
- 17. General: The manuscript now provides excellent evidence of the links between TKE and 2-m temperature through the heat flux changes. Figures 15–20 now tell a coherent story. Well done.
- 18. L. 408: I think that the discussion could be simplified and easier to explain and understand if the authors used the magnitude of the heat fluxes, rather than expressions such as "more negative heat flux". Here is what I mean. The magnitude of the heat flux is directly related to the amount of added TKE near the surface. Regardless of the sign of the heat flux, with more TKE the magnitude increases and with less TKE the magnitude decreases. Thus, in stable conditions, we know from Fig. 15 that TKE at the surface increases in the wind farms and decreases downwind in their wakes; therefore the magnitude of the heat flux is larger inside the farms and smaller in the wakes. Therefore, in stable conditions with negative heat fluxes, this causes warming inside the wind farms (large negative minus small negative equals small negative equals small positive upward, thus cooling). This is now beautifully demonstrated in the paper. Since added TKE is 100%, the warming inside the farm is somewhat strong, because there is a lot of added TKE there.

With 0% added TKE, we do not see the warming inside the wind farms because there is no added TKE there, but the reduction in the wake remains, thus the cooling and the positive heat flux difference (i.e., reduction of the heat flux magnitude, thus less warming coming down, thus cooling). 19. L. 434–439: Can we conclude that 0% added TKE is unphysical?

**Typos and style**

- 20. L. 3: The fist sentences of the abstract are about potential impacts of the future wind farms, because they are not built yet and therefore we cannot know for sure. As such, terms like "may" (L. 1) and "potential" (L. 4) are correctly used to indicate uncertainty. But on L. 3, the strong future tense "will" is used ("how future plants will change local meteorology"). I recommend the use of "may" instead of "will", given that any single sentence, out of context, can potentially be used to exaggerate such impacts.
- L. 10 (and many other instances): one should write either "temperature at 2 m" or "2-m temperature", with a dash "-" between 2 and m. Same for "2-km resolution" at L. 137 or "10-m wind speed" at L. 173.
- 22. General: acronyms are sometimes defined but not used, or not needed, or not defined. For example, WRF is not defined but used at L. 122 (it was originally defined at L. 92 but then erased for some reason). PBLH is defined at L. 81 but not used in the next two sentences. IBL is not really needed (L. 84).
- 23. General: Remove the space between 100 and %, thus "100%" (like at L. 108), not "100 %" (like at L. 111). Same for 0% and 25%.
- 24. L. 167: weird use of "of". Perhaps rephrase as: "too-small values of turbulence and too large decreases in wind speed".
- 25. L. 217: mispelling of "calculated".

---

## Author Response (AR2)

**Response to Associate Editor**

*Editor's comments appear in italics*, **our responses appear in boldface blue text**.

Dear authors,

Thanks for the revised version of the manuscript as well as for answering all the reviewers' comments and the interactive public comment. I would like to address that I do agree with the reviewers in some if not all of the main aspects that make the manuscript unpublishable at this stage. Particularly, after looking at your responses and the track-changes version, I find that:

1. The manuscript is still quite focused on describing the response of the model by looking at specific variables but I cannot see that clear specific research questions are being asked. An example of this is the abstract: more than half (the second half) of it addresses the specific results of your findings (lines 9-17) but then what? Can we trust these results or how uncertain they are? Is local meteorology really being affected?

**Thank you for clearly expressing this concern. In light of this and other questions raised by the associate editor and reviewers, we have reformulated the manuscript to focus on the sensitivity of micrometeorological impacts of wakes to the amount of added turbulence kinetic energy in the Fitch wind farm parameterization and included new figures showing how the results are sensitive to the amount of added TKE, to hopefully guide future research efforts. This focused research question is reflected in the new title, "Simulated Meteorological Impacts of Offshore Wind Turbines and Sensitivity to the Amount of Added Turbulence Kinetic Energy", a revised introduction and discussion of the results, as well as revised and/or additional Figures (9 through 22).**

**We have highlighted the research questions being investigated in our analysis in the Introduction:**
**"Given the scenario above, the following research questions guide this study:**
- **What are the year-long impacts of offshore turbines on simulated local meteorology?**
- **How does atmospheric stability influence the results?**
- **How does varying the amount of added TKE in the WRF wind farm parameterization (WFP) affect the results of the above questions?**
- **Can a reliable method be developed to automatically estimate wake characteristics from WRF WFP simulations?**
- **What is the relationship between simulated boundary-layer height and the extent of simulated wind plant wakes?"**

**Finally, the question "Is local meteorology really being affected?" is a question that simply cannot be answered with a simulation-based study. However, several simulation-**

based studies have been published on different aspects of wakes, demonstrating the broad interest in simulation-based assessments of local meteorology. For example, Golbazi et al. (2022) carry out a simulation-based study in this area over three months. To assess possible (entirely model-based) impacts of wakes on upwelling, Raghukumar et al. (2022, 2023) carry out simulation-based studies.

2. The choice of the setup. I understand this was the best setup from the 16 setups compared in Bodini et al. (2024) but I guess that that study only looked at setups with one WFP under one WFP setting? The challenge here is that your focus is on the effect of this WFP and it becomes really difficult to judge whether the impact of the wind turbines are much different when using other WFP or other WFP settings. So the question is also whether the title of the manuscript does really reflect what you describe in your work. I think both reviewers are really concerned about this issue.

We have changed the title of the paper to "Simulated Meteorological Impacts of Offshore Wind Turbines and Sensitivity to the Amount of Added Turbulence Kinetic Energy" to more accurately reflect the new focus on the impact of the amount of added TKE. Of course other model setups may have slightly different results, but it would not make sense to carry out large simulations for a setup that, in the no-wind-farm scenario, did not agree well with the available observations.

I also agree with you in that, e.g., 25% TKE is not a settle number, but that is perhaps not the main point. The main point is whether the impact on the variables you study will be significantly different if you use 33% or 25% or similar. I am not sure how to address this challenge (if you should do other simulations or similar) but it is a strong concern.

The original version of our manuscript used only the results with 100% TKE. To address this concern about the impact of a range of TKE values, our revised version now includes the same assessments with 0% TKE. These two values represent the bounds of possible values, and so bound the range of possible impacts.

3. You add some results of Rosencrans et al. (2024) to justify your choice of setup. First, I think that somewhere in the introduction you should briefly tell what are the differences between this work and that of Rosencrans et al. (2024).

We have added a sentence to the introduction:
"This focus on meteorology distinguishes this contribution from that of Rosencrans et al. (2024), who focus on hub-height wind speed and power production impacts."

 Second, the sensitivity study seems very short and difficult to scale to one year.

We are puzzled at this suggestion that one year is a short time for a sensitivity study considering that Golbazi et al.'s sensitivity study (2022) uses three months, Pryor and

**Barthelmie (2021, 2024) use 55 days. Our one year significantly extends the record of consideration.**

Third, and related to the community comment, could the subtle impacts you mentioned this latter study found be related to the way the simulations are performed? By "fixing" SSTs, impacts of close to surface variables might be minimal given this type of setup. I understand the challenge of coupling with a wave model, but then what if it does have a strong influence? Can we perhaps use a slab surface model over water instead and not necessarily coupling if we need to study effects close to surface? The point here, again, is that you are examining the local and close to surface impacts of your simulations but this could potentially be greatly impacted by the way you simulate them. Perhaps for analyzing wake areas and the extend of wakes, the simulations as they were carried out can be of great use but can they really be used to study these local/surface effects?

**We agree that the possible impact of the surface is an interesting area of future research, and we are in fact working on coupled simulations. (In general, the coupled simulations minimize wake effects even more than what has been shown here.) We appreciate that in other correspondence the editor has agreed that coupled models are not necessary at this stage of the work. This present work advances the science by assessing seasonality (beyond the one season or 55 days covered in previous studies), and future work should assess the impact of ocean-atmosphere coupling. Such an advance is beyond the scope of this paper, and we have added a full paragraph to the Conclusions to suggest this area as future work.**

**References**

**Golbazi, M., Archer, C.L. and Alessandrini, S., 2022. Surface impacts of large offshore wind farms. *Environmental Research Letters*, *17*(6), p.064021.**

**Pryor, S.C., Barthelmie, R.J. and Shepherd, T.J., 2021. Wind power production from very large offshore wind farms. *Joule*, *5*(10), pp.2663-2686.**

**Pryor, S.C. and Barthelmie, R.J., 2024. Wind shadows impact planning of large offshore wind farms. *Applied Energy*, *359*, p.122755.**

**Raghukumar, K., Chartrand, C., Chang, G., Cheung, L. and Roberts, J., 2022. Effect of floating offshore wind turbines on atmospheric circulation in California. *Frontiers in Energy Research*, *10*, p.863995.**

**Raghukumar, K., Nelson, T., Jacox, M., Chartrand, C., Fiechter, J., Chang, G., Cheung, L. and Roberts, J., 2023. Projected cross-shore changes in upwelling induced by offshore wind farm development along the California coast. *Communications Earth & Environment*, *4*(1), p.116.**

---

## Author Response (AR3)

**Response to Associate Editor**

*Editor's and reviewers' comments appear in italics*; **our responses appear in boldface blue text**.

*Dear authors,*

*I am recommending the manuscript to be accepted for publication subject to minor revisions that both of the reviewers want you to address.*
*Please also note that one of the reviewers commented on your tracked-changes file, which seemed to not have implemented all the changes you discussed.*

*Regards,*
*Alfredo*

**Thank you for your support of our research. We have addressed all the reviewer requests below and ensured that this tracked-changes file includes all the changes requested.**

**Response to First Reviewer**

*Second review of "Simulated Meteorological Impacts of Offshore Wind Turbines and Sensitivity to the Amount of Added Turbulence Kinetic Energy" by Quint, Lundquist, Bodini, and Rosencrans, submitted to Wind Energy Science*

*The authors are praised for their excellent job at improving the paper. I appreciate their efforts in doing additional simulations with 0% added TKE, although I would argue that 0% added TKE is unrealistic. The revised paper now provides upper and lower bounds to the effects of TKE and even interesting insights to what is going on in the ABL. As such, I recommend that the manuscript be published after the minor revisions below.*

**Thank you for the helpful suggestions which have improved the manuscript.**

*Minor Remarks*

*1. L. 45: The authors state that the maximum wind speed deficit is "generally occurring in the top half of the rotor disk", while it is generally near or at hub height. For example, all analytical wake loss models, like Jansen or Gaussian, predict it at hub height. The literature is rich of such evidence (e.g., Abkar and Porte-Agel 2015, Fig. 5; Xie and Archer 2017 Fig. 4). The paper cited to support the statement about the upper rotor peak (Bodini et al. 2021) refers to a figure for one location at two instantaneous times during which the maximum was at about 100 m, for a hub*

*height of, I believe, 91 m. This is not a big deal by any means, but I would prefer to read that the maximum wind speed deficit is near hub height.*

**We have changed this sentence to read "…with the strongest deficit generally occurring near hub-height (Abkar and Porté-Agel, 2015) or in the top half of the turbine rotor disk (Vanderwende et al., 2016)."**

*2. L. 50: By contrast, here the authors mention that the peak in TKE occurs "in the upper part of the rotor" with no reference. Generally the peak in TKE is actually near the rotor tip (e.g., Abkar and Porte-Agel 2015, Fig. 8; Lu and Porte-Agel 2015, Fig. 12). No big deal, but it would be more accurate to state that the peak in TKE is generally near the rotor tip.*

**We have changed this sentence to read "The most pronounced enhancement in TKE typically occurs in the upper half of the turbine rotor disk near the rotor tip where the largest shear occurs (Abkar and Porté-Agel, 2015; Vanderwende et al., 2016)."**

*3. L. 139: How can the temporal resolution of the WRF model be 10 minutes? Perhaps you mean the resolution of the output, not that of the NOW_WAKES simulations?*

**We have clarified that the output resolution of these simulations was ten minutes: "...at 10-minute output resolution; we used hourly time steps for our analysis."**

*4. L. 150: How are the wind turbines distributed in the grid cells (i.e., one or two per grid cell)? What layout was used (if regular, along which directions)? Please add a few sentences to clarify this, since it becomes important later in the discussion of the added TKE spatial patterns.*

**We have clarified "This spacing results in generally one turbine per cell (in 1018 cells), but 177 cells have two turbines, 6 cells have three turbines, and 7 cells have 4 turbines."**

*5. L. 163: There appears to be confusion between CTKE and α. It is α that is 1.0 in the case of 100% added TKE, not CTKE, as CTKE is not constant and not equal to one.*

**Thank you for catching this typo; we have corrected this to "separate simulations with 0% (\alpha = 0) and 100% (\alpha= 1.0) added TKE are available"**

*6. L. 164: unfinished sentence? Perhaps it should be: "default, since comparisons with large-eddy simulations by Vanderwende et al. (2016) and Garcia-Santiago et al. (2024) suggest ...".*

**Thank you, we have incorporated this suggestion.**

*7. L. 185: How do you determine stability here? One value for all lease areas or one value per grid cell? I believe you talk about it later (Eq. 4); if it's the same approach, then perhaps tell the reader that the info about stability will be discussed shortly.*

**We have added a sentence "(A discussion of how atmospheric stability is determined is found below in Section 2.2.)."**

*8. Eq. 4: I am confused about the heat flux here. Since it is based on θ $_0$, it is a potential temperature flux, thus not the same as the heat flux that is discussed, for example, in Section 4.4. How do you calculate it? Or, is L a direct output of the WRF? If so, please state it.*

**While the inverse Obukhov length can be directly output from WRF, our previous work has shown that there are differences between the values output from WRF and calculated directly from fluxes. We have added a sentence "We note that while the inverse Obukhov Length can be output directly from WRF, we calculate it directly from the fluxes as discussed in Quint et al. (2025)." There is no meaningful difference between the potential temperature flux and the heat flux when considering one shallow model level at sea level.**

*9. L. 250: also Golbazi et al. (2022) reported and discussed the same numerical noise issue.*

**We have included a reference to Golbazi et al. in this sentence:**

**"The deficits at these remote locations are presumed to be numerical noise as identified in Ancell et al. (2018); Lauridsen and Ancell (2018) and discussed in Appendix F of Rosencrans et al. (2024) and by Golbazi et al. (2022)."**

*10. Figure 7: How do you ultimately exclude the light-blue numerical patterns in c)?*

**One of the requirements is that the "wake" must be part of a cluster that includes the actual wind turbines. Because those light-blue areas in Fig7c are parts of clusters that are not contiguous with the cluster of the locations with wind turbines, their clusters are not identified as the wake cluster (only one wake cluster can be identified at each time step). The description of the method has been expanded to clarify this:**

**"In Fig. 7b, the small clusters -1, 0, 1, and 3-7 in the east of the domain are not contiguous with the wind turbines
and lack 15% of their points within the wind plant, so they are excluded from consideration as the wind plant wake."**

*11. Figures 9–22: It is difficult to discern the boundaries of the wind farms when they are grey-shaded but hidden below the shades of whichever field is being plotted. Would it be possible to replace the grey-shaded areas with black contours? This is not a requirement, just a suggestion.*

**Thank you for the suggestion. We have experimented with this, but the outlines of the wind plants become too complicated and distracting.**

*12. Figure 10: What is the small grey shape in the middle of e)?*

**This is a plotting error where the negative values exceed the plotting range. We have incorporated a new figure with an expanded range to avoid this problem. We provided both a new Fig 9 and Fig 10 to facilitate comparison between them.**

*13. L. 309: I would also conclude that the hub-height wind speed deficit is basically insensitive to the amount of added TKE, consistent with Archer et al. (2020), Fig. 7, as long as TKE advection is on.*

**There are some small differences between the LA100 and LA0 simulations so we would not want to state that the hub-height wind speed deficit is insensitive to the amount of added TKE.**

*14. L. 341: The sentence starting with "At 10 m" is a repetition of what was already discussed at L. 316–324. Consider removing it.*

**Because this section includes references to the previous literature, we would like to keep it in.**

*15. L. 356: Similarly, the sentence starting with "Increases in TKE" is redundant.*

**We have removed the sentence "Increases in TKE are localized to the cells with turbines, with the strongest values occurring where turbine density is the greatest."**

*16. Section 4.2.2: An important finding to add is that TKE at the surface is reduced in neutral and stable (fast-wind) conditions in the wakes regardless of the amount of TKE added.*

**We have added a sentence "Finally, regardless of the amount of added TKE, wake regions outside of the wind plant all show slight decreases in surface TKE (Fig. 15 b, e, and f and Fig. 16b, e, and f)."**

*17. General: The manuscript now provides excellent evidence of the links between TKE and 2-m temperature through the heat flux changes. Figures 15–20 now tell a coherent story. Well done.*

**Thank you for your supportive comments.**

*18. L. 408: I think that the discussion could be simplified and easier to explain and understand if the authors used the magnitude of the heat fluxes, rather than expressions such as "more negative heat flux". Here is what I mean. The magnitude of the heat flux is directly related to the amount of added TKE near the surface. Regardless of the sign of the heat flux, with more TKE the magnitude increases and with less TKE the magnitude decreases. Thus, in stable conditions, we know from Fig. 15 that TKE at the surface increases in the wind farms and*

*decreases downwind in their wakes; therefore the magnitude of the heat flux is larger inside the farms and smaller in the wakes. Therefore, in stable conditions with negative heat fluxes, this causes warming inside the wind farms (large negative minus small negative equals small negative, thus downward and warming) and cooling downstream (small negative minus large negative equals small positive upward, thus cooling). This is now beautifully demonstrated in the paper. Since added TKE is 100%, the warming inside the farm is somewhat strong, because there is a lot of added TKE there. With 0% added TKE, we do not see the warming inside the wind farms because there is no added TKE there, but the reduction in the wake remains, thus the cooling and the positive heat flux difference (i.e., reduction of the heat flux magnitude, thus less warming coming down, thus cooling).*

**We have revised the text as follows:**

**As with the 2-m temperature changes, heat flux changes induced by wind plants are small but vary with stability and with the amount of turbine-added turbulence (Figs. 19 and 20). The Appendix includes the average values for the heat fluxes for the NWF simulations (Fig. C1), the LA100 simulations (Fig. C2), and the LA0 simulations (Fig. C3) to emphasize the subtle nature of the wind-plant induced changes. Heat fluxes are least modified by the wind plants in neutral conditions (Figs. 19b and 20b), with changes of less than 1 W m −2 throughout the region. In unstable conditions (Figs. 19c and 20c), with the introduction of wind plants, wind plants slightly reduce the upward heat flux. The reduction is about 3 W m −2 for LA100 and 5 W m −2 for LA0.**

**In stable conditions, the effects show more spatial variability and are strongly linked to the amount of turbine-added turbulence. The ambient (NWF) background heat flux is slightly negative, as the water is typically colder than the air (Fig. C1).**

**Within the wind farm, for the LA100 simulations, the magnitude of the heat flux increases (Fig. 19 d,e,f). This stronger downward heat flux leads to warming inside the wind farm (Fig. 17 d,e,f). In contrast, in the LA0 stable cases (Fig. 20a,d,e,f), where turbine-added turbulence is absent, there is no warming inside the wind farms. The reduction in TKE results in cooling in the wake region (Fig. 18 d,e,f), consistent with the increase in heat flux magnitude downwind (Fig. 20 d,e,f). These results are in agreement with Golbazi et al. (2022), demonstrating how changes in surface TKE impact heat flux magnitudes, ultimately influencing surface temperature patterns both within and downstream of the wind farms.**

*19. L. 434–439: Can we conclude that 0% added TKE is unphysical? Typos and style*

**Thank you for catching typos, we have revised to**
**"In contrast, the stable conditions of the LA0 simulations show subtle decreases in boundary-layer height (Fig. 22a,d,e,f) because these simulations lack turbine-added turbulence that would nudge the boundary layer up. In these cases without added turbulence, the turbine-induced drag prevents the increase of PBL height. The slower**

winds in the wake may also reduce the development of the boundary-layer height, thereby reducing it. Only in the neutrally-stratified cases are there possible increases in boundary layer height, albeit constrained to 20 m or so. Observations (Abraham et al., 2024) and large-eddy simulations (Wu and Porté-Agel, 2017; Gadde and Stevens, 2021) indicate that the boundary-layer height should increase in the wake, so the LA0 simulations fail to represent a critical physical process.”

20. L. 3: The fist sentences of the abstract are about potential impacts of the future wind farms, because they are not built yet and therefore we cannot know for sure. As such, terms like “may” (L. 1) and “potential” (L. 4) are correctly used to indicate uncertainty. But on L. 3, the strong future tense “will” is used (“how future plants will change local meteorology”). I recommend the use of “may” instead of “will”, given that any single sentence, out of context, can potentially be used to exaggerate such impacts.

Thank you, corrected “will” to “may” in the abstract.

21. L. 10 (and many other instances): one should write either “temperature at 2 m” or “2-m temperature”, with a dash “-” between 2 and m. Same for “2-km resolution” at L. 137 or “10-m wind speed” at L. 173.

We also prefer this style as recommended by the reviewer, but had recently had another paper in our research group in Wind Energy Science be corrected by a technical writer at Copernicus to not use that style. And so we use the (what we consider) awkward and incorrect approach as appears in the paper.

Unfortunately, according to https://publications.copernicus.org/for_authors/manuscript_preparation.html#math, “It is our house standard not to hyphenate modifiers containing abbreviated units (e.g. "3-m stick" should be "3 m stick"). This also applies to the other side of the hyphenated term (e.g. "3 m long rope", not "3-m-long rope").”

For the record, we disagree with this style but for many other reasons would like to publish this paper in Wind Energy Science, so we conformed to the journal requirements.

22. General: acronyms are sometimes defined but not used, or not needed, or not defined. For example, WRF is not defined but used at L. 122 (it was originally defined at L. 92 but then erased for some reason). PBLH is defined at L. 81 but not used in the next two sentences. IBL is not really needed (L. 84).

The acronym for WRF is defined in the abstract but we have redefined it in the body. While PBLH is not used in the next two sentences, “PBL” is used extensively in the following paragraph as is IBL. PBLH is also used later in the paper. We have revised to:

**"In some circumstances, wind turbines' effects can also influence a deeper region of the atmosphere. The planetary boundary layer (PBL) height (PBLH) is the height of the layer of the atmosphere that is influenced by the surface. Wind plants increase the boundary-layer height in stable (Lu and Porté-Agel, 2011) and neutral (Wu and Porté-Agel, 2017) conditions. As wind turbines extract energy from incoming flow, kinetic energy is entrained from the flow above, resulting in an increased boundary layer depth. An internal boundary layer (IBL) develops above the wind plant as relatively slower flow at rotor height moves upwards, interacting with slightly faster air. The IBL grows; it may eventually interact with the free atmosphere and modify the PBLH (Wu and Porté-Agel, 2017; Gadde and Stevens, 2021). The growth of the IBL …"**

*23. General: Remove the space between 100 and %, thus "100%" (like at L. 108), not "100 %" (like at L. 111). Same for 0% and 25%.*

**The space appears because our LaTex includes a "~" to ensure that the number and the percentage sign are kept together (i.e. are not broken across lines). While the space is a little ugly, it would be uglier for the number and the percentage sign to be broken across lines.**

*24. L. 167: weird use of "of". Perhaps rephrase as: "too-small values of turbulence and too large decreases in wind speed".*

**We have rephrased to "with too small values of turbulence and too large of decreases in wind speed."**

*25. L. 217: mispelling of "calculated".*

**Corrected, thank you.**

**Response to Second Reviewer**

*Thank you to the authors for addressing the 1st review comments.*

*I have been through the revisions and the rebuttal and have annotated these in the pages below.*

*Overall, I think the paper is much improved, but still some small revisions remain to be made, in my opinion (see below). I think the paper now acknowledges more clearly the modelling scope of the study in respect to being a "one of" many possible configurations of the set-up. The tracked changes version uploaded, appears not to feature all the changes made, so this makes it a bit harder to determine the actual changes made.*

**Thank you for your careful review which has improved the manuscript. We have responded to all the "Review 2.2" comments below.**

*Sections in this document are prefixed in the following way.*
*Review 2.1: = Reviewer's original comments appear in italics*
*Author rebuttal: = The authors response*
*Review 2.2: Review's comments to revision: = Reviewer view on the response.*

*Review 2.1:*
*General comments:*

*This article explores the impact on a range of meteorological fields by the presence of large wind farm cluster off the USA east coasts using the WRF mesoscale model and using the WFP wind farm parameterization, using the 100% added TKE option.*

*Overall I find the article outlines a repeatable methodical approach and describes results, but lacks clarity on the motivation for the investigation and lacks discussion on the limitations of the method. What conclusions are to be drawn beyond describing the response of a model (WRF) in these "no wind farm" and "with wind farm" simulations? What research question is being asked? What is the hypothesis being tested? Please revise to address this.*

*Author rebuttal:*
*We thank the reviewer for their time and consideration in reviewing our manuscript. We have added the following paragraph to the Introduction:*
*"Given the scarcity of comprehensive offshore observations along the U.S. East Coast, this study aims to complete the first year-long assessment of how modeled offshore wind plants influence the modeled local environment. We achieve this by comparing WRF model (Skamarock et al. 2021) simulations with and without wind plants included. Our analysis focuses on the Massachusetts-Rhode Island offshore wind lease area, where we quantify the difference in hub-height and 10-m wind speed, boundary-layer height, 2-m temperature, surface heat flux, and TKE at the surface and at hub height. Our expectation is to demonstrate that different stability conditions are a key driver of the simulated micrometeorological impacts, and that these impacts also vary with different wind speeds, as wind turbine operation changes. Furthermore, we aim to assess the relationship between boundary-layer height and the extent of wind plant wakes, hypothesizing that deeper boundary layers will limit the extension of these wakes."*
*As detailed in our other responses to the reviewer's comments, we have also added more discussion throughout the manuscript, including the conclusions.*

*Review 2.2:*
*OK – I suggest changing "to demonstrate that different stability conditions are a key driver" to "to demonstrate that within this \*model set-up\* different stability conditions are a key driver"*

**Thank you for the suggestion. However, such a change would suggest that the role of atmospheric stability in determining wake behavior is only an artifact of this particular model set up. Extensive evidence with other simulation codes also confirms that atmospheric stability mediates wake behavior and impacts, so we prefer to keep our sentence as is.**

*Review 2.1*
*The choice of one WRF set-up, one WRF wind farm parameterization, and one setting for the added TKE option, is a severe limitation of the article. It means that the whole paper becomes a description of model results, rather than focussing on what might actually happen in nature itself.*

*Author rebuttal:*
*While we agree that the question of what actually happens in nature itself is extremely interesting, a very limited set of observations are available to quantify the real-world impact of wakes. In the absence of extensive observations, modeling studies such as the one presented here are needed, and typically used, to fill the knowledge gap. The modeling tools used here have repeatedly demonstrated results consistent with available observations in other locations (i.e., the comparison of the modeling studies of Xia et al. with the observations of Zhou et al., the intercomparison of aircraft observations with modeling studies of Siedersleben et al. 2018, 2020, and the validation studies of Larsén and Fischerei 2021 and Ali et al. 2023). However, the micrometeorological impacts of wakes over a complete annual cycle in this region with intensively planned offshore wind development has not yet been investigated, hence the effort here.*

*Regarding the choice of one WRF set-up: this set-up was based on a comparison of 16 set-ups in this region (Bodini et al. 2024 ESSD) and was the best-performing set-up. We have added a sentence to the beginning of Section 2:*
*"We note that the WRF setup used here resulted from a comparison of 16 different WRF setups against an observational dataset; this setup was the best performer (Bodini et al., 2024)."*

*Review 2.2:*
*OK - perhaps just state the results are showing consistent response to what has been found before, but now utilizing a full year of simulation of this particular part of the world.*

**In several places, including in the conclusions, we point out that we use a full year of simulations. We also suggest (in the conclusions) that a multiyear study could provide more insights.**

*Review 2.1:*
*The manuscript needs to be revised to include a comprehensive discussion of the limitations of WRF- WFP, and what that might mean for the given results.*
*Author rebuttal:*

*We have added a discussion to the conclusion about the existing validation of WRF-WFP and other ongoing research efforts that might affect the given results.*

*"Of course, this study relies on the accurate representation of wakes in the Fitch WRF wind farm parameterization. While wakes simulated with this parameterization compare reasonably well with the limited sets of observations available (Lee and Lundquist, 2017; Siedersleben et al., 2018b, a, 2020; Ali et al., 2023; Larsén and Fischereit, 2021), the availability of observations of wake effects at multiple distances and heights from wind farms, especially offshore, is limited. Ongoing experiments such as AWAKEN (Moriarty et al., 2024) may provide more extensive datasets to support modifications to wind farm parameterizations in mesoscale models. Additionally, comparisons of these mesoscale representations to more finely resolved large-eddy simulations of wind farms (Vanderwende et al., 2016; Peña et al., 2022) may suggest other improvements, although these comparisons should be carried out for a range of atmospheric stability conditions and wind farm geometries. Particular attention should be paid to effects on surface meteorology as well as dynamics directly relevant to wind turbine power production".*

*Review 2.2:*
*OK. Also, there are other places where the set-up is now more broadly discussed.*

**No response required.**

*Review 2.1:*
*Please include more justification for the model set-up, for example, why only a one year simulation? How might a longer period or different year impact the results?*
*Author rebuttal:*
*While of course a longer set of simulations would be interesting, the work presented here includes a complete annual cycle, which goes far beyond other wake studies in this region, i.e., 55 days (Pryor et al. 2021) or three months (Golbazi et al. 2022) investigated in previous work. This particular year was chosen, as discussed in Rosencrans et al. (2024), because of the availability of lidar data for validation of the no-wind-farm simulation.*
*Because this year includes a range of stability conditions and wind speeds typical for this region these results are not particularly sensitive to the choice of this particular year.*

*Review 2.2:*
*Yes, OK. But still perhaps write at some point what could be done in terms of making a study that was more climatologically representative.*

**In the conclusions we mention that "a multi-year study could provide more insights into interannual variability and how it might affect wakes."**

*Review 2.1:*
*I think there is a lack of physical mechanisms, and where mechanisms are conjectured, no model fields are used to back these up (see specific comments).*
*Author rebuttal:*

*We list below the main physical explanations we provide in the paper about the (modeled) changes in atmospheric variables. We have added some and expanded most of the existing ones, on top of what already discussed in the specific comments below:*

● *Hub-height wind speed:*

*"Within the wind plant, wind speeds are reduced by up to 2.7 m s-1 in stable conditions, and up 1.5 m s-1 in unstable conditions, as turbines extract momentum from the flow." "During unstable conditions, wind speeds are replenished faster due to increased mixing from aloft (Abkar et al. 2013), which reduces the extent of the wake"*

● *10-m wind speed:*

[revised manuscript text omitted]

*Review 2.2:*
*OK. But instead of "remote" use the term "distant from wind farms". Otherwise "remote" is rather vague term.*

**We have added "In contrast, distant from the wind farm, the turbine drag component causes the downwind surface cooling by reducing shear and promoting near-surface thermal stratification."**

*Review 2.1:*
*The paper several times states where results confirm what is already published, as a reader I would like more clarity on what are the most novel parts of the study and what led to these novel parts being of interest for investigation. Please revise to address this.*

*Author rebuttal:*
*As noted above, we have introduced a section in the introduction clearly stating the novelty of the study in looking at variability of wake impacts over an entire annual cycle. We have also revised the conclusions to emphasize the annual variability as well as the machine-learning approach demonstrated here for wake area and wake length characterization:*

*"We also develop and demonstrate a machine-learning approach to identify wind plant wakes, and use this method to demonstrate the relationship between boundary-layer height and both the area and length of the wind plant wake."*

*Review 2.2:*
*OK.*

**No response necessary**

*Review 2.1:*
*Latter sections seem a bit rushed.*
*Author rebuttal:*
*If the reviewer is requesting additions to the conclusion, we have expanded the conclusions and discussion of the results therein considerably.*

*Review 2.2:*
*OK.*

**No response necessary**

*Review 2.1:*
*Adding to the limitation discussion, it would be good to include what would be good further studies to pursue, and what might be an approach to the difficult question of validation. Please revise to address this.*
*Author rebuttal:*
*As noted above, we have added a discussion to the conclusion about the existing validation of WRF- WFP and other ongoing research efforts that might affect the given results.*
*"Of course, this study relies on the accurate representation of wakes in the Fitch WRF wind farm parameterization. While wakes simulated with this parameterization compares reasonably well with the limited sets of observations available (Lee and Lundquist, 2017; Siedersleben et al., 2018b, a, 2020; Ali et al., 2023; Larsén and Fischereit, 2021), the availability of observations of wake effects at multiple distances and heights from wind farms, especially offshore, is limited. Ongoing experiments such as AWAKEN (Moriarty et al., 2024) may provide more extensive datasets to support modifications to wind farm parameterizations in mesoscale models. Additionally, comparisons of these mesoscale representations to more finely resolved large-eddy simulations of wind farms (Vanderwende et al., 2016; Peña et al., 2022) may suggest other improvements, although these comparisons should be carried out for a range of atmospheric stability conditions and wind farm geometries. Particular attention should be paid to effects on surface meteorology as well as dynamics directly relevant to wind turbine power production."*

*Review 2.2:*
*OK.*

**No response necessary**

*Review 2.1:*
*Specific comments:*

*L16: "exceeding 100 m" -> "exceeding 1000 m"?*

*Author rebuttal:*
*We have removed numbers from this sentence to avoid ambiguities.*

*Review 2.2:*
*OK. But funny, I don't see that as a tracked change in the tracked changes version.*

**While we made this change, apparently it was made in a wrong version of the .tex file likely because we have multiple co-authors working on the revisions. The change is in the final version of the paper.**

*Review 2.1:*
*L52: "extreme scale" , suggest changing this term. "Extreme" 10 years ago is not "extreme" today.*
*Author rebuttal:*
*We have replaced it with "bigger".*

*Review 2.2:*
*OK. But funny, I don't see that as a tracked change in the tracked changes version.*

**Again, this is probably due to multiple co-authors working on the files. We have to keep one of the "extreme" references to match the language of Golbazi et al, but we have now replaced the second occurrence:**

**"While they find cooling at the surface for bigger turbines (using 25 % added TKE), we still find warming at …"**

*Review 2.1:*
*L81: The sentence "determine … how .. influence the local environment", it should be reformulated to say this is modelled local environment being investigated, not the actual environment in nature.*
*Author rebuttal:*
*We have changed it to "modeled environment".*

*Review 2.2:*
*OK.*

**No response necessary**

*Review 2.1:*
*L107: Please detail more about what is meant by "the model produced unrealistic wind speeds, … ". Please describe and state what it is that is unrealistic.*
*Author rebuttal:*
*We have expanded this discussion of the literature:*

*"Vanderwende et al. (2016) suggest that added TKE is critical. In that study, when TKE generation within the wind farm parameterization is disabled, the model produced unrealistic wind speeds, wind directions, and turbulence as compared to large-eddy simulations, with too-small of values of turbulence and too large of decreases in wind speed."*

*Review 2.2:*
*Well. Ok, but change "unrealistic" to "quite different".*

**The sentence now reads "...the model produced very different wind speeds, wind directions, and turbulence as compared to large-eddy simulations,..."**

*Review 2.1:*
*L96 and Table 1: Why was this period chosen?*
*Author rebuttal:*
*We have rephrased as "NOW-WAKES covers from 1 September 2019 00:00 UTC - 31 August 2020 23:50 UTC (chosen to overlap with lidar data availability in the region) at 10- minute resolution".*

*Review 2.2:*
*OK.*

**No response necessary**

*Review 2.1:*
*L94 and Figure 1: Why was the domain chosen as it is? What is the reason for the far eastward extent?*
*Author rebuttal:*
*We have added a sentence explaining:*
*"The Rosencrans et al. (2024) domain is consistent with other datasets for this region (Xia et al., 2022; Redfern et al., 2021; Bodini et al., 2024) and was initially chosen to optimize processor partitioning for the WRF simulations."*

*Review 2.2:*
*I still think it strange to go so far east, is there more to explain here?*

**No, there is nothing to explain. We also felt like this was wasted space and wasted calculations but this domain shape was suited to the supercomputer on which these simulations were run.**

*Review 2.1:*
*Figure 3: It is strange to have a caption referring to a later caption.*
*Author rebuttal:*
*We have changed the captions of Figures 3, 4, and 5 accordingly.*

*Review 2.2:*
*OK.*

**No response necessary**

*Review 2.1:*
*L136: In the description of the BLH definitions, what happens in transitions from one stability condition to another, is there a discontinuity in the BLH? Could the authors use a sentence or to to justify the use of the approach of Olson et al (2019) for this analysis. What is the most relevant BLH determination for a wind farm do the authors think or recommend?*
*Author rebuttal:*
*Yes, during stability transitions, there may be discontinuities in the estimation of BLH by the WRF model because of the transition from one approach to another. Because these simulations use the MYNN PBL scheme, the authors recommend using the PBLH estimation approach included in that scheme (the Olson et al. (2019) approach) for consistency. This approach has performed well in comparison to observations for some case studies (Bauer et al. 2023).*

*Review 2.2:*
*OK.*

**No response necessary**

*Review 2.1:*
*Table 2: It might be better to have "region 1" and "region 2" also part of the wind speed column in this table, to remind the reader of the reasoning behind the wind speed partitioning.*
*Author rebuttal:*
*We have added references to Region 1, 2, and 3 in the Table.*

*Review 2.2:*
*OK.*

**No response necessary**

*Review 2.1:*
*L162: I am a bit wary about this statement about the "tight coupling" because it suggests that everything can be explained by atmospheric stability, but there may be very important other aspects of the profile, and these might be overlooked by this approach. Please expand on the justification of the approach.*
*Author rebuttal:*
*We have replaced it as "correlation" to soften the message of the sentence.*

*Review 2.2:*
*OK.*

**No response necessary**

*Review 2.1:*
*L166: "a leveling of the power production", I think a better term here would be "the rated power production being reached and not increasing further".*
*Author rebuttal:*
*Changed.*

*Review 2.2:*
*OK.*

**No response necessary**

*Review 2.1:*
*L167: "To isolate" , again similar to the L162 comment. It is not just wind speed that is varying, even though you keep stability and direction within a certain band. Please discuss other things in the profile that might vary, given this constraint on stability and direction.*
*Author rebuttal:*
*We have rephrased it as "To more clearly identify".*

*Review 2.2:*
*OK.*

**No response necessary**

*Review 2.1:*
*L175: Why is 1 m/s deficit chosen as the measure of a wake? Why not other measures, such as relative deficit? What are the advantages of this measure, what is the impact of different wind speeds (NWF simulations) on this wake definition?*
*Author rebuttal:*
*We chose an absolute (rather than relative) definition of the wake threshold to be consistent with previous work. A relative deficit requires comparison with spatially heterogeneous unwaked fields which can make the assessment of the wake even noisier than an absolute definition. In the text, we have added an explanation:*
*"This wake definition is stronger than the 0.5 m s-1 threshold used in Golbazi et al. (2022); Rybchuk et al. (2022); and Rosencrans et al. (2024), and was chosen to aid in identifying contiguous wakes. A relative wake definition proved problematic by making the wake field even noisier."*

*Review 2.2:*
*OK. It does bring focus to this noisy and ill-defined wake aspect, once again. It is a pity a relative wake cannot be used.*

**No response necessary**

*Review 2.1:*

*L178: Please explain why there are "not contiguous" wind speed perturbations, could they be related to the wake? How do you discount that there may be a distant response to the wind farm, perhaps oscillation in wake above and below the 1 m/s threshold that has been chosen.*

*Author rebuttal:*

*The WRF wind farm parameterization is known to produce noise in wind fields similar to these remote patterns. We have added a sentence:*

*"The deficits at these remote locations are presumed to be numerical noise as identified in Ancell et al. (2018); Lauridsen and Ancell (2018) and discussed in Appendix F of Rosencrans et al. (2024)."*

*Review 2.2:*

*OK. But instead of "remote" use the term "distant from wind farms". Otherwise "remote" is rather vague term.*

**We have changed the sentence to "but some other locations distant from wind plants, not contiguous with the wind plant wake, also show a wind speed deficit"**

**Also, "The deficits at these locations distant from the wind plants are presumed to be numerical noise as identified in Ancell et al.…"**

*Review 2.1:*

*L192: "ill defined" wakes. This seems a bit subjective to me, perhaps wakes are not neat and tidy as we might expect. Please justify. And is the 15.2% of hours with "ill defined" wakes not quite a significant share of the time?*

*Author rebuttal:*

*Of course we do acknowledge that this definition is by necessity somewhat subjective. To address this subjectivity and to enable the analysis to be replicated by other research groups, we have clearly defined the criteria used to make a distinction between the 85% clearly defined wakes and the 15% ill defined wakes. While 15% of the wakes is a not trivial share of the time, it is clearly a minority of the time. Further, it is consistent with other machine learning approaches used to identify wind turbine wakes in heterogeneous fields, such as 87.18% in Aird et al. (2021), although in Aird et al. they are identifying wakes from individual turbines and not wind farm wakes.*

*Review 2.2:*

*Well, I think the ill-defined cases may be just as important to understand. Are there any systematic meteorological situations in which the ill-defined case emerge in terms of stability (or anything else)? This could introduce systematic bias.*

**We have not been able to define a systematic meteorological situation in which these ill-defined wakes occur. Recall that we already only considered stably stratified conditions, which tend to occur with southwesterly wind, and the partitioning by wind**

speed is more or less flat for the 15% "failed" ("ill-defined" wake) cases (see below) as for the cases in which the wake fitting succeeds. For wind speeds at hub-height faster than 19 m s-1 there are more ill-defined wakes than well-defined wakes (Figure 1), which makes sense considering that wakes in such circumstances would be generated by very small thrust coefficients and would be subject to strong shear-driven mixing.

[Figure]

The histogram of QKE also shows that time periods with very strong QKE (in excess of 6 m2 s-2) have very few wakes and the few wakes that occur are ill-defined.

[Figure]

**We find no evidence of a systematic bias that would undermine our conclusions.**

*Review 2.1:*
*L211: Please can the authors explain why the wake is compared in wind speed across the different stability classes? Is the mean wind speed the same for the different stability classes, if not, the difference in wake deficit can be partly due to this effect.*
*Author rebuttal:*
*We have addressed the roles of wind speed and stability by partitioning our results by both stability and by wind speed within the stable stratification class. We first emphasize stability classes because of the long history of observations that wakes are stronger in stably stratified conditions (e.g., see the summary in section 2.3 of Porté-Agel et al. 2020, with a sample of over 20 investigations documenting wake variability with atmospheric stability). In Figure 6 of the current manuscript we already demonstrate that the wind speed and direction distributions are different for stable vs neutral vs unstable conditions, demonstrating that faster winds occur in stable conditions. By further partitioning the stable results into the different wind speed regimes, we identify the differences in wakes due just to wind speed variation.*

*Review 2.2:*
*OK. Has anything be changed in the text?*

**In the last revision, we broke out the differences in wind speed regimes in different ways, moving the 0-3 m s-1 stable conditions to the appendix and providing two separate regimes within Region 3 of the power curve.**

*Review 2.1:*
*L214: The authors write ""due to increased mixing from aloft", but this statement is not argued with data from the model, but appears to be more like a hypothesis for a possible, and plausible mechanism. Please justify the statement or rephrase it.*
*Author rebuttal:*
*We justified the statement by including a reference to the sensitivity of wake replenishment from above (Abkar and Porte-Agel 2013).*

*Review 2.2:*
*Well, then sentence still says "due to" but actually the result can only be said to be consistent with. Please update.*

**The sentence now reads "During unstable conditions, wind speeds are replenished faster consistent with increased mixing from aloft (Abkar and Porté-Agel, 2013), …"**

*Review 2.1:*
*L219: It would help the reader to refer to region 3 next to the "above 11 m/s".*
*Author rebuttal:*

*We have modified the sentence to read "Hub-height wind speeds are reduced by up to 2.5 m s-1 for wind speeds in Region 2 of the turbine power curve, and up to 3.6 m s-1 for wind speeds above 11 m s- 1 in Region 3 of the power curve."*

*Review 2.2:*
*OK.*

**No response necessary**

*Review 2.1:*
*L225-228: Does this effect also show when wind speeds are in the range 15 m/s - 25 m/s where the thrust is dropping significantly? See Fig 2b.*
*Author rebuttal:*
*Thank you for this suggestion. As seen in the comparison of e) and f) below, the magnitude of the wind speed deficit decreases for wind speeds faster than 15 m s-1. We have added this figure to the appendix and expanded the discussion in the text:*
*"Of note, when the wind speeds exceed 15 m s-1 when the thrust coefficient is very small, the wind speed deficit starts to decrease again (see Appendix Fig. B1)."*

*Review 2.2:*
*OK.*

**No response necessary**

*Review 2.1:*
*L233: Same question as above.*
*Author rebuttal:*
*Thank you also for this suggestion. As seen in the comparison of e) and f) below, the magnitude of the 10-m wind speed acceleration increases for wind speeds faster than 15 m s-1. We have added this figure to the appendix and expanded the discussion in the text:*
*"For wind speeds faster than 15 m s-1, the accelerations are more widespread within the wind plant but the maximum accelerations are not faster than those in the range of 11 - 15 m s-1 (see Fig. B2)."*

*Review 2.2:*
*OK.*

**No response necessary**

*Review 2.1:*
*L240: "reduced more", more than what? Does the deficit increase, or does the absolute wind speed reduce? It reads more like the latter, but I think it is the former.*
*Author rebuttal:*

*We have rephrased it to "Under stable conditions and southwesterly winds, the deficit in 10~m wind speeds increases more with increasing ambient wind speeds".*

*Review 2.2:*
*OK.*

**No response necessary**

*Review 2.1:*
*L251: Please quantify "increase slightly".*
*Author rebuttal:*
*We have rephrased to "At 10 m, wind speeds accelerate slightly (less than 1 m s-1 within the wind plant during…"*

*Review 2.2:*
*OK.*

**No response necessary**

*Review 2.1:*
*L260: Temperature increases by "around 0.05 degrees". Is this significant?*
*Author rebuttal:*
*As the topic sentence of this paragraph suggests, these changes are "small". We have added "only" to the sentence including "0.05 degrees".*

*Review 2.2:*
*OK.*

**No response necessary**

*Review 2.1:*
*L261-263: Are these statements conjecture or justified by model fields of fluxes? Please reformulate so it is clearer.*
*Author rebuttal:*
*These statements are not conjecture but are rather consistent with a wide body of literature discussing mixing mechanisms dating back to Baidya Roy et al. 2004 and demonstrated in Fitch et al. (2012). We have reformulated the sentences as follows:*
*"During stable conditions, turbines mix warmer air from aloft down to the surface, resulting in a temperature increase (as also discussed in Fitch et al. (2013) and Siedersleben et al. (2018a), among others). In unstable conditions, the boundary layer is already well mixed, so that any mixing by wind turbines is simply remixing a well-mixed layer."*

*Review 2.2:*
*OK.*

**No response necessary**

*Review 2.1:*
*L289: The heading "heat flux", please clarify what kind of heat flux is being looked at. Surface heat flux, vertical heat flux, sensible heat flux, etc, etc.*
*Author rebuttal:*
*We have renamed the heading to "Wind plant wake impacts on surface sensible heat flux".*

*Review 2.2:*
*OK. Curious about when you refer to "wind farms" or "wind plants".*

**This is an artifact of which co-author is making the revisions; there is no meaningful distinction between a wind farm and a wind plant. We have revised most "farms" to "plants" to be consistent with NREL style recommendations. When necessary for consistency with previous publications, we have kept "farm". In the first sentence we demonstrate how these terms are synonymous: "Wind plants, also known as wind farms, along the northeastern U.S. coast are projected to undergo rapid expansion in the coming years…"**

*Review 2.1:*
*L407: The use of the word "promote" infers a causal relationship, is that what is meant?*
*Author rebuttal:*
*Yes, we intend to suggest a causal relationship.*

*Review 2.2:*
*OK.*

**No response necessary**